# Rectified Flows for Fast Multiscale Fluid Flow Modeling

## Abstract

The statistical modeling of fluid flows is very challenging due to their multiscale dynamics and extreme sensitivity to initial conditions. While recently proposed conditional diffusion models achieve high fidelity, they typically require hundreds of stochastic sampling steps at inference. We introduce a rectified-flow framework that learns a time-dependent velocity field, transporting input to output distributions along nearly straight trajectories. By casting sampling as solving an ordinary differential equation (ODE) along this straighter flow field, our method makes each integration step much more effective, using as few as eight steps versus (more than) 128 steps in standard score-based diffusion, without sacrificing predictive fidelity. In addition, we develop a curvature-aware integration scheme that monitors local path straightness and adaptively regularizes the velocity and step size, improving stability and accuracy at essentially no training cost. Experiments on challenging multiscale flow benchmarks show that rectified flows recover the same posterior distributions as diffusion models, preserve fine-scale features that MSE-trained baselines miss, and deliver high-resolution samples in a fraction of inference time.

## 1 Introduction

Partial differential equations (PDEs) Evans (2010) underpin models from atmospheric dynamics to aerodynamics and MHD. High-fidelity solvers (finite difference/element, spectral) Quarteroni & Valli (1994) are often prohibitive for *many–query* tasks Quarteroni et al. (2015) such as UQ, optimization, and real-time prediction, especially for nonlinear regimes with strong sensitivity to data and multi-scale structure. In these settings one seeks the *statistical solution*, i.e., the push-forward of an input measure by the PDE solution operator Fjordholm et al. (2017). Monte Carlo ensembles built via repeated high-fidelity solves Lanthaler et al. (2021) rapidly become infeasible as ensemble size grows.

To mitigate cost, data-driven surrogates are explored Mishra & Townsend (2024). In particular, *neural operators* learn PDE solution operators directly and are widely used Kovachki et al. (2023); Bartolucci et al. (2023), with variants including FNO, GNO, DeepONet, Transolver, GNOT, geometry-informed operators, and UPT Li et al. (2020a;b); Lu et al. (2021); Wu et al. (2024); Hao et al. (2023); Li et al. (2023); Alkin et al. (2024).

In the context of chaotic PDEs, *GenCFD* Molinaro et al. (2024) introduced a conditional diffusion framework that preserves spectral content across fine scales, mitigating the "spectral collapse" of naive MSE-trained surrogates, see also Oommen et al. (2024); Gao et al. (2024a;b;c). However, sampling a score-based Diffusion model requires simulating a reverse-time stochastic differential equation (SDE) over 100+ discretization steps, each invoking a large ML model on a moderate-to-high resolution spatial grid, resulting in high runtimes that undermine real-time or resource-constrained applications.

To overcome this limitation of diffusion models, we adapt *rectified flows* Liu et al. (2022): a deterministic ODE-based analogue of diffusion that "straightens" transport paths in probability space. Rather than small random increments, rectified flows solve an ODE whose velocity field is trained to match the diffusion model's instantaneous score, but whose trajectories remain nearly linear. This structure permits large time integration, reducing the number of solver steps by an order of magnitude or more, while provably recovering the same target distribution.

Hence, our main contributions are:

- *Rectified flows for conditional PDEs.* We learn a straightening velocity field and sample via an ODE, enabling larger steps and far fewer evaluations than score-based diffusion.

- *Curvature-aware integration.* An EMA-based curvature proxy triggers a light Tikhonov blend and adaptive step sizes, stabilizing under shift and cutting steps; curvature is used for control rather than OOD detection Heng et al. (2024); Abdi et al. (2025).

- *Multi-scale accuracy at speed.* On 2D incompressible and compressible flows we match diffusion-class fidelity with 8–10 ODE steps (vs. $\geq 128$), up to $22\times$ faster, and reach comparable accuracy in $\sim$120k iterations.

## 2 APPROACH

**Modeling multi-scale flows and their statistical solutions.** We consider time-dependent PDEs of the form

$$\partial_t u(x,t) + \mathcal{L}\big(u, \nabla_x u, \nabla_x^2 u, \dots \big) = 0, \qquad x \in D \subset \mathbb{R}^d,\ t \in (0,T), \tag{1}$$

with boundary conditions $\mathcal{B}(u) = u_b$ on $\partial D \times (0,T)$ and initial data $u(x,0) = \bar{u}(x)$. The unknown $u : D \to \mathbb{R}^m$ evolves under the (generally nonlinear) operator $\mathcal{L}$. Canonical instances include the incompressible Navier–Stokes and compressible Euler equations.

A hallmark of nonlinear, multiscale flows is *extreme sensitivity* to initial and boundary data, yielding chaotic or near-chaotic behavior. Rather than a single trajectory from $u_0$, one studies an *initial measure* $\mu_0$ over states and its evolution under the PDE solution operator $\mathcal{S}^t$, i.e. the *statistical solution* $\mu_t = \mathcal{S}^t_\# \mu_0$. In strongly nonlinear regimes, $\mu_t$ spreads across function space and becomes highly structured across scales. Accurately approximating this evolving distribution is crucial for UQ and design in rough regimes, yet direct characterization with high-fidelity solvers is prohibitive when large ensembles are needed.

**High cost of computing statistical solutions.** Estimating $\mu_t$ by Monte Carlo requires *many* forward solves from perturbed initial conditions, each at high spatial resolution and over multiple physical time steps. The total cost scales linearly with ensemble size and quickly exceeds practical budgets even on modern HPC systems. This motivates *surrogate models* that can *sample* physically realistic final states (and statistics thereof) without enumerating every PDE realization. Concretely, we ask:

*How can we efficiently approximate statistical quantities of the pushforward $\mathcal{S}^t_\# \mu_0$ for highly sensitive solution operators $\mathcal{S}^t$?*

**Rectified flows for faster sampling.** To mitigate these inefficiencies, we adopt the framework of *rectified flows* Liu et al. (2022): a deterministic alternative to conditional diffusion designed to reduce the number of required integration steps. In essence, rather than generating samples via small random "diffusion-reversal" increments, rectified flows construct an *ordinary differential equation* (ODE) with *nearly straight* trajectories in function space. This straighter transport map can be integrated more aggressively in time, enabling sampling with far fewer solver steps. Importantly, rectified flows retain the desirable property of matching the same final distributions that standard diffusion models learn, but at a fraction of the runtime cost.

## 3 RELATED WORK

### 3.1 DIFFUSION MODELS IN FLUID DYNAMICS

**Conditional diffusion surrogates.** *GenCFD* Molinaro et al. (2024) trains a conditional score-based diffusion model that starts from an initial or low-resolution flow field and iteratively refines it through hundreds of denoising steps, thereby capturing multi-scale turbulent structure. Similarly, the method of Oommen et al. (2024) conditions a diffusion model on a coarse fluid solution (generated by a learned operator) and "super-resolves" the fine-scale flow. Given these similarities, we refer to such diffusion-based solvers collectively as *conditional diffusion PDE solvers*.

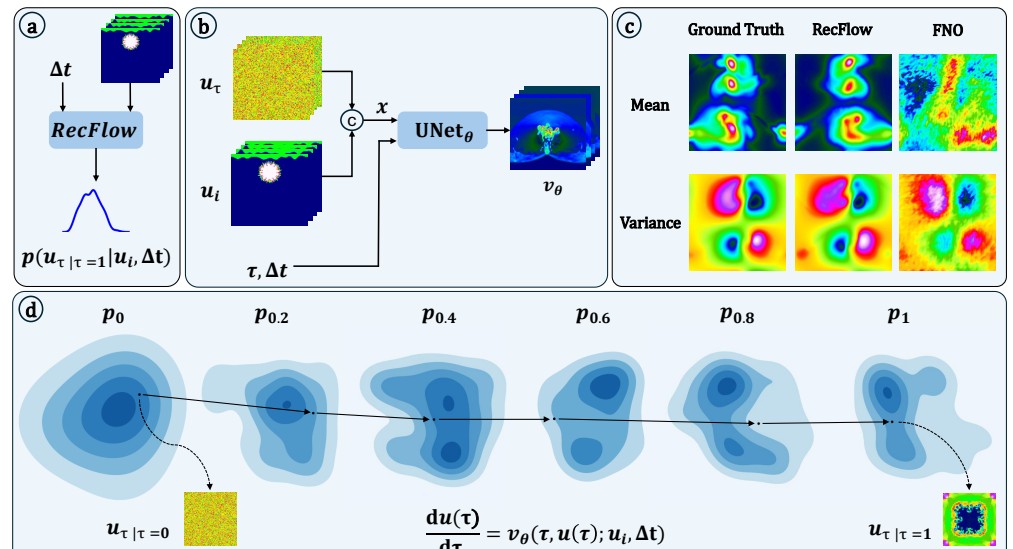

Figure 1: **Overview of RecFlow.** (a) Goal: match high–order flow statistics (e.g., pointwise $W_1$) while keeping inference cheap by transporting $p(u_{\tau=0} \mid u_i, \Delta t)$ to $p(u_{\tau=1} \mid u_i, \Delta t)$ along nearly straight ODE trajectories. (b) Architecture: a UViT predicts the velocity field $v_\theta$, conditioned on Gaussian noise, the initial state $u_i$ (and boundary data), diffusion time $\tau$ and physical lead time step $\Delta t$; conditioning signals are concatenated to the inputs. (c) Qualitative results on the cylindrical shear–layer dataset. (d) Evolution of the push-forward $p_\tau = (X_\theta(\tau))_\# \, p(u_{\tau=0} \mid u_i, \Delta t)$ visualized via PCA+KDE, illustrating near-linear paths from noise ($\tau{=}0$) to target ($\tau{=}1$) on Richtmyer–Meshkov. RecFlow solves a deterministic ODE with an approximately constant velocity field, enabling large steps and straight-line inference.

**Rectified flow alternative.** Each sample typically needs hundreds of network calls, limiting large-scale or real-time use. Our method replaces the stochastic SDE with a deterministic ODE whose nearly straight trajectories require far fewer steps, achieving similar statistical fidelity at much lower inference cost (see Figure 2 and **SM** 5). Crucially, we find that rectified flows are actually **more efficient** from the standpoint of training-time required to reach the same accuracy level with the conditional diffusion approaches; this is also confirmed by earlier work Esser et al. (2024) exploring the scalability of rectified flows with transformer-based backbones – we expand on this empirical phenomenon in **SM** D, **SM** D.2.

## 4 RECTIFIED FLOWS, CURVATURE-AWARE INTEGRATION, AND ERROR CONTROL

**Rectified flows in a nutshell.** Let $\mathcal{U}$ be the divergence-free function space of interest and fix a Galerkin surrogate $\mathbb{R}^{d_N} \subset \mathcal{U}$ (separable, Polish), so all constructions are finite-dimensional. For source/target measures $\mu_0, \mu_1 \in \mathcal{P}(\mathcal{U})$ and a coupling $\gamma \in \mathcal{P}(\mathcal{U} \times \mathcal{U})$ with marginals $\mu_0, \mu_1$, define the linear interpolants

$$T_\tau(u_0, u_1) = (1-\tau)u_0 + \tau u_1, \qquad \rho_\tau = (T_\tau)_\# \gamma, \qquad \tau \in [0,1].$$

The *barycentric* (rectified) velocity is the disintegration-average

$$v_\star(u, \tau) := \int_{T_\tau^{-1}(u)} (u_1 - u_0) \, \gamma_u^\tau(du_0, du_1),$$

and sampling proceeds by the *deterministic* ODE

$$\dot{u}_\tau = v_\star(u_\tau, \tau), \qquad u_0 \sim \mu_0. \tag{2}$$

In practice we learn a parametric $v_\theta$ via the squared-loss regression

$$\min_\theta \int_0^1 \mathbb{E} \left\| (U_1 - U_0) - v_\theta\big((1-\tau)U_0 + \tau U_1, \ \tau\big) \right\|^2 d\tau, \tag{3}$$

so that $v_\theta(u, \tau) \approx \mathbb{E}[U_1 - U_0 \mid U_\tau = u]$.

**EMA proxy, curvature score, and correction.** During sampling we write $v_t := v_\theta(u_t, \tau)$ for the current velocity prediction and maintain a low-pass proxy via an exponential moving average

$$v_t^{\text{ema}} = \lambda\, v_{t-\Delta\tau}^{\text{ema}} + (1 - \lambda)\, v_t, \qquad \lambda \in (0, 1). \tag{4}$$

The instantaneous *curvature (straightness) score*

$$s_t := \|v_t - v_t^{\text{ema}}\|_2$$

acts as a scale-adapted indicator of how rapidly the velocity field bends along the trajectory. When curvature grows, we gently regularize the update by convexly blending the raw prediction with its EMA trend,

$$\tilde{v}_t = (1 - \alpha_t)\, v_t + \alpha_t\, v_t^{\text{ema}}, \qquad \alpha_t = \lambda_c\, \phi\!\left(\frac{s_t}{\sigma_s}\right) \in [0, \lambda_c], \tag{5}$$

where $\phi$ is a saturating squashing map (e.g. hard-tanh) and $\lambda_c \le 1$ caps the correction. This update is the closed-form minimizer of a Tikhonov-regularized quadratic that trades fidelity to $v_t$ against smoothness toward $v_t^{\text{ema}}$.

**Adaptive step control.** We couple the blend with an adaptive step that scales inversely with the observed curvature to keep the local truncation error under control:

$$\Delta\tau_t \propto \frac{1}{\sqrt{\kappa_1\, s_t + \kappa_2}}, \qquad \Delta\tau_t \in [\Delta\tau_{\min}, \Delta\tau_{\max}], \tag{6}$$

with $\kappa_2$ absorbing baseline Lipschitz effects (e.g. $\kappa_2 \approx L\,\|v_t\|$). When $s_t$ is small, $\alpha_t \approx 0$ and the method takes larger steps; as curvature increases, $\alpha_t$ rises smoothly and $\Delta\tau_t$ contracts. The state update is

$$u_{t+\Delta\tau_t} = u_t + \Delta\tau_t\, \tilde{v}_t.$$

### 4.1 ERROR DECOMPOSITION AND BOUNDS

We now quantify why equation 4–equation 6 help. Write the *ideal* and *learned* flows

$$\dot{u}_\tau = v_\star(u_\tau, \tau), \qquad \dot{\hat{u}}_\tau = v_\theta(\hat{u}_\tau, \tau),$$

with the same $u_0$. Assume $v_\theta(\cdot, \tau)$ is $L$-Lipschitz in $u$ and $v_\star \in L^2$.

**Theorem 1** (Terminal error decomposition). *Let $U_\tau$ solve $\dot{U}_\tau = v_\star(U_\tau, \tau)$ with $U_0 = u_0$, and let $\hat{u}_\tau$ solve $\dot{\hat{u}}_\tau = v_\theta(\hat{u}_\tau, \tau)$ with $\hat{u}_0 = u_0$. Define the time–averaged velocity $\bar{v}_\theta(u) := \int_0^1 v_\theta(u, \tau)\, d\tau$ and*

$$\varepsilon_{\text{fit}}^2 := \int_0^1 \mathbb{E}\big\|\bar{v}_\theta(U_\tau) - v_\star(U_\tau, \tau)\big\|^2\, d\tau, \qquad \varepsilon_{\text{curv}}^2 := \int_0^1 \mathbb{E}\big\|v_\theta(U_\tau, \tau) - \bar{v}_\theta(U_\tau)\big\|^2\, d\tau.$$

*Then, for the continuous–time flows,*

$$\mathbb{E}\big\|\hat{u}_1 - u_1\big\| \le e^L\left(\varepsilon_{\text{fit}} + \varepsilon_{\text{curv}}\right). \tag{7}$$

**Interpretation.** The terminal error splits into a *reconstruction* term ($\varepsilon_{\text{fit}}$) and a *straightness* term ($\varepsilon_{\text{curv}}$). Blending equation 4 shrinks the within-time variation, directly tightening the bound; the adaptive stepping in equation 6 triggers corrections where the curvature proxy $s_t$ spikes.

**Lemma 1** (Local truncation error for explicit Euler on $\dot{u} = v(u, \tau)$). *Assume $v$ is $C^1$ in both arguments with $\|\partial_\tau v(u, \tau)\| \le M_\tau$, $\|J_u v(u, \tau)\| \le L$, and $\|v(u, \tau)\| \le M$ on the region traversed. One explicit-Euler step from $(u, \tau)$ with size $\Delta\tau$ has LTE*

$$\text{LTE}(\tau; \Delta\tau) = \frac{(\Delta\tau)^2}{2}\big\|\partial_\tau v(u, \tau) + J_u v(u, \tau)\, v(u, \tau)\big\| + \mathcal{O}((\Delta\tau)^3) \le \frac{(\Delta\tau)^2}{2}(M_\tau + LM) + \mathcal{O}((\Delta\tau)^3).$$

**From Lemma 1 to the step rule.** The leading LTE term depends on $\|\partial_\tau v\|$ and $\|J_u v\, v\|$. The EMA deviation $s_t = \|v_t - v_t^{\text{ema}}\|$ is a computable proxy for the (local) time variation $\|\partial_\tau v\|$ (up to a scale factor tied to the EMA bandwidth), while $L\,\|v_t\|$ upper-bounds the spatial contribution. Enforcing LTE $\lesssim$ tol yields the square-root rule equation 6, with $\kappa_1$ absorbing the EMA bandwidth and $\kappa_2 \approx L\|v_t\|$. Under blending, $\|\tilde{v}_t - v_t^{\text{ema}}\| = (1 - \alpha_t)s_t$, so the curvature term and the required step shrink *together*, stabilizing integration where needed and enlarging steps elsewhere. See **SM** A.4 for proofs and additional theoretical context.

## 4.2 IMPLEMENTATION DETAILS FOR PDEs

**Learning velocity fields in PDE settings.** We use two regimes that share one implementation: (i) a *terminal* mapping with pairs $(u_i, u_f)$ (final snapshot), and (ii) an *all-to-all* physical-time mapping with pairs $\big(u(t), u(t + \Delta t)\big)$. In both cases the network is conditioned on the *source* field $u_{\mathrm{src}}$ and the *lag* $\Delta t$ only (not the absolute time), while diffusion time $\tau \in [0, 1]$ controls the rectified transport straightening. We embed $\Delta t$ with $\Phi(\Delta t)$ (sinusoidal + MLP) and use a separate time embedding $\Gamma(\tau)$ for diffusion time. The target inside the rectified-flow objective is always the displacement $u_{\mathrm{tgt}} - u_{\mathrm{src}}$.

---

**Algorithm 1** Rectified-Flow Training with Lag-Only Physical-Time Conditioning

---

**Require:** Trajectories $\{u(\cdot\,; \omega)\}$ or pairs $\{(u_i, u_f)\}$; velocity model $v_\theta$; diffusion noise schedule $\sigma(\tau)$; lag embedding $\Phi(\cdot)$; diffusion-time embedding $\Gamma(\cdot)$

1: **for** each iteration **do**
2:     **if** TERMINAL regime **then**
3:         Sample $(u_i, u_f)$ and set $u_{\mathrm{src}} \leftarrow u_i$, $u_{\mathrm{tgt}} \leftarrow u_f$, $\Delta t \leftarrow T$
4:     **else**                                                             $\triangleright$ ALL2ALL regime
5:         Sample a trajectory $u(\cdot\,; \omega)$
6:         Sample $t \sim \mathcal{U}(0, T - \Delta t_{\max})$,   $\Delta t \sim p_\Delta$ with $0 < \Delta t \le \Delta t_{\max}$
7:         Set $u_{\mathrm{src}} \leftarrow u(t; \omega)$,   $u_{\mathrm{tgt}} \leftarrow u(t + \Delta t; \omega)$
8:     **end if**
9:     Sample diffusion time $\tau \sim \mathcal{U}(0, 1)$ and noise $\xi \sim \mathcal{N}(0, I)$
10:   Form rectified state:   $x_\tau \leftarrow \tau\, u_{\mathrm{tgt}} + \sigma(1 - \tau)\, \xi$
11:   Conditioning:   $c \leftarrow \big(u_{\mathrm{src}},\ \Phi(\Delta t),\ \Gamma(\tau)\big)$
12:   Loss:
$$\mathcal{L}(\theta) \;=\; \big\|\, (u_{\mathrm{tgt}} - u_{\mathrm{src}}) \;-\; v_\theta\big(x_\tau,\, c\big) \big\|^2$$
13:   Update $\theta$ by gradient descent
14: **end for**

---

**Overview of the model.** For brevity, we suppress explicit physical-time notation and present the simplified terminal setting $(u_i \to u_f)$, with the understanding that the architecture extends straightforwardly to lag-conditioned pairs.

Our network learns a velocity (or denoising) map

$$v_\theta(u_\tau, u_i, \tau) \;=\; \mathrm{UNet}_\theta\big(u_\tau, u_i,\ \Gamma(\tau)\big),$$

where $u_\tau$ is a low-frequency representation of $u_f$, $u_i$ is the initial datum, and $\Gamma(\cdot)$ is an MLP-based time embedding:

$$\Gamma(\tau) \;=\; \mathrm{MLP}\big(\mathrm{Sinusoid}(\tau)\big).$$

Within the U-Net (see **SM** A.1), we alternate residual blocks with multi-head attention layers, downsampling to a bottleneck and then upsampling via skip connections. A final $1 \times 1$ projection recovers either the field increment $\hat{u} \approx u_f - u_i$ or a noise residual. By injecting the rectification time $\Gamma(t)$ into each block, the model performs a fully deterministic trajectory integration instead of stochastic diffusion.

**Sampling procedure.** Once trained, we generate new samples by numerically integrating the ODE

$$\frac{du_\tau}{d\tau} = v_\theta\big(u_\tau,\, u_i,\, \tau\big), \quad u_\tau|_{\tau=0} = \xi, \quad \xi \sim \mathcal{N}(0, I).$$

Any standard ODE solver (e.g. forward Euler, Runge-Kutta) can be used; rectified flows often tolerate relatively large step sizes due to their straighter velocity field.

**Efficiency and accuracy.** Because rectified flows explicitly encourage a "straighter" velocity field, accurate solutions often require fewer integration steps than their diffusion-based counterparts. This advantage can be crucial in computational fluid dynamics and other PDE applications where each solver step is expensive, as can be seen in Figure 2. See **SM** D.3 and **SM** G for a detailed comparison of step-count vs. accuracy trade-offs.

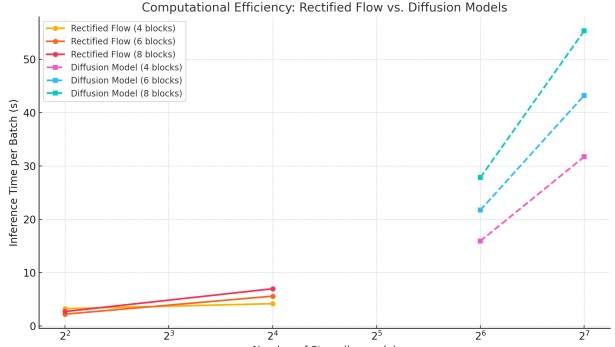

**Computational efficiency.** Diffusion surrogates (e.g. GenCFD) generate samples by integrating a reverse-time SDE whose trajectory bends through latent noise; even with accelerated samplers the procedure still requires tens of network calls. Rectified flows, in contrast, follow a single deterministic ODE with straighter paths and tolerate larger or adaptive steps.

Figure 2: Inference cost vs. model size. Rectified flows: 4–8 steps; GenCFD: 64–128.

## 5 EXPERIMENTS

### 5.1 EXPERIMENTAL SETUP

We benchmark our method, termed henceforth as *RecFlow*, against four (main) neural surrogate models. *GenCFD* and *Diffusion ∘ FNO* Oommen et al. (2024) are conditional diffusion surrogates following the framework introduced earlier in this section, whereas FNO and UViT serve as one-shot deterministic baselines. Detailed descriptions of all baselines are provided in **SM** C.

**Datasets for Training and Evaluation.** We consider three canonical 2D multi-scale PDE problems (see the Supplementary Material for precise mathematical formulations and solver details):

**SL2D** *Shear Layer in 2D*. This setup models the evolution of an incompressible shear layer, initially concentrated near a horizontal interface, which rolls up into intricate vortical structures over time. We generate approximate solutions by numerically solving the 2D Navier–Stokes system (with periodic boundaries) at a resolution of $512^2$, then downsample to $128^2$ for training and testing. The data has been generated with the spectral viscosity code Tadmor (1989); Rohner & Mishra (2024).

**CS2D** *Cloud-Shock Interaction in 2D*. In this popular compressible-flow test, a shock wave moves through a higher-density "cloud" region, triggering complex wave interactions and small-scale turbulent mixing in the wake. We solve the compressible Euler system at an effective resolution of $256^2$ using a GPU-optimized finite-volume ALSVINN code of Lye (2020); Fjordholm et al. (2020), then also downsample to $128^2$.

**RM2D** *Richtmeyer–Meshkov in 2D*. In this case, a shock wave interacts with a density interface separating two fluids, depositing vorticity and triggering vigorous mixing. We numerically solve the compressible Euler system on a $256^2$ grid, then downsample to $128^2$ to create the training and test data. The data has been generated with a high-resolution finite-volume ALSVINN code of Fjordholm et al. (2020)

**Training and test protocol.** We train RecFlow and all baselines on initial conditions $u_i \sim \mu$ (data construction details in **SM** B.2). For each $u_i$, a high-fidelity PDE solver (spectral or finite-volume) is run forward to produce snapshots that serve as supervision.

*Training objectives.* We use two complementary regimes: (i) **terminal pairs** $(u_i, u_f)$ with $u_f = \mathcal{S}(u_i)$ at a fixed physical time, where the model learns a rectified velocity conditioned on $(u_i, \tau)$ and is *only* exposed to pairs rather than the full conditional law $p(u(t) \mid u_i)$; (ii) **all-to-all (lag) pairs** $(u_t, u_{t+\delta t})$ drawn from the same forward trajectory, where the network additionally receives the physical lag $\delta t$ (not the absolute time) together with the diffusion time $\tau$.

*Test–time goal: Dirac initial measures.* We evaluate in the practically relevant setting where the initial condition is fixed,

$$\widehat{\mu}_i = \delta_{\widehat{u}_i},$$

so any stochasticity reflects chaotic, multi-scale evolution rather than variability in $u_i$. To obtain a reference ensemble for $p(u(t) \mid \widehat{u}_i)$, we use an *ensemble-perturbation* procedure (see **SM** B.2): small perturbations of $\widehat{u}_i$ are advanced with a high-resolution solver (e.g., Azeban for incompressible flows or a finite-volume code for compressible cases), yielding a ground-truth sample set.

*Distribution settings.* We report results in two regimes: (1) **matched**: $\widehat{u}_i$ is drawn from the same family $\mu$ used for training; although $\delta_{\widehat{u}_i}$ is technically out-of-distribution in the sense of being a Dirac, the underlying field belongs to the same flow class; (2) **distribution shift**: $\widehat{u}_i \sim \nu$ with $\nu \neq \mu$, probing generalization under shifts in the initial-condition distribution (details in **SM** B.2).

## 5.2 EVALUATION METRICS

We evaluate the generative performance of each method using three main metrics, with details provided in **SM** D:

- **Mean Error.** We measure the $L^2$ difference between the model's predicted mean flow $\mu$ and the ground-truth mean $\mu_{\text{exact}}$. A low mean error indicates accurate large-scale behavior.

- **Standard Deviation Error.** The normalized $L^2$ distance between the predicted standard deviation $\sigma$ and the reference $\sigma_{\text{exact}}$ captures whether the model reproduces correct spatial variability.

- **One-Point Wasserstein Distance.** We compute an Wasserstein-1 distance at each spatial point (and average over the domain) to quantify local distributional agreement between ground truth and model outputs.

Table 1: Comparison of models across datasets CS2D, SL2D, and RM2D, in terms of mean relative error ($e_\mu$), standard deviation relative error ($e_\sigma$), and Wasserstein-1 distance ($W_1$). Best-performing model is colored in Blue and second-best performing model in Orange.

| Model | Metric | CS2D | | | | SL2D | | RM2D | | | |
|---|---|---|---|---|---|---|---|---|---|---|---|
| | | $\rho$ | $m_x$ | $m_y$ | $E$ | $u_x$ | $u_y$ | $\rho$ | $m_x$ | $m_y$ | $E$ |
| RecFlow | $e_\mu$ | 0.0477 | 0.0332 | 0.041 | 0.015 | 0.034 | 0.189 | 0.018 | 0.032 | 0.020 | 0.021 |
| | $e_\sigma$ | 0.1005 | 0.072 | 0.078 | 0.0584 | 0.071 | 0.077 | 0.032 | 0.046 | 0.046 | 0.085 |
| | $W_1$ | 0.0091 | 0.0107 | 0.0124 | 0.0143 | 0.0214 | 0.0164 | 0.0033 | 0.0036 | 0.0028 | 0.0017 |
| GenCFD | $e_\mu$ | 0.0830 | 0.0619 | 0.127 | 0.0280 | 0.039 | 0.267 | 0.0049 | 0.0010 | 0.0025 | 0.0025 |
| | $e_\sigma$ | 0.195 | 0.197 | 0.244 | 0.169 | 0.092 | 0.072 | 0.0351 | 0.0254 | 0.0435 | 0.0435 |
| | $W_1$ | 0.0151 | 0.0184 | 0.0185 | 0.0247 | 0.0276 | 0.0213 | 0.0092 | 0.0016 | 0.0049 | 0.0049 |
| UViT | $e_\mu$ | 0.111 | 0.078 | 0.127 | 0.0479 | 0.233 | 0.623 | 0.070 | 0.070 | 0.071 | 0.052 |
| | $e_\sigma$ | 0.415 | 0.343 | 0.412 | 0.296 | 0.28 | 0.178 | 0.425 | 0.570 | 0.570 | 0.617 |
| | $W_1$ | 0.087 | 0.085 | 0.158 | 0.172 | 0.270 | 0.256 | 0.060 | 0.055 | 0.060 | 0.036 |
| FNO | $e_\mu$ | 0.0989 | 0.0748 | 0.0930 | 0.0383 | 0.058 | 0.408 | 0.085 | 0.070 | 0.063 | 0.067 |
| | $e_\sigma$ | 0.2573 | 0.2779 | 0.3188 | 0.2777 | 0.137 | 0.110 | 0.366 | 0.443 | 0.436 | 0.450 |
| | $W_1$ | 0.1455 | 0.1660 | 0.1879 | 0.1884 | 0.184 | 0.150 | 0.096 | 0.051 | 0.056 | 0.062 |
| Diffusion ∘ FNO | $e_\mu$ | 0.0638 | 0.0454 | 0.0603 | 0.0204 | 0.055 | 0.393 | 0.040 | 0.027 | 0.039 | 0.041 |
| | $e_\sigma$ | 0.1099 | 0.0969 | 0.1245 | 0.0974 | 0.131 | 0.100 | 0.091 | 0.075 | 0.091 | 0.090 |
| | $W_1$ | 0.0467 | 0.0534 | 0.0774 | 0.0743 | 0.166 | 0.135 | 0.0269 | 0.0158 | 0.0293 | 0.0273 |

**Baselines and Evaluation Settings.** To stay within reasonable training times for our infrastructure, we restrict the total number of training iterations up to 120,000 for each model and recorded the best achievable results. Under this budget, only a reduced-capacity GenCFD (with two downsampling stages in its U-Net backbone, versus three in Rectified Flow) can match the accuracy reported in Table 1. We also evaluated larger GenCFD variants trained for 500,000 iterations, but observed no meaningful improvement over the results shown here. Meanwhile, at inference time Rectified Flow requires only **8** denoising steps to achieve comparable or superior accuracy, whereas diffusion baselines (including GenCFD) need **128** steps, underscoring RF's efficiency in both training and sampling. FNO and UViT also require smaller models than RecFlow for optimal performance.

Since FNO, and UViT produce *deterministic* predictions, we approximate their spread by an *ensemble perturbation* strategy (**SM** B.2), where each baseline is fed slightly perturbed inputs and outputs are aggregated to form a sample set. In contrast, both the original GenCFD (baseline) and our *RecFlow* are inherently *probabilistic*, requiring no auxiliary procedure to generate diverse samples.

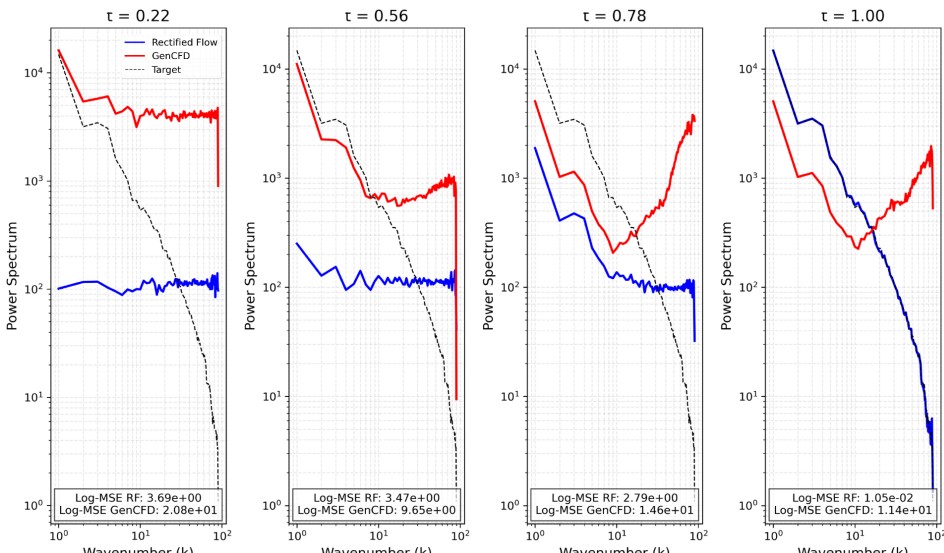

Figure 3: Energy-spectrum evolution for a random cloud–shock initial datum. RecFlow (blue) tracks high-wavenumber energy faster and more accurately than GenCFD (red).

## 5.3 RESULTS.

Table 1 compares five surrogates: our rectified-flow model (*RecFlow*) and four baselines, evaluated by mean error ($e_\mu$), spread error ($e_\sigma$), and Wasserstein-1 ($W_1$). On Cloud–Shock (CS2D) and Shear Layer (SL2D), *RecFlow* attains the best scores on all three metrics, capturing both mean behaviour and higher-order statistics. The hybrid *Diffusion ∘FNO* narrows the gap relative to *GenCFD* but still trails *RecFlow*. Deterministic one-shot baselines (FNO, UViT) deviate most, particularly in $e_\sigma$ and $W_1$, indicating difficulty modeling variability. Richtmeyer–Meshkov (RM2D) is the only strictly in-distribution test; in this easier setting the deterministic baselines come closest to diffusion-class methods, but only here. With identical training budgets and matched architecture/size, *RecFlow* yields lower errors than all baselines, including *GenCFD*, on CS2D and SL2D, indicating superior convergence efficiency.

**Relation to ODE-based diffusion samplers and standard flow matching.** To clarify the distinction, we also run GenCFD using the probability-flow ODE (PF-ODE) under matched step budgets and matched wall-clock. Following Karras et al. (2022), we used a second-order Heun solver and swept 4, 8, 16, 32, 64, and 128 steps. We also evaluated standard (non-rectified) flow matching under identical budgets. Neither PF-ODE nor standard flow matching improved distributional or spectral metrics: both induce stiffer fields than our rectified-flow (RF) velocity, so many steps ($> 8$) are still required to approach comparable accuracy. These additional approaches are deferred to the appendix, with implementation details in **SM** C, and results in **SM** D.1.

Moreover as visualized in the **SM** G, RecFlow generates samples of very high quality, matching the qualitative features of the ground truth and capturing features such as (interacting) vortices as well as propagating shock waves very accurately. Another aspect where RecFlow really shines in being able to generate the correct point pdfs. Although already indicated in the very low 1-point Wasserstein distance errors from Table 1, we present 1-point pdfs in the **SM** G to illustrate this observation.

Energy spectra are a sensitive diagnostic for multiscale fidelity Molinaro et al. (2024). In Fig. 13 we plot the log spectrum of the energy field for a random Cloud–Shock instance at four diffusion times, $\tau \in 0.25, 0.50, 0.75, 1.00$. The flow is strongly multiscale, exhibiting an approximate power–law decay. With only about five ODE steps, *RecFlow* tracks the ground-truth spectrum across high wavenumbers, while a score-based diffusion model (GenCFD) under-resolves small scales at the same step budget and requires many more evaluations to catch up. This aligns with Fig. 4, which compares density trajectories: *RecFlow* inpaints fine structure early in $\tau$ with few model calls, whereas GenCFD remains noticeably noisy at small step counts. Together these results illustrate

why *RecFlow* attains accurate statistics with far fewer solver evaluations. See **SM** D.3 for further visualizations of this effect.

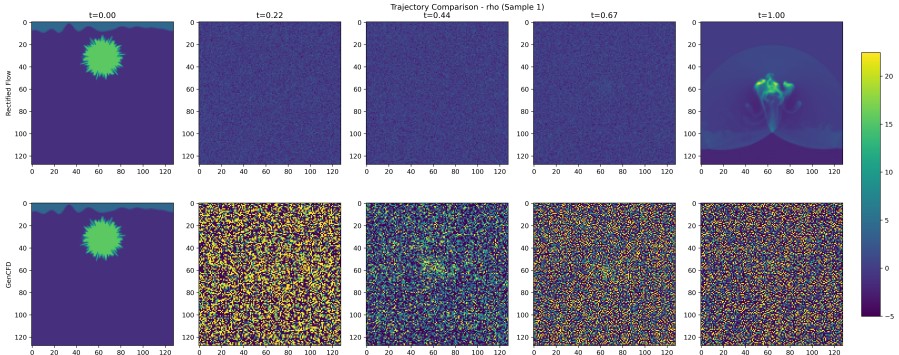

Figure 4: Density trajectories for Sample 1: RecFlow (top) versus GenCFD (bottom).

## 5.4 EMA–GUIDED CURVATURE CONTROL: RESULTS AND PRACTICE

We evaluate the curvature–aware integrator on two benchmarks and observe consistent gains over regular RF sampling. In both cases, the EMA deviation $s_t = \|v_t - v_t^{\text{ema}}\|_2$ serves as a reliable curvature signal: when curvature is modest, the sampler proceeds largely unmodified; when curvature grows, a gentle convex blend with the EMA trend (Eq. equation 5) plus adaptive step sizing stabilizes updates and yields the largest benefits.

Table 2: Curvature–aware integration summary. "Median straightness" is the median EMA straightness score $s_t$. "Improvement" is the relative error reduction over regular RF sampling (mean±std across macros). "Correction rate" is the fraction of (high-curvature) steps where blending engaged.

| Dataset | Median straightness $s_t$ | Avg. improvement | Correction rate |
|---|---|---|---|
| CloudShock2D | $\approx 0.028$ | $+2.6\% \pm 0.2\%$ | 13%–15% |
| ShearLayer2D | $\approx 0.026$ | $+6.0\% \pm 1.3\%$ | 12%–21% |

**Per–dataset details.** *CloudShock2D.* Average relative error reduction **2.6**% (std 0.2%) over regular RF sampling with modest curvature (median $s_t \approx 0.028$). Blending engages in 13%–15% of steps, and per–macro gains concentrate tightly in the 2.2%–3.0% range.

*ShearLayer2D.* Average relative error reduction **6.0**% (std 1.3%), with slightly higher curvature variability (median $s_t \approx 0.026$). Corrections occur in 12%–21% of steps; per–macro gains span 3.7%–8.1%, indicating that high–curvature regions benefit most from the regularized update.

## 6 CONCLUSION AND OUTLOOK

Rectified flows provide an efficient surrogate for sampling multi-scale PDE solutions. Across incompressible and compressible benchmarks, RecFlow matches diffusion-class fidelity with only **8–10** ODE steps (vs. $\geq 128$), yielding up to **22** $\times$ faster ( **SM** D, Fig. 2) inference while reproducing energy spectra, vorticity statistics, and pointwise Wasserstein-1 within sampling error under both matched and shifted initial conditions (**SM** G). A curvature-aware controller using an EMA straightness proxy with a light Tikhonov blend and adaptive step sizes (**SM** A.4)—keeps trajectories nearly straight (**SM** D.3) and permits larger, stabler steps; training is likewise efficient, reaching comparable accuracy in ∼120k iterations (vs. 500–600k for diffusion baselines) **SM** D.2.

**Limitations and future work.** We aim to scale RecFlow to three-dimensional turbulence and coupled multi-physics settings, investigate lightweight architectures for real-time deployment, and integrate the sampler into uncertainty-quantification pipelines for design and control. Continued benchmarking at higher resolutions and transfer across flow regimes will further validate robustness and practicality.

**Reproducibility Statement.** We document all ingredients needed to reproduce our results. Data generation, preprocessing, train/val/test splits, and the ensemble–perturbation protocol for Dirac initial measures are specified in the Supplementary Material (**SM**; App. B.2), including solver versions and random seeds. Model architectures, loss formulations, time embeddings, and all hyperparameters (optimizer, schedules, batch sizes, EMA decay) are given in Secs. 5 and **SM** (App. C); the curvature–aware sampler is fully specified in Sec. 5.4 with theory in App. A.4. Evaluation metrics (mean error, standard–deviation error, one–point $W_1$), the FFT pipeline for spectra, and exact sampling counts per figure/table are detailed in Sec. 5.2 and **SM**. We also attach an *anonymized minimal working version* of the code along with all the data as supplementary material, including scripts to regenerate Tables/Figures. Hardware and runtime footprints are reported in **SM** (App. E). These materials allow end-to-end replication of all quantitative results under the stated seeds and compute budgets.

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

# Supplementary Material for:

*Rectified Flows for Fast Multiscale Fluid Flow Modeling*

## Table of Contents

## A  ARCHITECTURE OF THE RECFLOW MODEL

### A.1  ARCHITECTURAL COMPONENTS

**Motivation.** Multi-scale PDE data contain wide-ranging spatial scales, so a *multi-resolution* encoder–decoder is natural. We incorporate time embeddings to condition each layer on the interpolation fraction $t$.

**Architecture Layout.** We employ a UNet-based architecture augmented with attentional layers, typically termed as **UViT** by Saharia et al. (2022). In addition to the MLP-based time embedding $\Gamma(\tau)$, the input signal is lifted into a higher-dimensional embedding space and later projected back into physical space through convolutional layers. The data flows from high to low resolution across

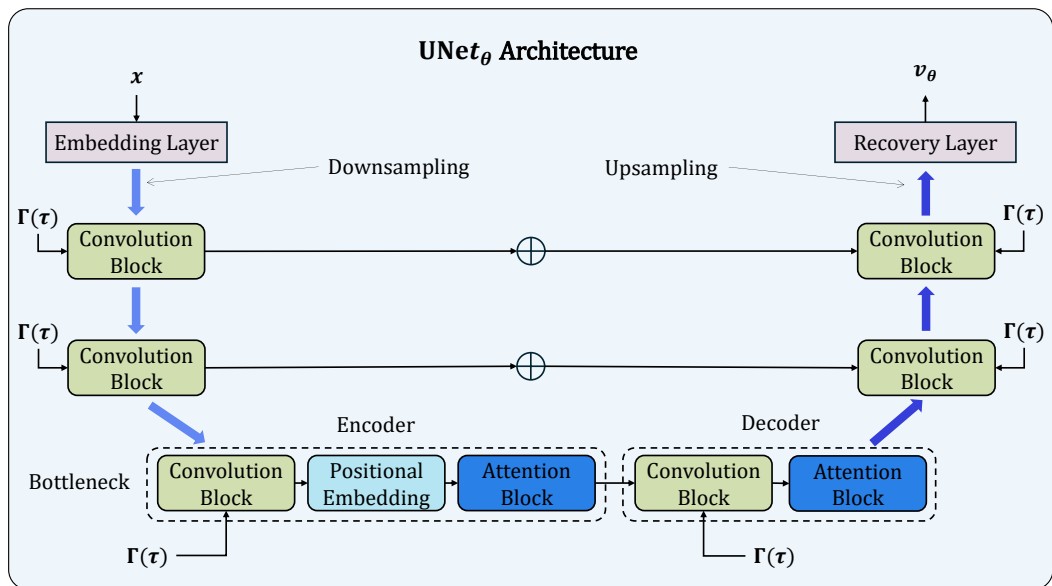

Figure 5: **UNet Backbone Architecture Used in RecFlow.** This schematic illustrates the core UNet-based architecture used within the RecFlow framework, structured across three resolution levels. For clarity, the number of blocks per level is set to one in this illustration. In the actual GenCFD and RecFlow configurations, each block is repeated four times per level. The bottleneck shows an asymmetry between the encoder and decoder sides: the block on the encoder side includes a Convolution Block, Positional Embedding, and Attention Block, while the corresponding block on the decoder side omits the Positional Embedding.

a down-sampling stack composed of convolution blocks into a bottleneck layer that encodes the fine-scale features of the inputs across its channel dimension and mixes them via a multi-head attention mechanism.

On the encoder side, exclusively at the bottleneck level, each attention block preceded by a simple linear positional embedding. This positional encoding helps preserve spatial locality before global mixing through attention. The decoder side does not apply any positional embedding.

The UNet is conditioned at every level through normalized Feature-wise Linear Modulation (FiLM) techniques embedded within the convolution blocks. The conditioning signal modulates the normalization layers via learned scale and shift parameters. This conditioning mechanism is consistently applied across all encoder and decoder stages.

The output is symmetrically reconstructed via an up-sampling pipeline. Downsampling is performed using standard convolutional layers with stride, while upsampling can be implemented either via transposed convolutions or by applying a standard convolution followed by a non-learnable pixel shuffle operation, which rearranges elements of the tensor spatially according to a fixed upsampling factor.

Importantly, the architecture described here resembles the one presented in Molinaro et al. (2024), ensuring consistency with prior state-of-the-art practices in high-fidelity generative modeling for scientific computing.

### A.1.1 CONVOLUTION BLOCK

Each UNet level employs dedicated convolution blocks that process features while enabling conditioning on the diffusion time $\tau$. These blocks consist of two convolutional layers interleaved with group normalization and Swish activations. Between the convolutions, a normalized FiLM layer adaptively scales features using the conditioning embeddings. A residual connection combines the processed features with a projected skip connection, ensuring stable gradient flow. Crucially, these convolution blocks form the fundamental building units across all encoder and decoder stages, not just the bottleneck, with consistent application of temporal conditioning through the FiLM mechanism.

### A.1.2 Normalized FiLM Layer

Each convolution block uses a normalized FiLM layer and applies a convolution layer interleaved with normalization and an activation, then adds a residual skip:

$$\mathbf{z} \; = \; \text{Conv}\big(\text{Norm}(\mathbf{x})\big), \qquad \mathbf{z} \; = \; \sigma(\mathbf{z} + \alpha\,\gamma(t)),$$
$$\mathbf{x}_{\text{out}} \; = \; \mathbf{x} \; + \; \text{Conv}\big(\mathbf{z}\big), \tag{8}$$

where Norm is often `RMSNorm`, GroupNorm or LayerNorm, $\sigma$ is a nonlinearity (e.g. SiLU), and $\gamma(t)$ represents the time embedding (an MLP on $\text{Sinusoid}(t)$) that modulates scale/shift. This temporal conditioning is crucial for guiding the generative process through rectified diffusion trajectories.

### A.1.3 Attention Block

The architecture strategically employs multi-head attention exclusively at the bottleneck layer to balance global interaction modeling with computational efficiency. Each attention block processes normalized features through a spatial-channel reformatting, where features are reshaped by a flattening operation. This preserves the channel structure while collapsing the spatial grid into a single flattened dimension, allowing attention to operate across spatial locations. At certain scales, a multi-head attention (MHA) mechanism captures distant flow interactions. We define

$$\mathbf{Q} \; = \; W_Q\,\mathbf{h}, \quad \mathbf{K} \; = \; W_K\,\mathbf{h}, \quad \mathbf{V} \; = \; W_V\,\mathbf{h}, \quad \mathcal{A}(\mathbf{h}) \; = \; \text{softmax}\Big(\tfrac{\mathbf{QK}^\top}{\sqrt{d}}\Big)\,\mathbf{V}, \tag{9}$$

where $\mathbf{h} \in \mathbb{R}^{(H\cdot W)\times d}$ denotes the flattened input sequence, with each of the $H \cdot W$ spatial positions represented as a $d$-dimensional feature vector. In practice, attention is split into multiple heads for better representational power, then recombined. This global mixing is key for PDE flows dominated by far-field couplings (e.g. vortex merging). The MHA operator is defined as

$$\text{MHA}(\mathbf{h}) \; = \; W^O\,\text{Concat}\left(\mathcal{A}_1(\mathbf{h}), \dots, \mathcal{A}_n(\mathbf{h})\right), \tag{10}$$

where $n \in \mathbb{N}$ denotes the number of attention heads and $W^O \in \mathbb{R}^{C\times C}$ is a learned output projection matrix. The attention block applies MHA to a normalized input, then adds a residual connection to preserve the original signal and stabilize training while enabling global context to influence local features:

$$\mathbf{z} \; = \; \text{MHA}\big(\text{Norm}(\mathbf{x})\big),$$
$$\mathbf{x}_{\text{out}} \; = \; \mathbf{x} \; + \; \mathbf{z}. \tag{11}$$

### A.2 Training Pipeline Details

The training pipeline follows the algorithm described in the main text faithfully.

**Trainer Workflow.** A high-level `Trainer` class manages:

1. Batching PDE pairs $(u_0, u_1)$. These normalized pixel-wise with statistics extracted per each entire dataset.

2. Sampling $\tau \in [0, 1]$ and random noise $\xi$ according to a given noise scheduler $\sigma(\cdot)$.

3. Partial interpolation + noise addition: $\tilde{u}_\tau := \tau\,u_1 + (1-\tau)\,\xi$ (or the flow-based variant). We have experimented with non-linear interpolation methods, but have not noticed meaningful oscillations in performance.

4. Minimizing the squared deviation between model output and target displacement/noise, optionally with a consistency/EMA update.

5. Throughout training, we monitor validation performance on a separate holdout dataset across all main metrics. We explicitly note that different validation splits have been tested out to ensure there is no data leakage that would bias performance metrics.

Table 3: Representative hyperparameters for 2D fluid-flow tasks. See main text for justification of these choices.

| Parameter | Value (Optimal empirical performance) | Notes |
|---|---|---|
| Channels per level | (128, 256, 256) | Number of channels per level. |
| Downsample Ratios | (2, 2, 2) | Downsampling ratio per level. |
| Attention Heads | 8 | Each attention block. |
| Dropout Rate | 0.1 | In ConvBlock for generalization. |
| Noise Schedule | `uniform`, `log-uniform`, `cos-map` | Helps with multi-scale noise. |
| ODE Steps (Sampling) | 4–8 | Often $> 15\times$ fewer vs. diffusion. |
| Batch Size | 16 | Tied to GPU capacity. |
| EMA Decay | 0.9999 | Teacher model for consistency. |
| Learning Rate | $3 \times 10^{-4}$ | Cosine decay in code. |

In practice we noticed that the most robust denoising schedule was just the `uniform` one. For details on the `cos-map` schedule, we direct the reader to its description in Esser et al. (2024).

### A.3 SUMMARY AND KEY TAKEAWAYS

The `RectifiedFlow` class, combined with a time-aware `UNet2D` architecture, builds a deterministic transport model well-suited to multi-scale PDE tasks. By interpolating between noise and data along nearly-straight paths, it reduces sampling overhead compared to conventional diffusion PDE solvers while retaining the flexibility to model complex turbulent phenomena. Further experimental and theoretical results appear in **SM** Section D. Instead of the multi-step *stochastic* integration in diffusion-based models, `RectifiedFlow` fits a *deterministic* ODE from noise to data. This leverages straighter trajectories in frequency space, enabling fewer network evaluations at inference time.

**ODE Integration for Sampling.**  At inference:

$$\frac{d\tilde{u}_\tau}{d\tau} \;=\; v_\theta\big(\tilde{u}_\tau,\, u_0,\, \tau\big), \quad \tau \in [0,1], \quad \tilde{u}_{\tau=0} \;=\; \text{random noise.}$$

We solve this with `odeint`. Few steps $(8-10)$ typically suffice, contrasting with the tens or hundreds required by full diffusion samplers.

### A.4 CURVATURE–AWARE RECTIFIED FLOWS: THEORETICAL CONSIDERATIONS

**Setup.**  Let the *ideal* transport be the nonautonomous ODE

$$\dot{u}_\tau \;=\; v_\star(u_\tau, \tau), \qquad u_0 \sim \mu_0, \quad \tau \in [0,1], \tag{12}$$

and the *learned* flow be

$$\dot{\hat{u}}_\tau \;=\; v_\theta(\hat{u}_\tau, \tau), \qquad \hat{u}_0 = u_0, \tag{13}$$

driven by a neural velocity $v_\theta$. We assume: (i) $v_\theta(\cdot, \tau)$ is $L$-Lipschitz in $u$, uniformly in $\tau$; (ii) $v_\star \in L^2([0,1] \times \mathcal{U})$; (iii) on the region traversed by the trajectories, the vector field $v$ under consideration is $C^1$ with $\|\partial_\tau v(u, \tau)\| \le M_\tau$, $\|J_u v(u, \tau)\| \le L$, $\|v(u, \tau)\| \le M$. Sampling uses explicit Euler with (possibly variable) step sizes $\{\Delta\tau_k\}_k$ and a curvature–aware *blend*

$$\tilde{v}_\tau \;=\; (1 - \alpha_\tau)\, v_\tau \;+\; \alpha_\tau\, v_\tau^{\text{ema}}, \qquad v_\tau := v_\theta(u_\tau, \tau), \quad v_\tau^{\text{ema}} := \lambda\, v_{\tau-\Delta\tau}^{\text{ema}} + (1 - \lambda)\, v_\tau, \tag{14}$$

where $\lambda \in (0,1)$ and $\alpha_\tau \in [0,1]$ increases smoothly with the curvature proxy $s_\tau := \|v_\tau - v_\tau^{\text{ema}}\|_2$. (Initialization can be taken as $v_0^{\text{ema}} = v_0$.) When we analyze numerical sampling with a field $w$ (either $w = v_\theta$ or $w = \tilde{v}$), we let $L$ denote a uniform Lipschitz constant for $w(\cdot, \tau)$.

**A Grönwall–type terminal error split.**  We separate modeling error into a *reconstruction* part and a *straightness* (time–variation) part.

**Theorem 2** (Terminal error decomposition). *Let $U_\tau$ solve $\dot{U}_\tau = v_\star(U_\tau, \tau)$ with $U_0 = u_0$, and let $\hat{u}_\tau$ solve $\dot{\hat{u}}_\tau = v_\theta(\hat{u}_\tau, \tau)$ with $\hat{u}_0 = u_0$. Define the time–averaged velocity $\bar{v}_\theta(u) := \int_0^1 v_\theta(u, \tau)\, d\tau$ and*

$$\varepsilon_{\mathrm{fit}}^2 := \int_0^1 \mathbb{E}\big\|\bar{v}_\theta(U_\tau) - v_\star(U_\tau, \tau)\big\|^2 d\tau, \qquad \varepsilon_{\mathrm{curv}}^2 := \int_0^1 \mathbb{E}\big\|v_\theta(U_\tau, \tau) - \bar{v}_\theta(U_\tau)\big\|^2 d\tau.$$

*Then, for the continuous–time flows,*

$$\mathbb{E}\big\|\hat{u}_1 - u_1\big\| \leq e^L \Big(\varepsilon_{\mathrm{fit}} + \varepsilon_{\mathrm{curv}}\Big). \tag{15}$$

*Proof.* Write $v_\theta - v_\star = (\bar{v}_\theta - v_\star) + (v_\theta - \bar{v}_\theta)$. By triangle inequality and Cauchy–Schwarz in $\tau \in [0, 1]$, $\int_0^1 \mathbb{E}\|v_\theta(U_\tau, \tau) - v_\star(U_\tau, \tau)\|\, d\tau \leq \varepsilon_{\mathrm{fit}} + \varepsilon_{\mathrm{curv}}$. With $e_\tau := \hat{u}_\tau - u_\tau$,

$$\dot{e}_\tau = v_\theta(\hat{u}_\tau, \tau) - v_\star(u_\tau, \tau) = \big(v_\theta(\hat{u}_\tau, \tau) - v_\theta(u_\tau, \tau)\big) + \delta_\tau,$$

$\| v_\theta(\hat{u}_\tau, \tau) - v_\theta(u_\tau, \tau) \| \leq L\|e_\tau\|$, $\delta_\tau := v_\theta(u_\tau, \tau) - v_\star(u_\tau, \tau)$. Thus $\frac{d}{d\tau}\|e_\tau\| \leq L\|e_\tau\| + \|\delta_\tau\|$, so $\|e_1\| \leq e^L \int_0^1 \|\delta_\tau\|\, d\tau$. Taking expectations completes the proof. $\qquad \square$

**Interpretation.** The terminal error decomposes into (i) a bias term $\varepsilon_{\mathrm{fit}}$ measuring how well the time–average $\bar{v}_\theta$ matches $v_\star$, and (ii) a straightness term $\varepsilon_{\mathrm{curv}}$ measuring the within–time variability of $v_\theta$ around its own mean. The curvature–aware blend pulls $v_\theta$ toward the low–pass proxy $v^{\mathrm{ema}}$, directly shrinking $\varepsilon_{\mathrm{curv}}$.

**Tikhonov optimality of the blend.** At fixed $(u_\tau, \tau)$, the blended velocity uniquely minimizes

$$\tilde{v}_\tau = \arg\min_w \Big\{ \|w - v_\tau\|_2^2 + \beta_\tau \|w - v_\tau^{\mathrm{ema}}\|_2^2 \Big\}, \qquad \alpha_\tau = \frac{\beta_\tau}{1 + \beta_\tau} \in [0, 1), \tag{16}$$

so $\tilde{v}_\tau = (1 - \alpha_\tau)v_\tau + \alpha_\tau v_\tau^{\mathrm{ema}}$. If $\alpha_\tau = \varphi(s_\tau)$ with $\varphi$ bounded and Lipschitz, then $\tilde{v}(\cdot, \tau)$ remains Lipschitz in $u$ with a constant comparable to $L$ (convex combination plus a mild dependence of $\alpha_\tau$ on $u$ through $s_\tau$).

**Local truncation error (LTE) and a curvature–aware step rule.** We record the LTE for explicit Euler on $\dot{u} = v(u, \tau)$, where $v$ is the field being integrated (either $v_\theta$ or $\tilde{v}$).

**Lemma 2** (LTE for explicit Euler). *If $v \in C^1$ with $\|\partial_\tau v(u, \tau)\| \leq M_\tau$, $\|J_u v(u, \tau)\| \leq L$, $\|v(u, \tau)\| \leq M$ along the trajectory, then one step of size $\Delta\tau$ from $(u, \tau)$ has*

$$\mathrm{LTE}(\tau; \Delta\tau) = \frac{(\Delta\tau)^2}{2} \big\|\partial_\tau v(u, \tau) + J_u v(u, \tau)\, v(u, \tau)\big\| + \mathcal{O}((\Delta\tau)^3) \leq \frac{(\Delta\tau)^2}{2}(M_\tau + LM) + \mathcal{O}((\Delta\tau)^3). \tag{17}$$

*Proof.* Taylor expansion gives $u(\tau + \Delta\tau) = u + \Delta\tau\, v(u, \tau) + \frac{(\Delta\tau)^2}{2}\ddot{u}(\tau) + \mathcal{O}((\Delta\tau)^3)$. Differentiating $\dot{u} = v(u, \tau)$ yields $\ddot{u} = \partial_\tau v(u, \tau) + J_u v(u, \tau)\, v(u, \tau)$. Subtract the Euler update and bound by the sup norms. $\qquad \square$

To enforce a per–step tolerance $\mathrm{LTE} \lesssim \mathrm{tol}$, set

$$\Delta\tau_\tau = \mathrm{clip}\left(\sqrt{\frac{2\,\mathrm{tol}}{\kappa_1\, \widehat{M}_{\tau,\tau} + \kappa_2}},\ \Delta\tau_{\min},\ \Delta\tau_{\max}\right), \qquad \kappa_2 \approx L\, \|v_\tau\|, \tag{18}$$

where $\mathrm{clip}(x, a, b) := \min\{b, \max\{a, x\}\}$. Here $\widehat{M}_{\tau,\tau}$ is a proxy for $\|\partial_\tau v\|$. The EMA deviation $s_\tau = \|v_\tau - v_\tau^{\mathrm{ema}}\|$ provides an *upper* bound proxy (Lemma 3 below), so a safe choice is

$$\widehat{M}_{\tau,\tau} = \gamma\, \frac{1 - \lambda}{\lambda}\, \frac{s_\tau}{\Delta\tau_\tau}, \qquad \gamma \geq 1, \tag{19}$$

where $\gamma$ is a safety factor (larger $\gamma$ increases conservatism). Under blending, $\|\tilde{v}_\tau - v_\tau^{\mathrm{ema}}\| = (1 - \alpha_\tau)s_\tau$, so both the curvature proxy and the step size shrink in tandem.

**Discrete curvature proxy bound.** When $v(\cdot, \tau)$ is smooth in $\tau$ and steps have size $\Delta\tau$, the EMA deviation controls the time–variation:

**Lemma 3** (EMA deviation vs. time derivative). *Let* $v_\tau := v(u_\tau, \tau)$ *and* $v_\tau^{\mathrm{ema}} = (1 - \lambda) \sum_{j=0}^\infty \lambda^j v_{\tau-j\Delta\tau}$. *If* $\|\partial_\tau v(u, \tau)\| \leq M_\tau$ *along the path, then*

$$\|v_\tau - v_\tau^{\mathrm{ema}}\| \leq \frac{\lambda}{1-\lambda} M_\tau \Delta\tau. \tag{20}$$

*Proof sketch.* Telescoping $v_\tau - v_{\tau-j\Delta\tau} = \sum_{m=0}^{j-1}(v_{\tau-m\Delta\tau} - v_{\tau-(m+1)\Delta\tau})$, each increment is bounded by $\int_{\tau-(m+1)\Delta\tau}^{\tau-m\Delta\tau} \|\partial_\tau v\| \, d\tau \leq M_\tau\Delta\tau$. Weight by $(1-\lambda)\lambda^j$ and use $\sum_{j\geq 0} j\lambda^j = \lambda/(1-\lambda)^2$. $\qquad\square$

**Global discretization error under adaptive steps.** Let $\mathrm{LTE}_k$ be the (unnormalized) one–step error at step $k$, and suppose $\mathrm{LTE}_k \leq \mathrm{tol}$ for all $k$.

**Proposition 1** (Global error with tolerance–controlled steps). *Under the Lipschitz assumption on the integrated field $w$ with constant $L$, the global discretization error at $\tau = 1$ satisfies*

$$\|u_1^{\mathrm{num}} - \hat{u}_1\| \leq e^L \sum_{k=0}^{N-1} \mathrm{LTE}_k \leq e^L N \, \mathrm{tol}, \tag{21}$$

*where $N$ is the number of steps. If the step rule equation 18 is used with $\widehat{M}_{\tau,\tau} \approx M_\tau$, then*

$$N \approx \int_0^1 \frac{d\tau}{\Delta\tau(\tau)} = \frac{1}{\sqrt{2\,\mathrm{tol}}} \int_0^1 \sqrt{M_\tau + \kappa_2} \, d\tau, \quad \Rightarrow \quad \|u_1^{\mathrm{num}} - \hat{u}_1\| \lesssim e^L \sqrt{\frac{\mathrm{tol}}{2}} \int_0^1 \sqrt{M_\tau + \kappa_2} \, d\tau. \tag{22}$$

*Proof sketch.* For one–step methods with Lipschitz right–hand sides, $\|e_{k+1}\| \leq (1 + L\Delta\tau_k)\|e_k\| + \mathrm{LTE}_k$. Iterate and bound $\prod_k(1 + L\Delta\tau_k) \leq \exp(L \sum_k \Delta\tau_k) = e^L$. The estimate for $N$ follows from $N \approx \int d\tau/\Delta\tau(\tau)$ under the chosen rule. $\qquad\square$

**Remarks and takeaways.** (i) If sampling uses $\tilde{v}$, interpret $L, M_\tau, \kappa_2$ for $\tilde{v}$, and Theorem 2 still applies to the continuous $v_\theta$ vs. $v_\star$ comparison; a further perturbation term $\int_0^1 \|\tilde{v}(u, \tau) - v_\theta(u, \tau)\| \, d\tau$ can be added if one wishes to relate $\tilde{v}$– and $v_\theta$–flows directly.
(ii) The blend is the Tikhonov–optimal contraction toward a low–pass trend, shrinking the straightness component in Theorem 2.
(iii) The EMA deviation furnishes a computable proxy for $\|\partial_\tau v\|$ (Lemma 3), enabling a principled step rule that controls LTE (Lemma 2).
(iv) Together, blending and adaptive stepping yield a global discretization error that scales like $\sqrt{\mathrm{tol}}$ under the tolerance–controlled schedule (Proposition 1), while preserving large steps on straight segments and stabilizing precisely where curvature spikes.

# B  DATASETS

## B.1  MULTI-SCALE FLOW DATASETS

Following the dataset generation procedure of Herde et al. Herde et al. (2024), the Richtmyer–Meshkov ensemble is created by imposing a randomized sinusoidal perturbation on a two-fluid interface and driving it with a planar shock via prescribed pressure jumps. For the CloudShock and ShearFlow benchmarks, we follow the GenCFD mesh-generation and solver setup of Molinaro et al. Molinaro et al. (2024), initializing CloudShock with concentric density perturbations and ShearFlow with orthogonal shear jets on adaptive Cartesian grids, and employing the same high-order finite-volume scheme and dissipation settings to ensure consistent numerical fidelity across both datasets.

**Overview.** We experiment on three paired-field benchmarks $(u_0, u_1)$:

- **Richtmyer–Meshkov (RM)**: 64 000 samples

- **Cloud-Shock (CS)**: 40 000 samples

- **Shear-Layer (SL)**: 79 200 samples

All datasets track the $u_0 \to S^t(u_0)|_{t=1}$ solution mapping from inputs to the evolved outputs at $t = 1$. The first two datasets monitor the evolution of density ($\rho$), momentum components ($m_x, m_y$), and energy respectively ($E$). The shear-flow dataset monitors the velocity components of the flow $(u_x, u_y)$.

**Train/Validation Split.** For each dataset, we reserve 80% of the samples for training and 20% for validation.

**Macro–Micro Ensemble Evaluation (CS & SL).** To probe out-of-distribution generalization under small perturbations, we adopt a two-stage ensemble protocol:

1. *Macro-sampling:* select $M_{\mathrm{macro}} = 10$ distinct base initial conditions.

2. *Micro-perturbation:* for each base, generate $M_{\mathrm{micro}} = 1,000$ perturbed copies within a small radius.

3. *Metrics:* compute all performance measures (mean-field error $e_\mu$, standard-deviation error $e_\sigma$, average $W_1$, spectral agreement) by first averaging over each micro-ensemble, then averaging across the $M_{\mathrm{macro}}$ cases.

**In-Distribution Evaluation (RM).** For the Richtmyer–Meshkov dataset, we additionally evaluate on a held-out test set drawn from the same distribution (no macro–micro perturbations). This allows us to study how deterministic and probabilistic models scale with training set size under standard, in-distribution conditions.

### B.2 Preprocessing and Data Organization

#### Data Processing

Before training, we compute and store global, channel-wise statistics over the entire training dataset. Concretely, for each physical variable $c$ we calculate

$$
\mu_c = \frac{1}{N_{\mathrm{train}}\, H\, W\, [D]} \sum_{i=1}^{N_{\mathrm{train}}} \sum_{x,y,[z]} u_c^{(i)}\big(x, y, [z], t_0\big),
$$

$$
\sigma_c = \sqrt{\frac{1}{N_{\mathrm{train}}\, H\, W\, [D]} \sum_{i=1}^{N_{\mathrm{train}}} \sum_{x,y,[z]} \Big(u_c^{(i)}\big(x, y, [z], t_0\big) - \mu_c\Big)^2}.
$$

where $N_{\mathrm{train}}$ is the number of training samples, $H \times W\, [\times D]$ the spatial grid size, and $u_c(x, y, [z], t_0)$ the initial field for channel $c$. We average over all spatial locations to yield a single mean $\mu_c$ and standard deviation $\sigma_c$ per channel. The same statistics are also computed over the final-time fields $u_c(t_f)$, producing $\mu'_c$ and $\sigma'_c$ for the output channels.

During both training and evaluation, each input field $u_c$ is standardized via

$$
\tilde{u}_c = \frac{u_c - \mu_c}{\sigma_c + 10^{-16}}, \qquad \tilde{v}_c = \frac{v_c - \mu'_c}{\sigma'_c + 10^{-16}},
$$

where $u_c$ and $v_c$ denote the initial and target fields, respectively. We then concatenate $\{\tilde{u}_c\}$ and $\{\tilde{v}_c\}$ along the channel axis before feeding them to the network. Because we always apply the same pre-computed statistics at test time, no information from the evaluation set ever influences these normalization parameters.

**Test-time ensemble perturbation (evaluation only).** We adopt the same testing setup proposed in *GenCFD* Molinaro et al. (2024). To evaluate how well our model captures the *distribution* of solutions $u(t)$ arising from a fixed, chaotic initial field $\bar{u}$, we generate a small perturbed ensemble around $\bar{u}$ and evolve each member with a high-fidelity PDE solver—*but only at evaluation time*. Specifically:

1. **Micro-ensemble generation.** Around the test initial condition $\bar{u}$, draw $M_{\mathrm{micro}}$ perturbed copies $\{\bar{u}_j\}_{j=1}^{M_{\mathrm{micro}}}$ by sampling uniformly in a ball of radius $\varepsilon$ centered at $\bar{u}$.

2. **Reference propagation.** Integrate each perturbed field $\bar{u}_j$ forward to time $t$ using the reference solver, producing end-states $\{u_j(t)\}$.

3. **Empirical conditional law.** Form the empirical measure

$$\hat{P}_t(\,\cdot\mid\bar{u})\;=\;\frac{1}{M_{\mathrm{micro}}}\sum_{j=1}^{M_{\mathrm{micro}}}\delta_{u_j(t)},$$

which approximates the true chaotic conditional law $P_t(\,\cdot\mid\bar{u})$.

When the input distribution itself is non-degenerate, one would first draw $M_{\mathrm{macro}}$ base fields from that distribution and then apply the micro-ensemble procedure to each. For our Dirac initial-condition tests we set $M_{\mathrm{macro}}=1$.

**Training vs. evaluation.** During training the model only ever sees independent pairs $(u_0, u_1)$—one target per input—and never observes any ensemble. The above ensemble perturbation is used *only* to construct a ground-truth distribution at test time.

## C    MODELS AND BASELINES

We tested our method against 6 other baselines, out of which 4 are diffusion-based algorithms of the same kind. The other two baselines rely on traditional operator learning setup, with the UViT baseline satisfying also the purpose of an ablation study that quantifies the gain obtained by adding diffusion on top.

**Shared Backbone**. We stress that for GenCFD, Diffusion $\circ$ FNO and UViT we use the exact same UViT backbone architecture outlined in the earlier parameterization of our rectified flow as in Table 3 for the sake of consistency. The architecture is further illustrated schematically in Figure 5, which shows the UNet-based design in detail.

### C.1    GENCFD

GenCFD Molinaro et al. (2024) is an end-to-end conditional score-based diffusion model that directly learns the mapping from an input flow field $u_0$ to the target field $u_1$. It uses the same UViT backbone described in Table 3, with noise levels $\sigma_\tau$ corresponding to the standard deviation of the added Gaussian perturbation, in contrast to our RecFlow model which evolves over a rectified diffusion time $\tau \in [0, 1]$. Common choices for noise schedulers include exponential and tangent formulations. In our case, we consistently adopted an exponential noise schedule, as it yielded the best results across all datasets and metrics reported. For score-based diffusion models, either variance-exploding (VE) or variance-preserving (VP) formulations are typically used. In all GenCFD models, we opted for the VE setting throughout. During training we minimize the denoising score-matching loss over $(u_0, u_1)$ pairs sampled uniformly in $\tau \in [0, 1]$, using Adam with initial learning rate $3 \times 10^{-4}$ (cosine decay), batch size 16, and EMA decay 0.9999. As is common in score-based diffusion models, we applied denoiser preconditioning and used uniform weighting across noise levels. At inference we solve the corresponding SDE (via an Euler-Maruyama scheme with discretization 128 steps) and incorporate reconstruction guidance to condition on $u_0$ so that we recover the conditional posterior measure $p(u_1|u_0)$.

## C.2 Diffusion ○ FNO

In this hybrid approach, a Fourier Neural Operator (FNO) Li et al. (2020a) is first trained under an $\ell_2$ loss to predict $u_1$ from $u_0$. Its output $\hat{u}_1^{\text{FNO}}$ is concatenated channel-wise with $u_0$ and fed into the GenCFD score network at both train and test time. All other architectural and optimization settings (backbone, scheduler, SDE sampler) remain identical to GenCFD, allowing the FNO to provide fast low-frequency priors while the diffusion stage refines fine-scale turbulent features.

**Custom implementation.** This schematic closely parallels the pipeline of Oommen et al. (2024), but with two key distinctions. First, we fully replace both the diffusion backbone and its denoising stages with GenCFD. Second, and most important, we condition not only on the FNO's coarse solution but also on the original high-resolution initial condition. This ensures that our model retains fine-scale structures that would otherwise be lost if we relied solely on the FNO predictions.

## C.3 PF-ODE (PROBABILITY–FLOW ODE)

For completeness, we also evaluate the ODE sampler obtained by replacing the GenCFD SDE with its probability–flow ODE. We use the same trained UViT score network and the same exponential (VE) noise schedule as above. At inference, we integrate the PF-ODE with a second-order Heun solver and sweep 4, 8, 16, 32, 64, and 128 steps under matched wall-clock and step budgets, using the same reconstruction guidance to condition on $u_0$ and recover $p(u_1 \mid u_0)$. Despite this tuning (following the setup popularized by Karras et al., 2022), PF-ODE does not improve distributional or spectral metrics: the induced field is stiffer than our rectified-flow velocity, so many steps ($> 8$) are still required.

## C.4 STANDARD (NON-RECTIFIED) FLOW MATCHING

We further include a conventional flow-matching baseline. Using the same UViT backbone, we train a time-dependent velocity $v_\theta(x, t, u_0)$ to match the derivative of a reference conditional path between $u_0$ and $u_1$. Concretely, we sample $t \sim \mathcal{U}(0, 1)$ and $\varepsilon \sim \mathcal{N}(0, I)$, form

$$x_t = (1 - t)\,u_0 + t\,u_1 + \sigma(t)\,\varepsilon, \qquad \dot{x}_t = (u_1 - u_0) + \dot{\sigma}(t)\,\varepsilon,$$

with an exponential schedule $\sigma(t)$, and minimize $\mathbb{E}\|v_\theta(x_t, t, u_0) - \dot{x}_t\|^2$ using Adam (lr $3 \times 10^{-4}$ with cosine decay), batch size 16, and EMA 0.9999. At test time we integrate $\dot{x} = v_\theta(x, t, u_0)$ with a Heun solver under the same step budgets as PF-ODE. This non-rectified FM underperforms our rectified-flow sampler on distributional and spectral metrics and similarly requires many steps to approach comparable accuracy.

## C.5 FNO

Our FNO baseline follows Li *et al.* Li et al. (2020a) and is implemented in PyTorch. First, each spatial location of the input field $u_0$ is lifted from $C_{\text{in}}$ to 256 channels via a two-layer MLP with a SiLU activation, then projected down to a 128-channel hidden representation. This is followed by four `FnoResBlock`s: in each block, the hidden features undergo a Conditional LayerNorm (time embedding size 128), SiLU, and two spectral convolutions (`SpectralConv`) with $num\_modes = (24, 24)$, combined through a learnable "soft-gate" residual skip. Finally, a two-layer projection MLP ($128 \rightarrow 256 \rightarrow C_{\text{out}}$, with SiLU in between) produces the output field $\hat{u}_1$. We train for 100k iterations minimizing $\|u_1 - \hat{u}_1\|_2^2$ with Adam (learning rate $10^{-3}$, batch size 16), and perform inference in one forward pass—no diffusion or autoregression is used.

## C.6 UViT

Our UViT surrogate Saharia et al. (2022) uses the identical UNet2D backbone of Table 3, injecting time embeddings and employing 8-head multi-head attention at the bottleneck. It is trained deterministically with MSE loss on $(u_0, u_1)$, using Adam at $3 \times 10^{-4}$, batch size 16, for 100 k iterations (EMA 0.9999). For multi-step forecasting we roll out autoregressively, feeding each prediction back as the next input. This ablates out the diffusion component while matching all other design choices.

Table 4 summarizes key architectural settings for all baseline models considered in this work. These configurations were selected to reflect the strongest performance achievable within a shared training budget of up to 120,000 iterations, as used in the quantitative comparison reported in Table 1.

**Padding and boundary conditions.** Each dataset required padding choices tailored to its physical boundary conditions. SL and RM, which exhibit periodic boundaries, use circular padding in all convolutional layers. This significantly improved the performance of the diffusion-based models. In contrast, CS features non-periodic boundaries and is thus trained with standard zero padding. These padding choices apply to all UNet-based architectures except RecFlow, which uniformly uses zero padding across all datasets regardless of boundary conditions.

Table 4: Summary of architectural depth and approximated parameter count for each model evaluated. All UNet-based models (RecFlow, GenCFD, Diffusion ∘ FNO, and UViT) share a consistent backbone design to ensure consistency across diffusion and non-diffusion baselines, while the FNO baseline uses a structurally different architecture. These configurations reflect the best-performing settings under a training budget of up to 120,000 iterations, as used in the statistical comparisons reported in Table 1.

| Model | # Levels | # Blocks | # Params | Noise Schedule | Diffusion Scheme |
|-------|----------|----------|----------|----------------|------------------|
| RecFlow | 3 | 4 | 22M | uniform | rectified continuous time variable $t \in [0, 1]$ |
| GenCFD | 2 | 4 | 5M | exponential | variance-exploding |
| Diffusion ∘ FNO | 2 | 4 | 5M | exponential | variance-exploding |
| UViT | 2 | 4 | 5M | — | — |
| FNO | — | 4 | 11M | — | — |

# D RESULTS

## D.1 PERFORMANCE METRICS

To compare our generated ensembles against the reference Monte Carlo samples, we employ three complementary measures:

- **Mean-field error.** Let $\mu_{\text{ref}}(x)$ and $\mu_{\text{model}}(x)$ denote the pointwise spatial means of the reference and model ensembles, respectively. We measure their discrepancy by the $L^2$-norm

$$e_\mu = \big\| \mu_{\text{ref}} - \mu_{\text{model}} \big\|_2 .$$

- **Standard-deviation error.** Similarly, let $\sigma_{\text{ref}}(x)$ and $\sigma_{\text{model}}(x)$ be the pointwise standard deviations. We therefore compute

$$e_\sigma = \big\| \sigma_{\text{ref}} - \sigma_{\text{model}} \big\|_2 ,$$

Both metrics are reported after normalization by the $L^2$ norm of the ground truth.

- **Average 1-Wasserstein distance.** For a fixed initial condition $u_0$, let $\{u_{\text{ref}}^{(j)}(x)\}_{j=1}^N$ and $\{u_{\text{model}}^{(j)}(x)\}_{j=1}^N$ be the ensembles of final-time values produced by the reference solver and our model, respectively. Since these fields are discretized on $M$ spatial points $\{x_i\}_{i=1}^M$, we obtain two empirical 1D distributions at each $x_i$. Denoting their inverse CDFs by $F_{\text{ref}}^{-1}(\cdot\,;x_i)$ and $F_{\text{model}}^{-1}(\cdot\,;x_i)$, the pointwise Wasserstein-1 distance is

$$W_1\big(p_{\text{ref}}(\cdot \mid x_i),\, p_{\text{model}}(\cdot \mid x_i)\big) = \int_0^1 \big| F_{\text{ref}}^{-1}(q;x_i) - F_{\text{model}}^{-1}(q;x_i)\big| \,\mathrm{d}q.$$

We then report the spatial average

$$\overline{W}_1 \;=\; \frac{1}{M} \sum_{i=1}^{M} W_1\big(p_{\mathrm{ref}}(\cdot \mid x_i),\, p_{\mathrm{model}}(\cdot \mid x_i)\big).$$

**Energy spectra.** Finally, to assess how well different scales are captured, we compute the discrete energy spectrum of each velocity field. Given a sample $u(x)$ on a $d$-dimensional periodic grid with spacing $\Delta$, let $\widehat{u}_k$ be its Fourier coefficient at integer wavevector $k \in \mathbb{Z}^d$. We bin modes by their $\ell_1$ radius $|k|_1 = k_1 + \cdots + k_d$, defining

$$E_r \;=\; \frac{\Delta^d}{2} \sum_{\substack{k \in \mathbb{Z}^d \\ |k|_1 = r}} \big\|\widehat{u}_k\big\|^2, \quad r = 0, 1, 2, \dots$$

and compare the ensemble-average spectra of reference and model. Our plots show $E_r$ versus $r$ on log–log axes to highlight scale-by-scale agreement.

ADDITIONAL FLOW-BASED BASELINES

**PF–ODE control for diffusion baselines   Protocol.** We re–evaluated GenCFD by replacing its SDE sampler with the probability–flow ODE (PF–ODE), reusing the trained UViT score network and the same VE exponential noise schedule and reconstruction guidance as in the main text. Following Karras et al. (2022; arXiv:2206.00364), we integrated the PF–ODE with a second-order Heun solver and swept 4, 8, 16, 32, 64, and 128 steps under matched wall-clock and step budgets.

**Result.** PF–ODE reduces error somewhat faster at very low step counts but does not surpass the SDE sampler at comparable or higher budgets; the induced deterministic field remains comparatively stiff and still requires $> 8$ steps to preserve spectra. Representative 128-step metrics on *Shear-Layer* (defined in the main text) are reported below; the full sweep is summarized in SM D.

Table 5: PF–ODE vs. SDE GenCFD on *Shear-Layer* (128 steps). Metrics $e_\mu$, $e_\sigma$, and $W_1$ are as defined in the main text; lower is better.

| Channel $W_1$ | Sampler | Steps | $e_\mu$ | $e_\sigma$ |
|---|---|---|---|---|
| $v_x$ 0.127 | ODE | 128 | 0.0445 | 0.0937 |
| $v_x$ 0.0276 | SDE | 128 | 0.0390 | 0.0920 |
| $v_y$ 0.103 | ODE | 128 | 0.2890 | 0.0743 |
| $v_y$ 0.0213 | SDE | 128 | 0.2670 | 0.0720 |

**Takeaway.**   In our setting PF–ODE converges faster initially but yields worse final distributional and spectral metrics than the SDE-based GenCFD sampler at matched budgets.

**Standard (non-rectified) flow matching   Protocol.** We trained a conventional flow-matching model with the same UViT backbone and an exponential noise schedule, minimizing the standard velocity-matching objective. At test time we integrated the learned ODE with a Heun solver; the table reports 200-step results on *Shear-Layer*.

**Takeaway.**   The non-rectified flow-matching baseline underperforms both ODE and SDE GenCFD samplers on all reported metrics in this regime. For reference, a standard DDPM baseline performed worse than flow matching and is omitted for brevity.

Table 6: Standard flow matching on *Shear-Layer* (200 steps). Lower is better.

| Channel | $e_\mu$ | $e_\sigma$ | $W_1$ |
|---------|---------|------------|-------|
| $v_x$ | 0.0611 | 0.119 | 0.165 |
| $v_y$ | 0.448 | 0.104 | 0.144 |

## D.2 TRAINING EFFICIENCY OBSERVATIONS

To assess training efficiency, we compare RecFlow with GenCFD. For fairness, we use a GenCFD variant with the same setup and parameter count (22M), matching the three-level UViT architecture of RecFlow instead of its standard lightweight configuration (5M). Both models were trained for up to 160,000 iterations with a batch size of 16.

Figure 6 summarizes performance over the training trajectory. At approximately 100,000 steps, RecFlow consistently outperforms GenCFD across all key metrics: relative $L^2$ error in the mean ($e_\mu$), relative $L^2$ error in the standard deviation ($e_\sigma$), and the average pointwise Wasserstein-1 (W1) distance—evaluated for both $u_x$ and $u_y$ velocity components. Notably, the $y$-axis uses a logarithmic scale to emphasize differences in error magnitude across training steps.

Across all metrics, RecFlow begins to outperform GenCFD at around 100,000 training steps and continues to improve at a sharper rate. While neither model reaches full convergence within 160,000 iterations, RecFlow consistently exhibits lower errors and more favorable metric trajectories in later stages of training. These results support the observation that RecFlow is more training-efficient within a fixed compute budget and ultimately achieves stronger generalization performance in this setting. To further probe the long-term capabilities of GenCFD, we extended its training to 600,000 iterations and checked whether it eventually surpasses RecFlow's performance at its final recorded checkpoint (150,000 steps). Table 7 summarizes the earliest training steps at which GenCFD overtakes RecFlow for each evaluation metric. While GenCFD does catch up and exceed RecFlow in several cases, it never surpasses RecFlow on the Wasserstein-1 distance for $u_y$ within the training horizon considered.

Table 7: Number of training iterations at which GenCFD surpasses RecFlow's performance at 150,000 iterations. If not surpassed, indicated by "—".

| Metric | GenCFD Iteration | GenCFD Value | RecFlow Value @ 150,000 iter. |
|--------|------------------|--------------|-------------------------------|
| $u_x$ Mean Error ($e_\mu$) | 280,000 | $3.33 \times 10^{-2}$ | $3.45 \times 10^{-2}$ |
| $u_y$ Mean Error ($e_\mu$) | 400,000 | $1.70 \times 10^{-1}$ | $1.88 \times 10^{-1}$ |
| $u_x$ Std Error ($e_\sigma$) | 400,000 | $6.48 \times 10^{-2}$ | $7.24 \times 10^{-2}$ |
| $u_y$ Std Error ($e_\sigma$) | 340,000 | $7.91 \times 10^{-2}$ | $7.97 \times 10^{-2}$ |
| $u_x$ Wasserstein-1 ($W_1$) | 200,000 | $3.89 \times 10^{-2}$ | $3.95 \times 10^{-2}$ |
| $u_y$ Wasserstein-1 ($W_1$) | — | — | $4.11 \times 10^{-2}$ |

## D.3 EXPERIMENTAL ADVANTAGES OF STRAIGHTNESS

This subsection presents empirical evidence that the "rectified" trajectories generated by our Rectified Flow model (RecFlow) enable faster convergence and yield higher-fidelity reconstructions compared to the baseline GenCFD diffusion approach.

**Evaluation settings.** We use the CloudShock dataset throughout for exposition purposes. However, we stress that the same results hold across datasets. For each initial condition, both RecFlow and GenCFD are allotted $T = 10$ diffusion steps over normalized **diffusion time** $\tau \in [0, 1]$. We record the model outputs at five evenly spaced timesteps $t = \{0.0, 0.25, 0.50, 0.75, 1.0\}$ and analyze:

- *Trajectory inpainting* of the density field $\rho$.

- *Latent-Space Trajectory Visualization via PCA* via PCA.

- *Per-Sample Average Error Evolution*: average MSE across physical channels.

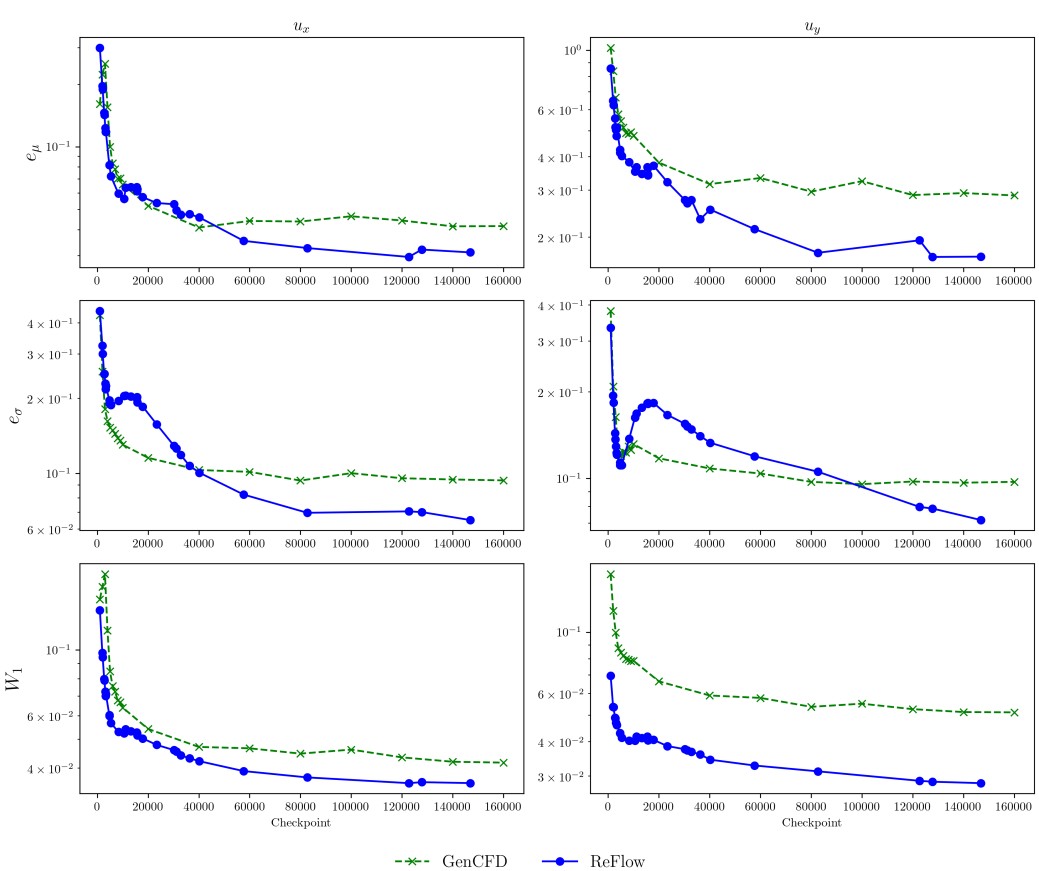

Figure 6: **Training efficiency comparison.** Performance of RecFlow and GenCFD on the Macro-Micro SL ensemble dataset evaluated at multiple training checkpoints. Metrics match those used in the main text: relative $L^2$ error of the mean ($e_\mu$) and standard deviation ($e_\sigma$), and the average 1-point Wasserstein-1 distance (W1), computed for both velocity components ($u_x$, $u_y$). The $y$-axis uses a logarithmic scale, while the $x$-axis (training iteration) is linear.

- *Evolution of the Energy Spectrum*: radial power-spectrum of energy $E$ as inpaintig progresses.

All other components (backbone, noise schedule, batch size) are held equal for fair comparison.

TRAJECTORY INPAINTING

We compare Rectified Flow (RecFlow) and GenCFD reconstructions of the density field $\rho$ over five normalized **diffusion** timesteps $t \in \{0.00, 0.25, 0.50, 0.75, 1.00\}$. RecFlow inpaints fine-scale features in far fewer steps, while GenCFD remains overly diffusive. The leftmost panel corresponds to the initial condition $u_0$ upon which both model are conditioned, while the following panels showcase the evolution of the model's output as diffusion time progresses.

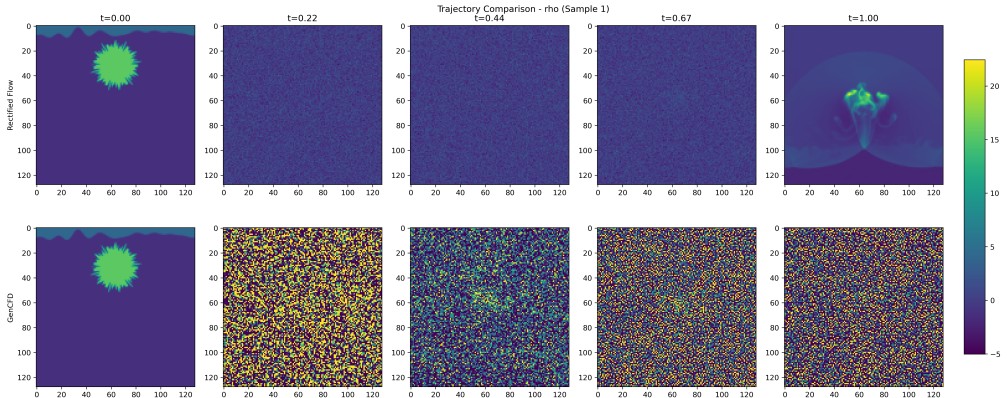

Figure 7: Density trajectories for Sample 1: RecFlow (top) vs. GenCFD (bottom). RecFlow already recovers sharp shock fronts by diffusion time $t = 1.0$.

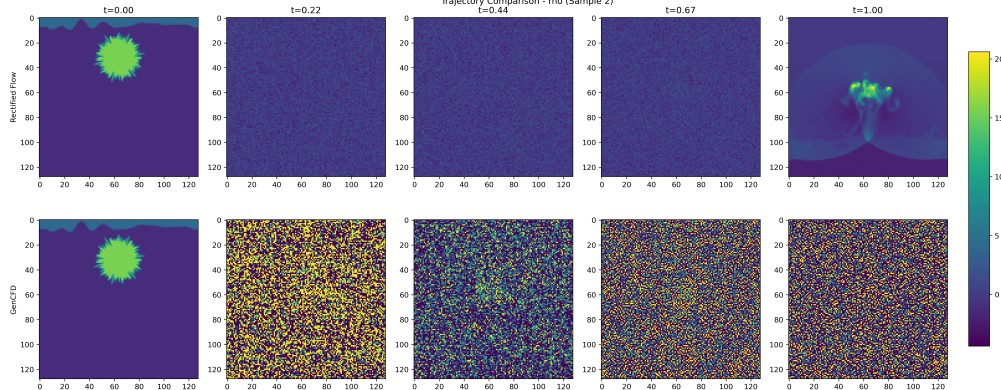

Figure 8: Density trajectories for Sample 2.

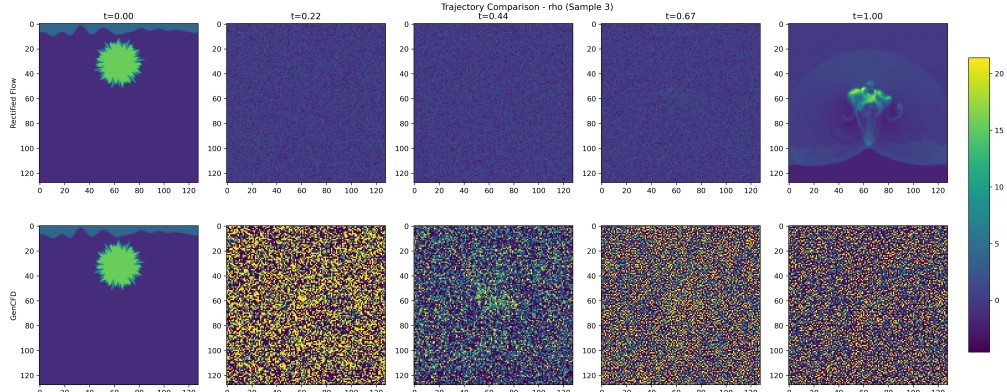

Figure 9: Density trajectories for Sample 3.

LATENT-SPACE TRAJECTORY VISUALIZATION VIA PCA

To quantify how directly each model moves through the data manifold, we flatten the $(H \times W \times C)$ field at each timestep into a single feature vector, stack all Rectified Flow and GenCFD vectors, fit a 3-component PCA, and then project each timestep into PC1/PC2/PC3. A straighter path indicates fewer diffusive detours.

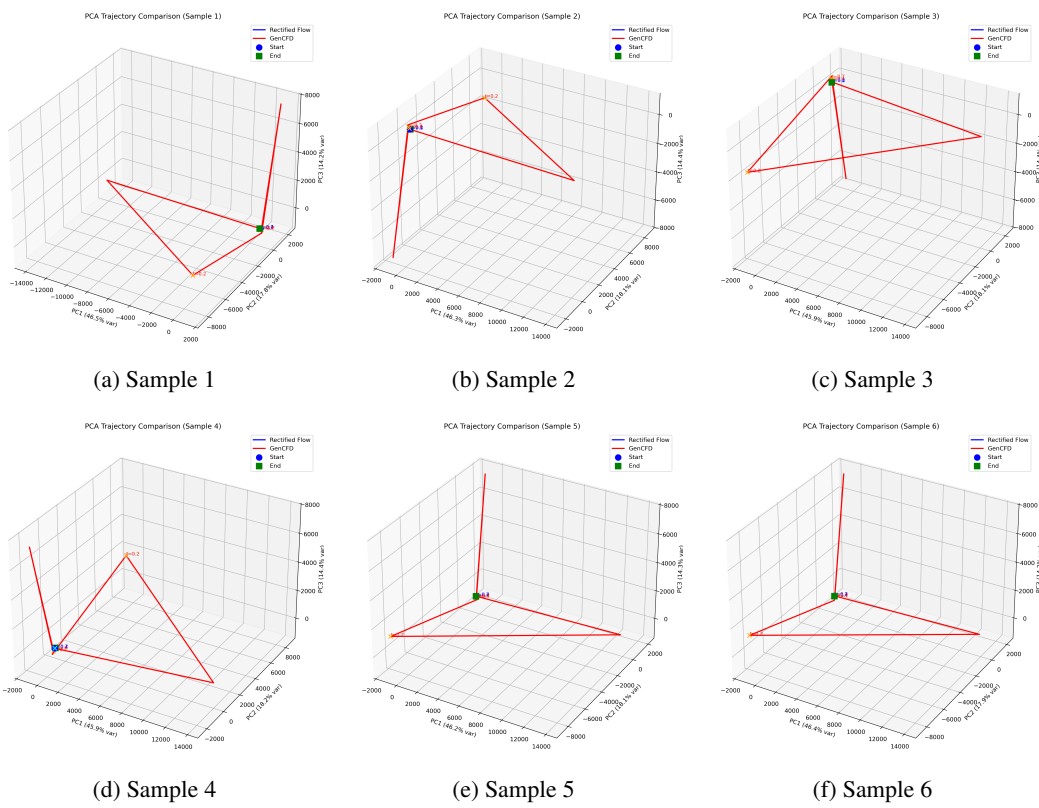

(a) Sample 1  (b) Sample 2  (c) Sample 3

(d) Sample 4  (e) Sample 5  (f) Sample 6

Figure 10: **3D PCA latent-space trajectories (Samples 1–6).** We project the flattened snapshots at each of $T$ timesteps into the top three PCA components. Rectified Flow (blue) consistently traces a straighter path than GenCFD (red), confirming its "rectified" evolution. Circles mark $t = 0$, squares mark $t = 1$, and intermediate timesteps are annotated.

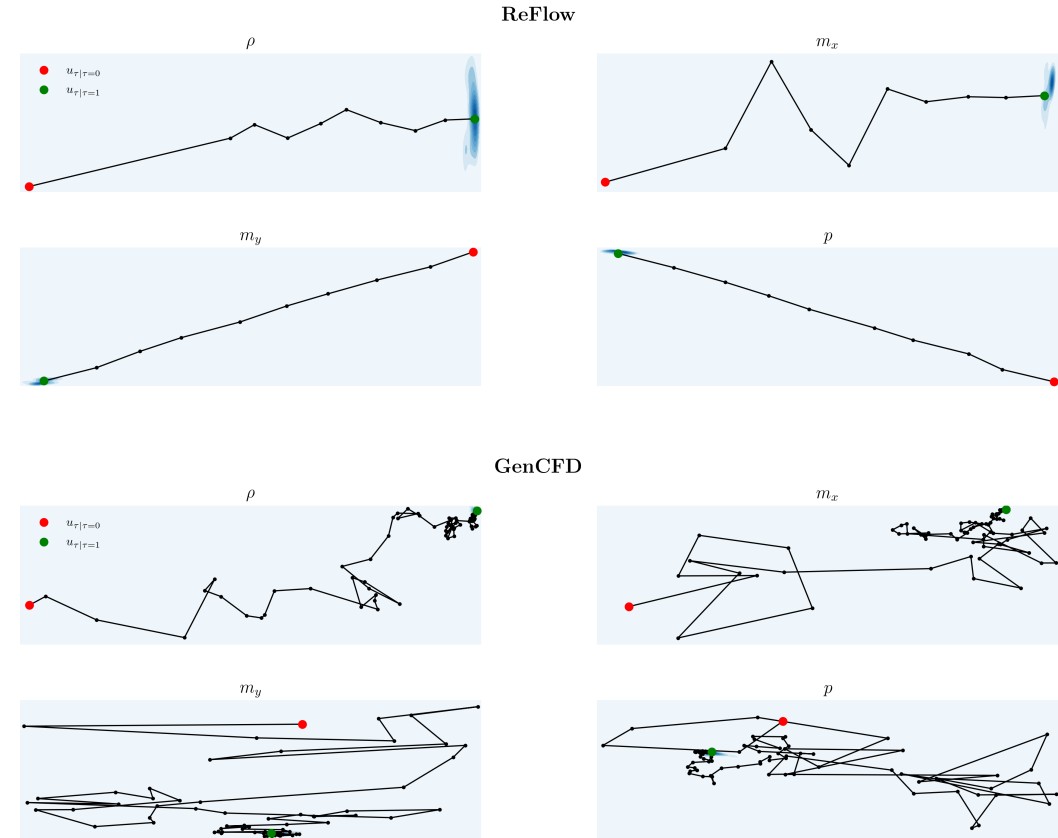

Figure 11: **2D PCA latent-space trajectories.** We project the flattened velocity snapshots from the RM dataset into a 2D PCA space, computed independently for each channel ($\rho$, $m_x$, $m_y$, $p$) across all $T$ time steps. To visualize the underlying distribution of target fields, we apply kernel density estimation (KDE) per channel. The RecFlow model (top) evolves over 10 rectified time steps, producing visibly straighter and more structured trajectories compared to the score-based diffusion model GenCFD (bottom), which uses 128 stochastic steps. Although the projection into two principal components necessarily truncates the high-dimensional dynamics, the qualitative difference remains evident, illustrating the smoother and more directed progression learned by RecFlow.

To further illustrate the nature of each model's generative dynamics, we visualize single-sample generation trajectories from both models in a 2D PCA space for the RM dataset (Figure 11). For each physical channel ($\rho$, $m_x$, $m_y$, $p$), we apply PCA to the full set of ground truth snapshots to define a common projection basis. We then overlay the generative trajectory of a single sample in this space. To contextualize the generated dynamics, we add kernel density estimates (KDEs) showing the empirical data distribution in the same projected space.

RecFlow (top) again traces a straighter, more compact path over just 10 rectified steps, moving coherently along the data manifold. GenCFD (bottom), by contrast, uses 128 stochastic denoising steps and produces a more diffuse and nonlinear trajectory, often deviating from high-density regions of the ground truth distribution. While the PCA projection inherently reduces dimensional complexity, the consistent difference in shape and alignment supports the conclusion that RecFlow's rectified flow better preserves manifold structure and sample efficiency.

PER-SAMPLE AVERAGE ERROR EVOLUTION

We compute the mean-squared error at each timestep $t$ and average across all $C$ channels:

$$\overline{\mathrm{MSE}}(t) = \frac{1}{C} \sum_{c=1}^{C} \mathrm{MSE}_c(t).$$

Figure 12 shows, for Samples 1–6, the log-scaled average MSE (left) and the normalized error $\overline{\mathrm{MSE}}(t)/\overline{\mathrm{MSE}}(0)$ (right). Rectified Flow (blue) consistently reduces error faster and to a lower residual than GenCFD (red).

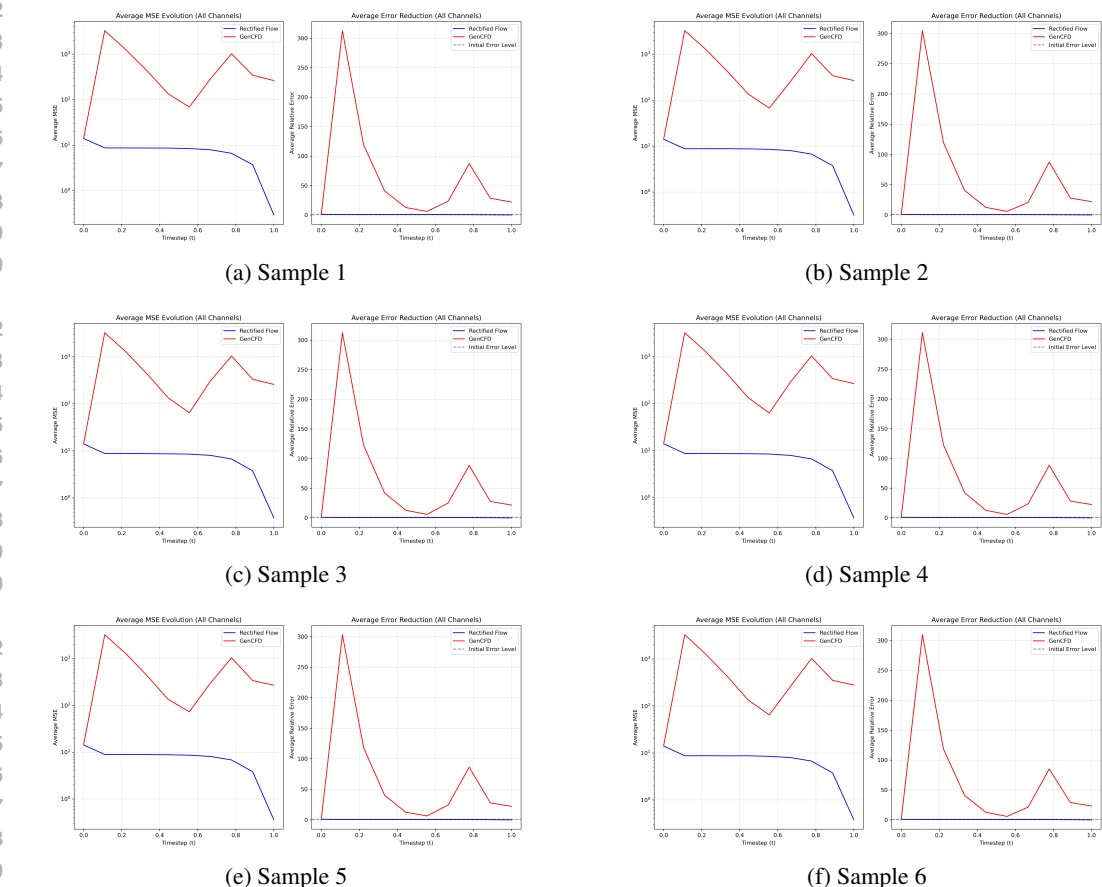

(a) Sample 1            (b) Sample 2

(c) Sample 3            (d) Sample 4

(e) Sample 5            (f) Sample 6

Figure 12: **Per-sample average MSE evolution.** Each subfigure plots, left half: $\log_{10} \overline{\mathrm{MSE}}(t)$; right half: $\overline{\mathrm{MSE}}(t)/\overline{\mathrm{MSE}}(0)$. Rectified Flow (blue) rapidly drives error down and keeps it orders of magnitude below GenCFD (red) across all samples.

EVOLUTION OF THE ENERGY SPECTRUM

We examine the 2D radial power spectrum of the energy field $E$ at five normalized diffusion times $\tau \in \{0.00, 0.25, 0.50, 0.75, 1.00\}$. Each figure below shows log–log power vs. wavenumber for Rectified Flow (blue), GenCFD (red), and the ground-truth target (black dashed), with inset log-MSE error annotations. The leftmost snapshot corresponds to the spectrum of the initial data $u_0$ upon which the model is conditioned. Subsequent panels track the spectral evolution of the model predictions, as diffusion time progresses.

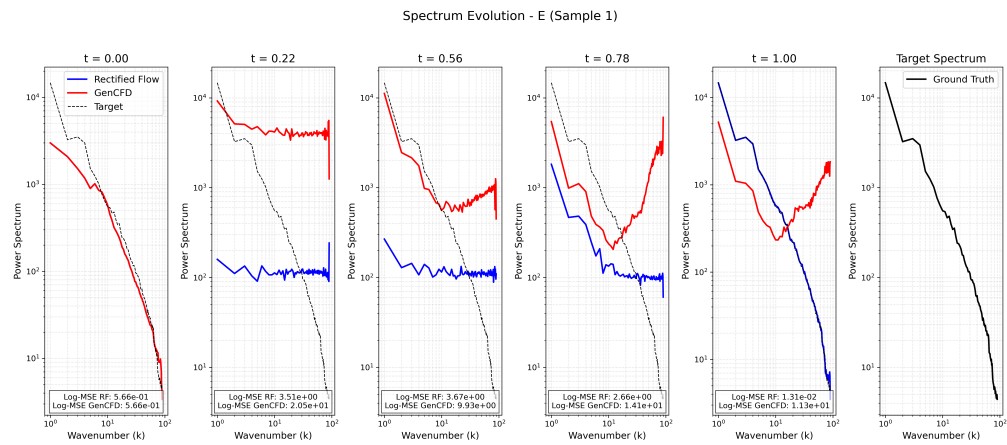

Figure 13: Energy spectrum evolution for Sample 1. RecFlow (blue) outperforms GenCFD (red) in capturing high-wavenumber energy.

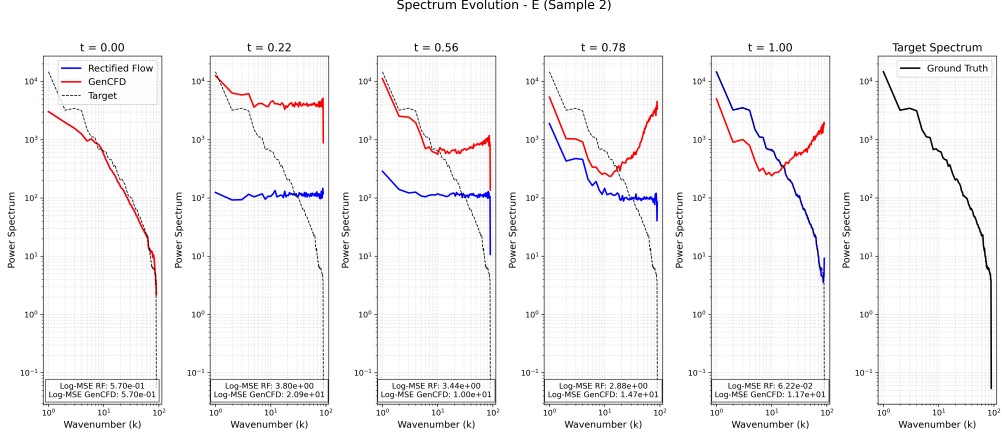

Figure 14: Energy spectrum evolution for Sample 2. RecFlow (blue) remains closer to the ground truth (dashed) than GenCFD (red).

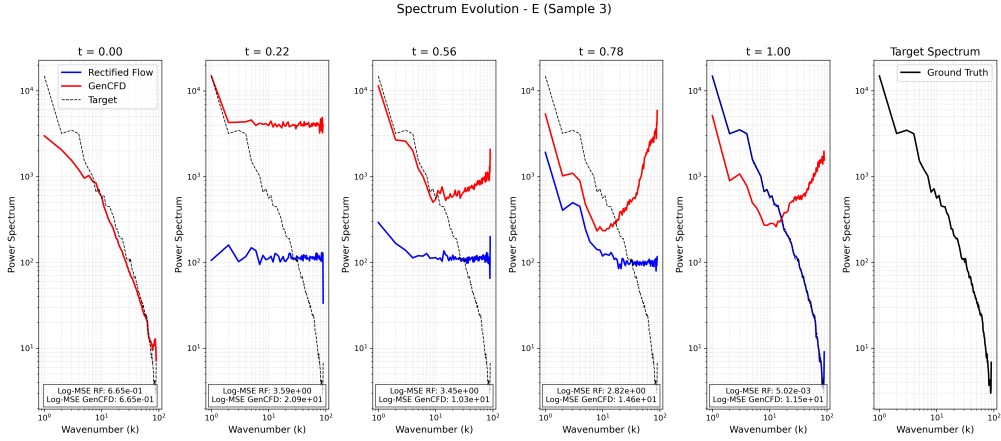

Figure 15: Energy spectrum evolution for Sample 3. The inset log-MSE annotations quantify RecFlow's advantage at each $t$.

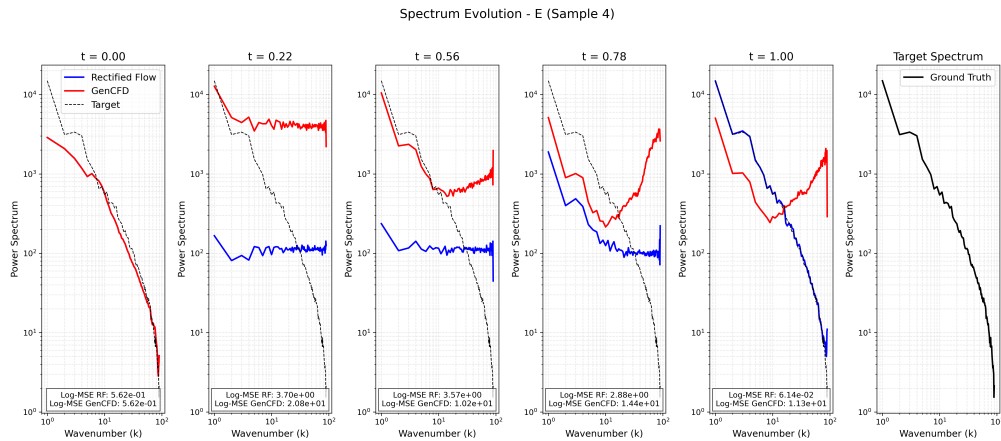

Figure 16: Energy spectrum evolution for Sample 4. RecFlow's spectrum (blue) gets closer to the target (dashed) as $\tau \to 1.0$.

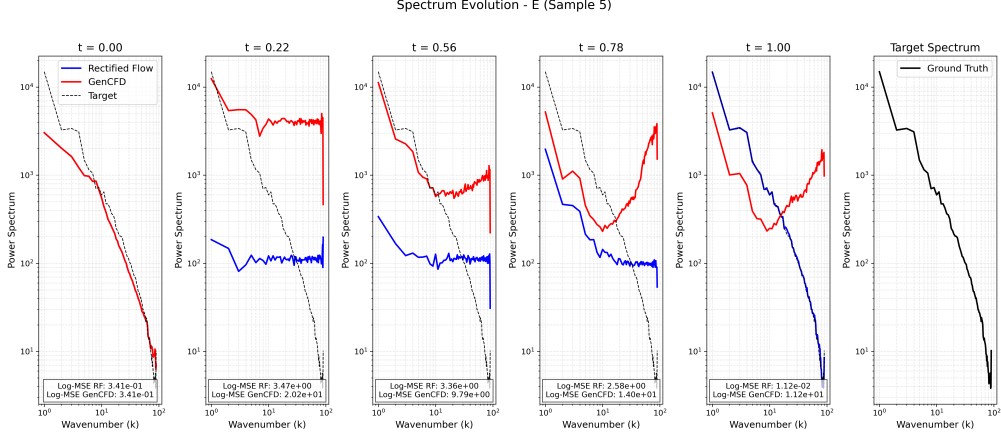

Figure 17: Energy spectrum evolution for Sample 5. RecFlow's superior high-$k$ fidelity is evident throughout.

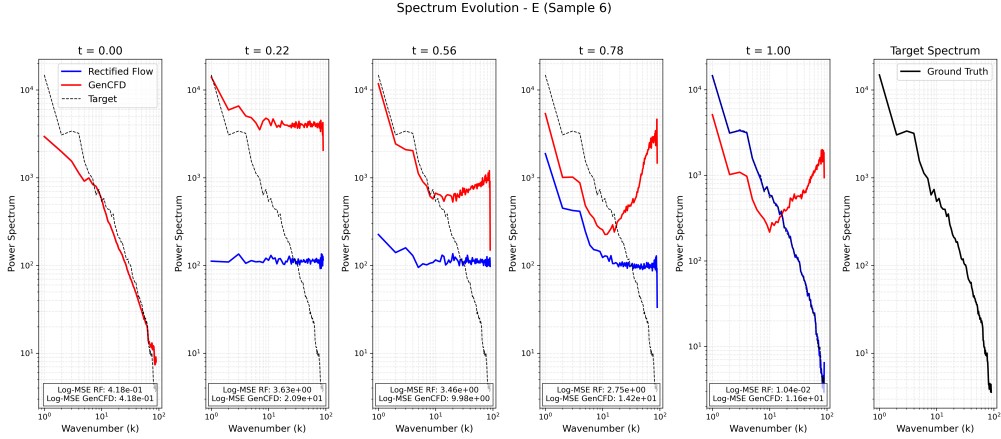

Figure 18: Energy spectrum evolution for Sample 6. RecFlow better captures the ground-truth spectrum's slope across all timesteps.

# E  COMPUTATIONAL CONSIDERATIONS

All of our RecFlow models for 2D fluid-flow tasks use the same network configuration summarized in Table 3 (channels $(128, 256, 256)$, downsample ratios $(2, 2, 2)$, 8 attention heads, dropout 0.1, batch size 16, etc.). Consequently, every model—regardless of dataset—has the same total parameter count (approximately 8–10M parameters) and identical layer-wise dimensions. We ran each training job for 100 000 iterations on a single NVIDIA Quadro RTX 6000 (Driver 570.124.06, CUDA 12.8) in persistence mode. Resource utilization averaged as follows over the full 100 h window:

Table 8: Aggregate hardware utilization for all "usual-setup" runs.

| Metric | Range | Typical |
|---|---|---|
| GPU power draw (W) | 200–260 | 230 |
| GPU utilization (%) | 30–80 | 55 |
| GPU memory allocated (GB) | 12–17 | 15 |
| GPU temperature (°C) | 40–60 | 50 |
| Process GPU memory usage (%) | 60–80 | 70 |
| Process GPU memory access time (%) | 10–50 | 30 |
| Process CPU threads in use | 7 | 7 |
| System CPU utilization (per core, %) | 60–100 | 80 |
| Process memory in use (GB) | up to 50 | 25 |
| Disk I/O read (MB total) | up to $1.2 \times 10^6$ | $5 \times 10^5$ |
| Disk utilization (%) | 70–75 | 72 |
| Network traffic (bytes total) | $\approx 3 \times 10^{13}$ | $3 \times 10^{13}$ |

Since all datasets are paired with models of identical size and the hardware envelope (GPU power, memory, temperature, CPU threads, I/O) is effectively the same across runs, any observed performance differences are attributable to the sampling strategy (noise schedule, ODE steps) rather than to model-size or hardware variations (the network traffic is due to Wandb usage).

All experiments were run on a single NVIDIA Quadro RTX 6000 GPU (Driver 570.124.06, CUDA 12.8), operating in persistence mode.

Each model was trained for 100 000 iterations. We explored architectures ranging from approximately 5 million to 10 million trainable parameters; across this range we observed no significant differences i n final predictive performance or convergence behavior. All other hyperparameters (learning rate schedules, batch sizes, noise schedules, etc.) were held fixed when evaluated against each other, ensuring that differences in accuracy and runtime reflect only the conditioning and sampling strategies under study.

# F  CODE AND DATA AVAILABILITY

Prior to acceptance, we will make available all three datasets, the complete data-processing scripts, and a fully functional training and evaluation pipeline for RecFlow, including a provided model checkpoint that can be loaded and tested. A more comprehensive, user-friendly code release will follow shortly thereafter.

# G  ADDITIONAL VISUALIZATIONS

**Test-time ensemble perturbation (evaluation only).**  Following the protocol of *GenCFD* Molinaro et al. (2024), we construct small "micro-ensembles" around each test initial condition $\bar{u}$ by sampling $M_{\mathrm{micro}}$ perturbed fields uniformly in an $\varepsilon$–ball centered at $\bar{u}$. Each perturbed member is then advanced to the evaluation time $t$ with a high-fidelity PDE solver to produce a reference ensemble $\{u_{\mathrm{ref}}^{(j)}(t)\}_{j=1}^{M_{\mathrm{micro}}}$. Our model generates a corresponding ensemble $\{u_{\mathrm{model}}^{(j)}(t)\}_{j=1}^{M_{\mathrm{micro}}}$ from the same perturbed inputs, but without ever observing ensembles during training (which only uses independent pairs $(u_0, u_1)$). We compare these two ensembles via mean-field error $e_\mu$, standard-deviation error $e_\sigma$, and spatially averaged 1-Wasserstein distance $\overline{W}_1$ (see **SM** §D).

**Out-of-distribution testing (cloud–shock and shear-flow).** Because the micro-perturbations render each initial field slightly OOD, this evaluation probes the model's ability to generalize under small chaotic perturbations. In these tasks, our Rectified Flow ensemble closely tracks the reference uncertainty and multiscale statistics, whereas deterministic baselines collapse to a single trajectory and fail to capture the conditional variability.

**In-distribution scaling (Richtmyer–Meshkov).** For the Richtmyer–Meshkov dataset, we additionally study the effect of training set size using a held-out test set from the same distribution. Here, even deterministic approaches improve with more data, but still underperform on the micro-ensemble test: they require significantly larger training sets to match RecFlow's accuracy and exhibit poor calibration under small perturbations.

In this appendix we provide a comprehensive set of qualitative comparisons between the baselines (GenCFD variants, UViT, FNO) models and our Rectified Flow (RecFlow) approach across three representative 2D fluid-flow benchmarks (Richtmyer–Meshkov, shear layer roll-up, and cloud-shock interaction). For each task we show:

- **Mean fields** Ensemble averages reveal how each deterministic model diffuses fine-scale features (UVit/FNO) versus perserving sharp interfaces (RecFlow, GenCFD variants).

- **Uncertainty maps** Per-pixel standard deviations illustrate whether uncertainty localizes in physically meaningful regions (e.g. shocks, shear interfaces) or remains overly diffuse.

- **Random samples** To qualitatively assess generative fidelity, we include representative random samples from RecFlow alongside outputs from selected baselines, specifically the deterministic FNO and the hybrid Diffusion ∘ FNO. The latter combines FNO's low-frequency structural prediction with GenCFD's generative modeling of fine-scale turbulence. This comparison highlights RecFlow's ability to produce realistic, high-resolution flow fields that preserve physical coherence across different scales.

- **Spectral & pointwise comparisons** (Fig. 22: Richtmyer–Meshkov global $E(k)$, local spectra, and pointwise density histograms; Fig. 25: Shear-layer global $u_x$ spectra and pointwise $u_x$ distributions; Fig. 29: Cloud–shock global $m_y$ spectra and pointwise energy distributions) — combined Fourier-space and histogram panels quantify multiscale fidelity and calibration of the predictive ensembles.

Together, these visualizations underscore RecFlow's ability to maintain sharper physical structures, produce better-calibrated uncertainties, and faithfully reproduce multiscale spectral statistics.

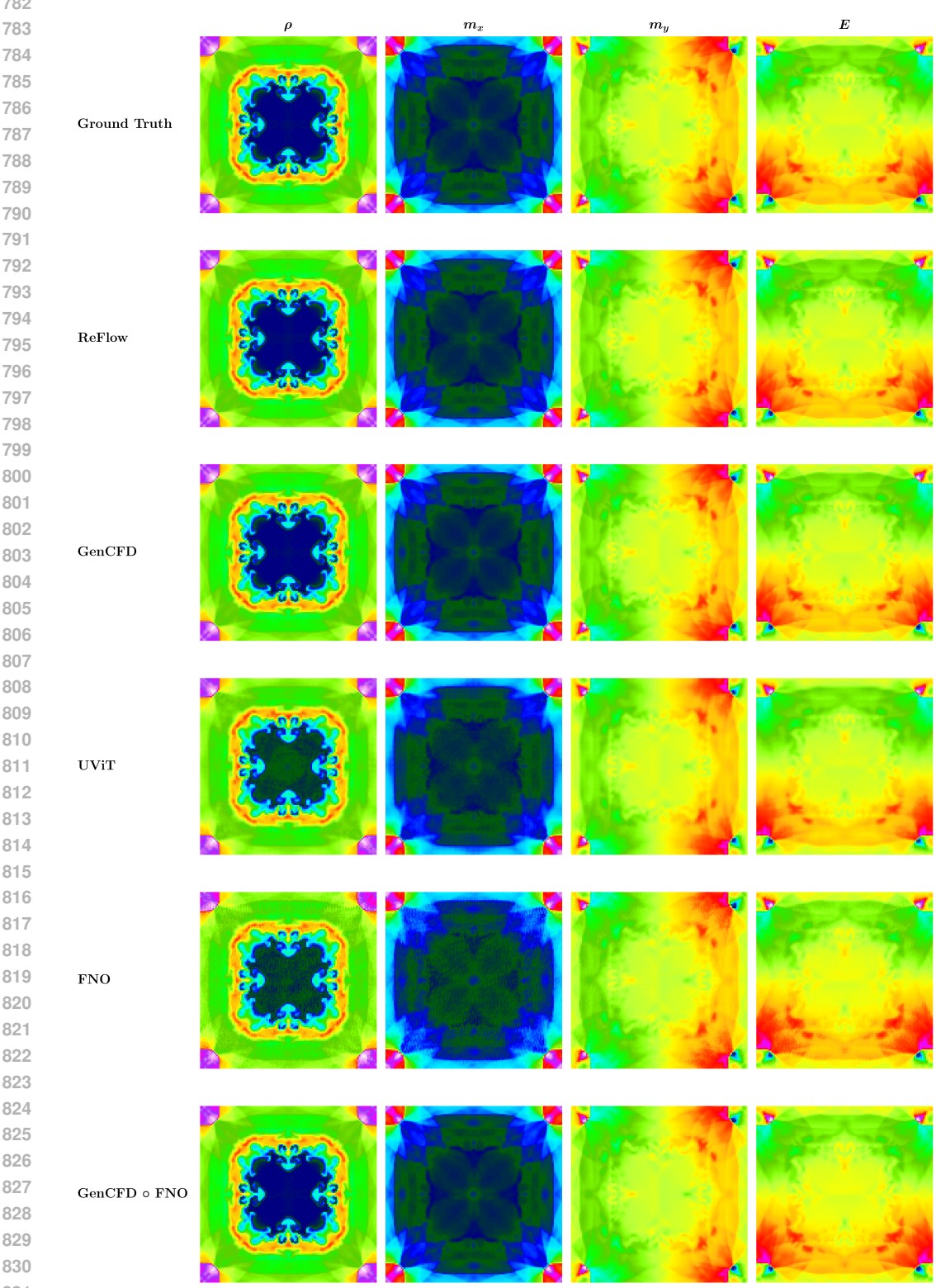

Figure 19: **Richtmyer–Meshkov Mean Field Comparison**: Mean velocity fields for the Richtmyer–Meshkov dataset, shown across all physical channels and models. In many cases, predictions closely resemble the ground truth, with differences often imperceptible, underscoring the overall high quality and consistency of the generated mean fields.

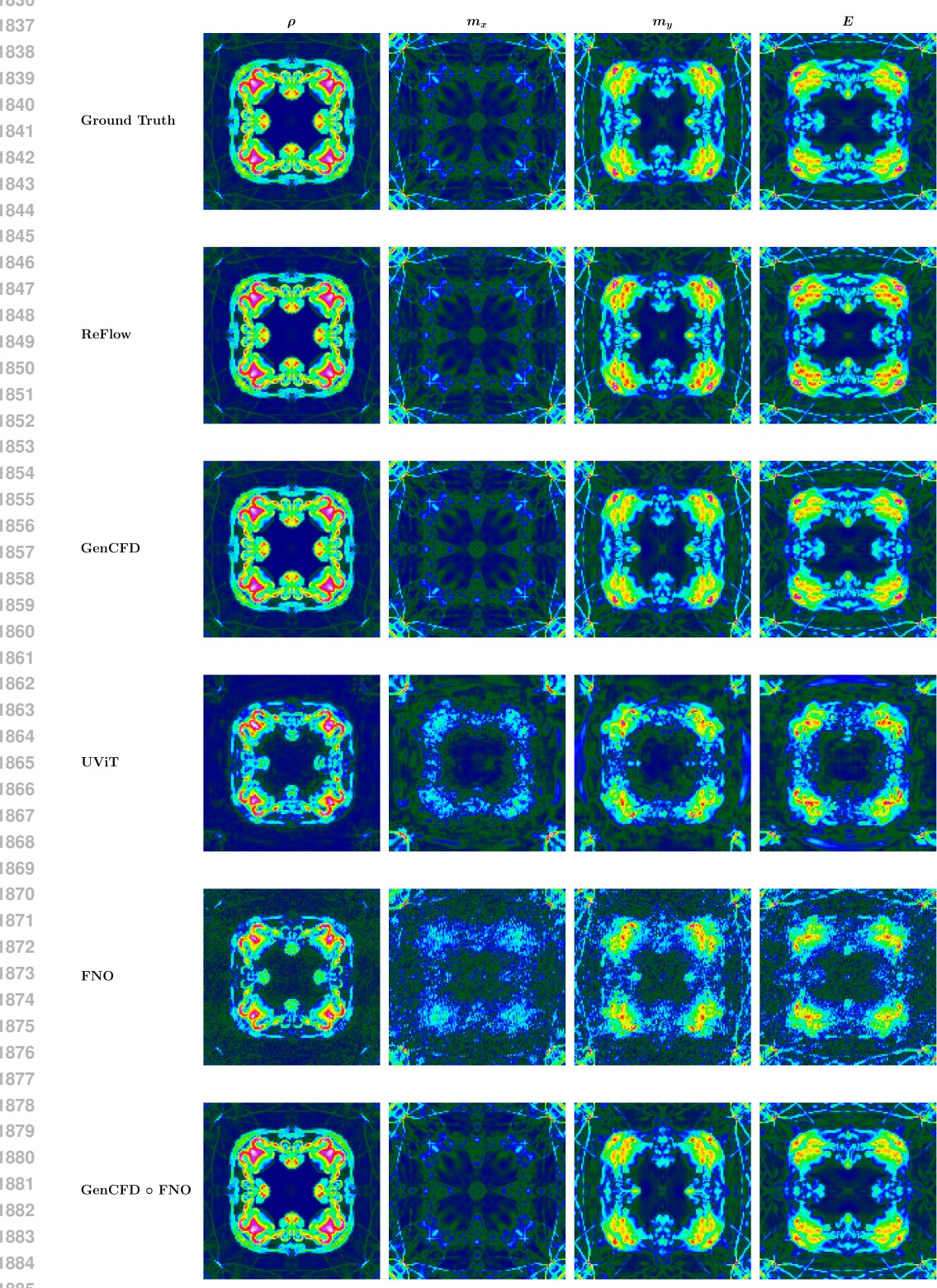

Figure 20: **Richtmyer–Meshkov Uncertainty Comparison**: Per-pixel standard deviation fields from the Richtmyer–Meshkov dataset across all models. Stochastic generative models capture spatially coherent uncertainty patterns, outperforming the deterministic baselines (FNO and UViT), which exhibit limited variability and poor uncertainty localization.

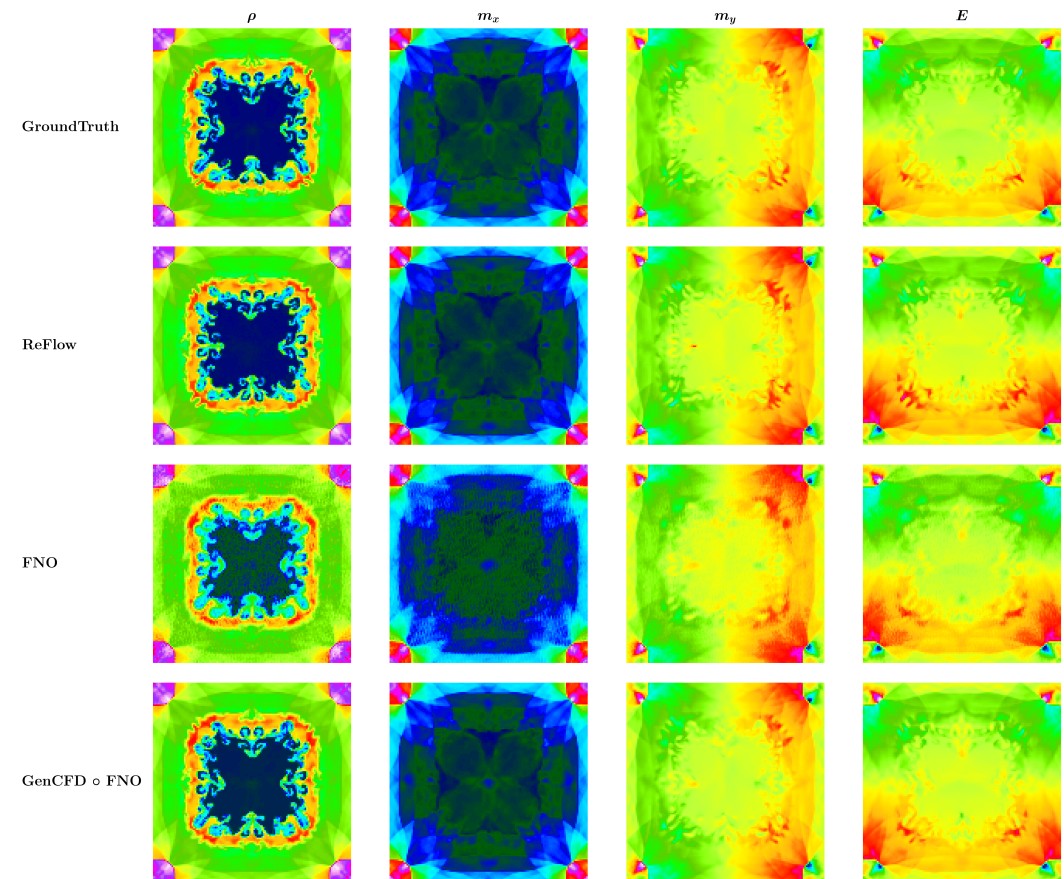

Figure 21: **Richtmyer–Meshkov Individual Sample Comparison**: A representative random test sample from the Richtmyer–Meshkov dataset, visualized across all four physical channels ($\rho$, $m_x$, $m_y$, $p$). All models produce physically plausible results, with only little variation in fidelity to fine-scale structures. ReFlow offers the closest visual match to the ground truth, preserving coherent details and sharp interfaces. Notably, the GenCFD ∘ FNO hybrid enhances spectral richness by reconstructing high-frequency features based on a low-frequency FNO prior, illustrating its capacity to recover sharper modes even when starting from a coarse prediction.

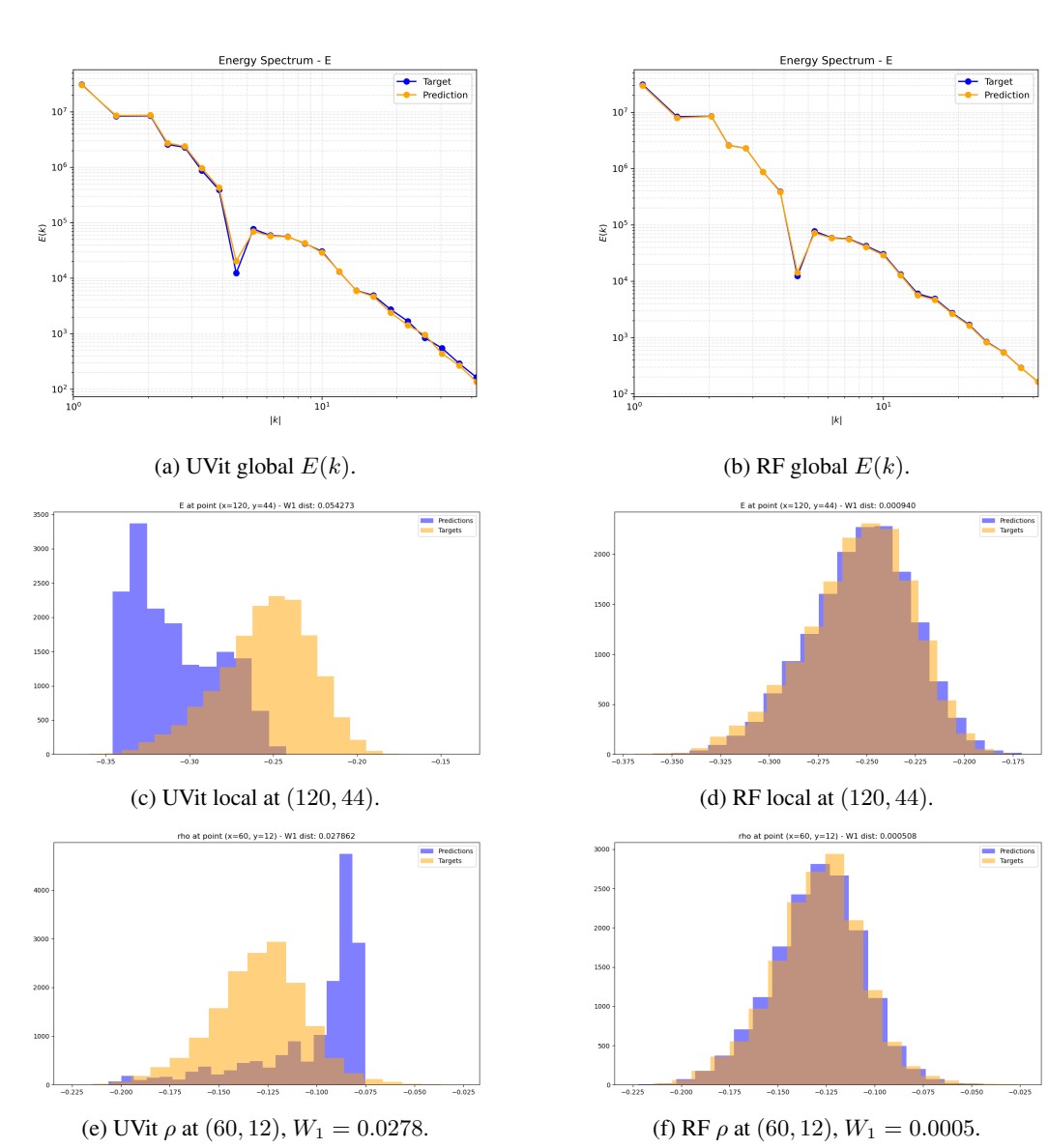

(a) UVit global $E(k)$.

(b) RF global $E(k)$.

(c) UVit local at $(120, 44)$.

(d) RF local at $(120, 44)$.

(e) UVit $\rho$ at $(60, 12)$, $W_1 = 0.0278$.

(f) RF $\rho$ at $(60, 12)$, $W_1 = 0.0005$.

Figure 22: Richtmyer–Meshkov: spectral and pointwise distribution comparisons between UVit and Rectified Flow.

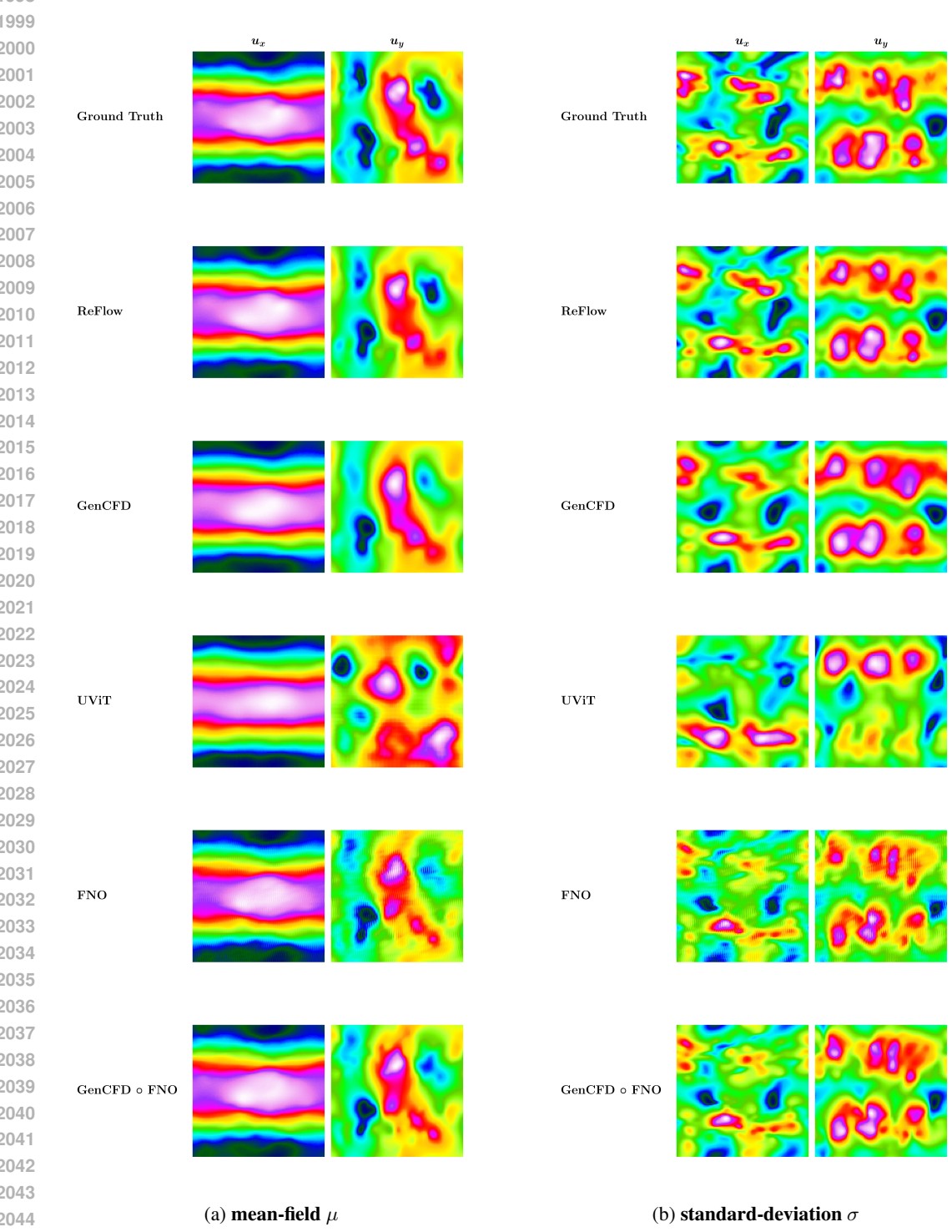

(a) **mean-field** $\mu$           (b) **standard-deviation** $\sigma$

Figure 23: **Shear-Layer Mean and Standard Deviation Comparison**: Visualization of mean-field ($\mu$) and standard-deviation ($\sigma$) predictions for the Shear-Layer dataset across both velocity components ($u_x$, $u_y$). For each model, the left half shows the predicted mean fields and the right half shows per-pixel standard deviations. The top row contains the reference ground truth. Despite the reduced channel complexity of this dataset, clear differences emerge in how well models capture coherent structures and localized uncertainties. ReFlow preserves sharper features and more accurately reflects the spatial distribution of uncertainty.

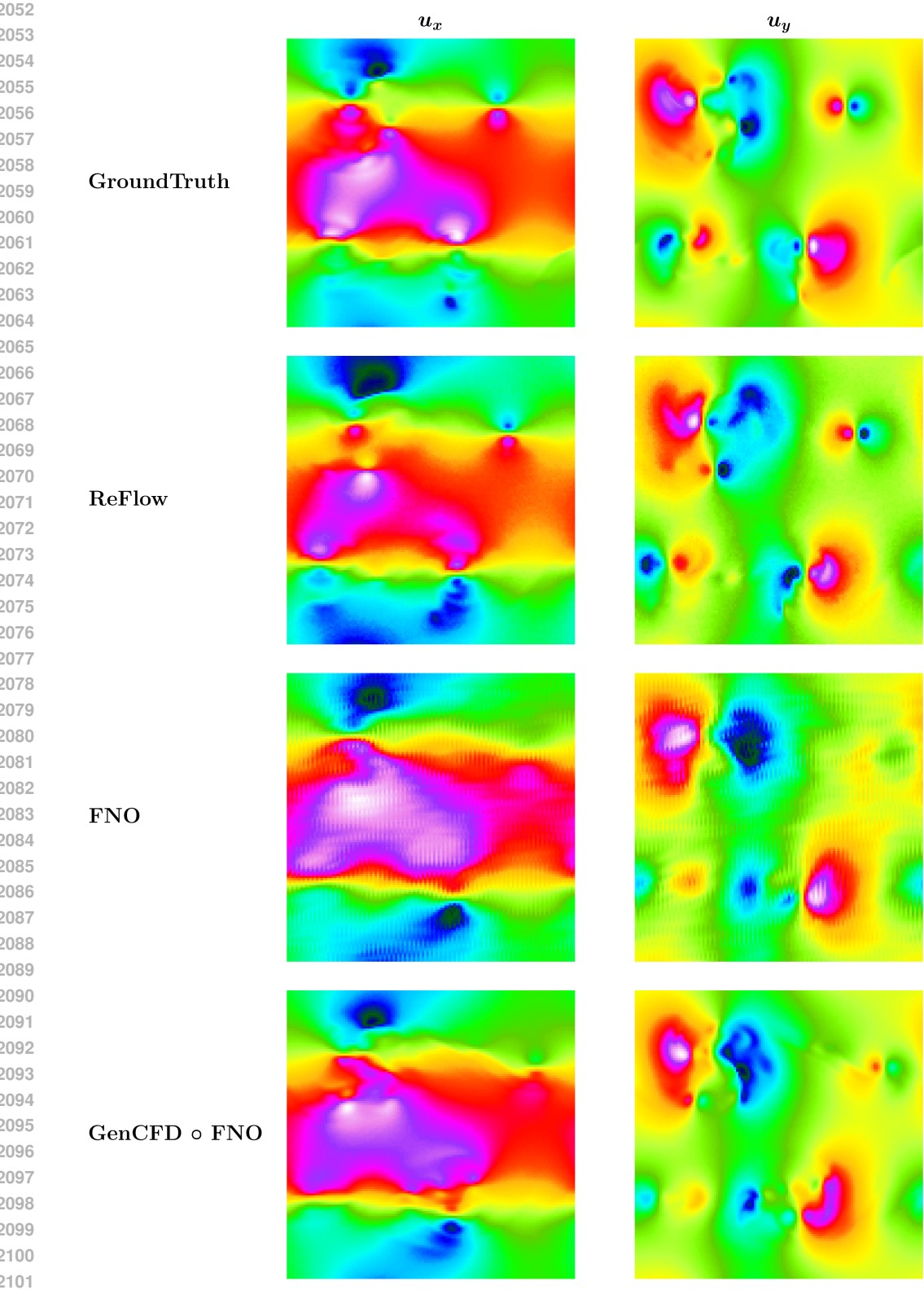

Figure 24: **Shear-Layer Individual Sample Comparison**: A representative random test sample from the Shear-Layer dataset, shown across both physical channels ($u_x$, $u_y$). All models generate physically plausible results, though with varying degrees of detail and sharpness. ReFlow and GenCFD ∘ FNO most closely resemble the ground truth, preserving fine-scale structures and coherent interfaces. GenCFD ∘ FNO enhances high-frequency detail by leveraging the low-frequency prior from the initial FNO prediction.

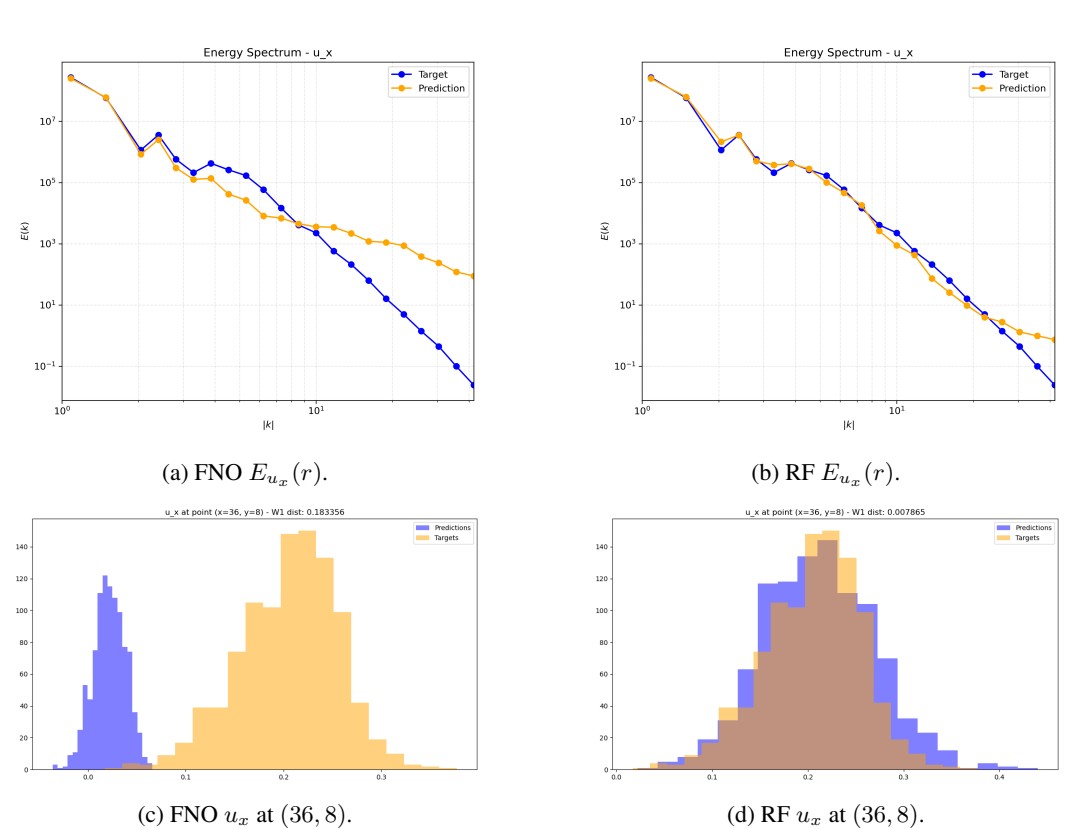

(a) FNO $E_{u_x}(r)$.

(b) RF $E_{u_x}(r)$.

(c) FNO $u_x$ at $(36, 8)$.

(d) RF $u_x$ at $(36, 8)$.

Figure 25: Shear-layer: global $u_x$ spectra and pointwise $u_x$ distributions.

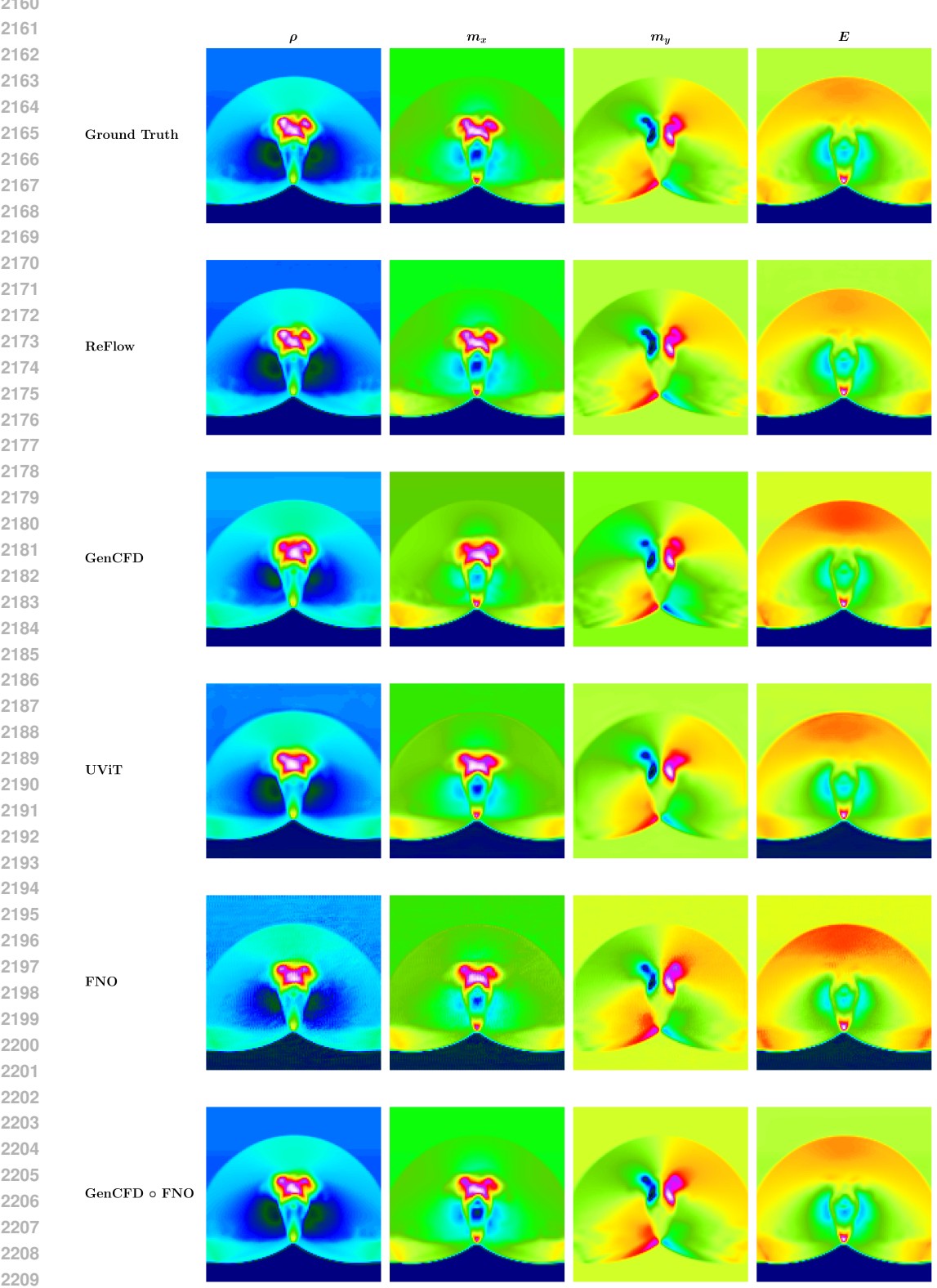

Figure 26: **Cloud-Shock Mean Field Comparison**: Mean velocity fields for the Cloud-Shock dataset, shown across all physical channels and models. In most cases, predictions closely resemble the ground truth.

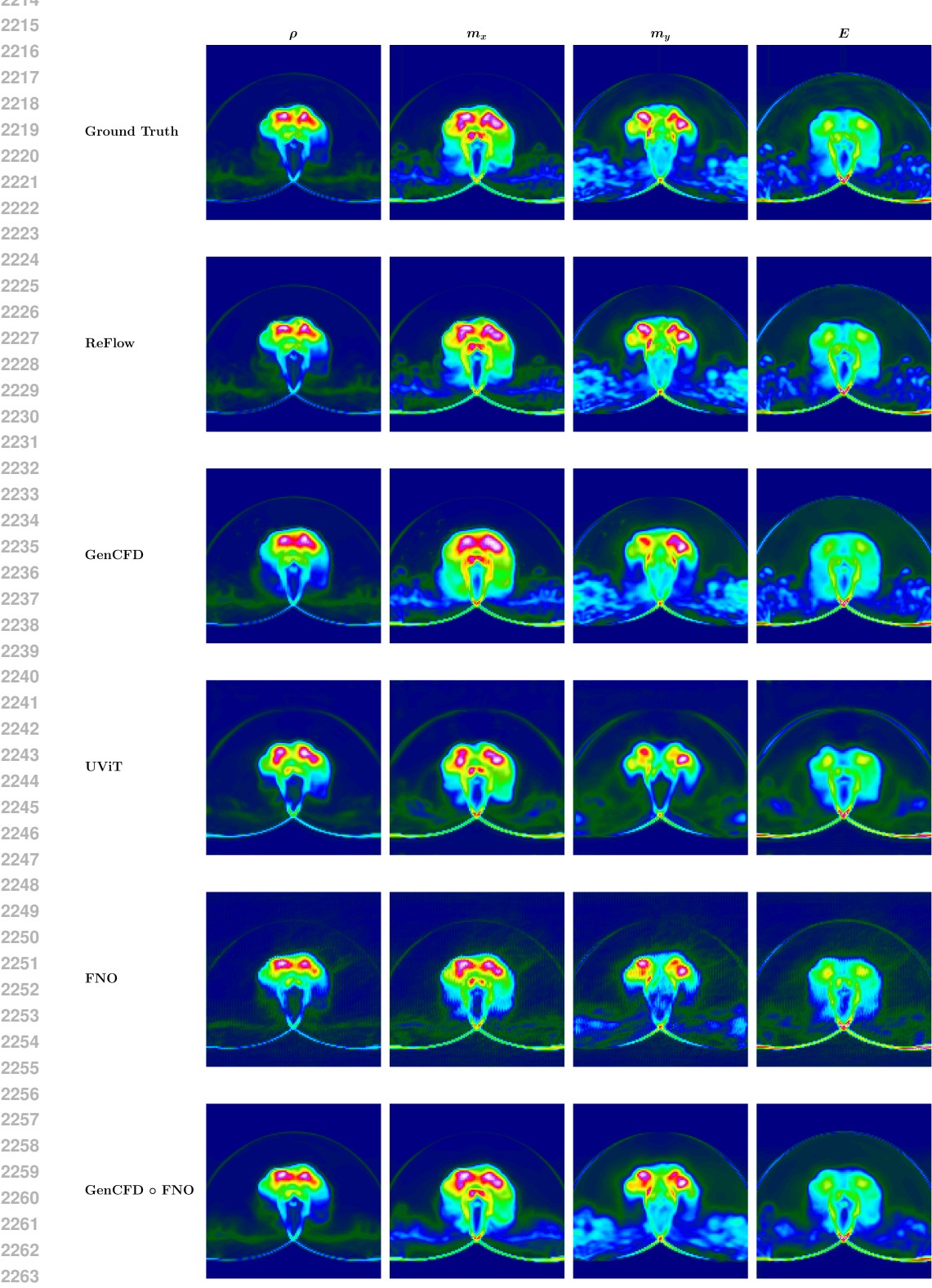

Figure 27: **Cloud–Shock Uncertainty Comparison**: Per-pixel standard deviation visualizations for the Cloud–Shock dataset across all models. While all methods capture key uncertainty patterns, stochastic models produce more localized and physically meaningful variability.

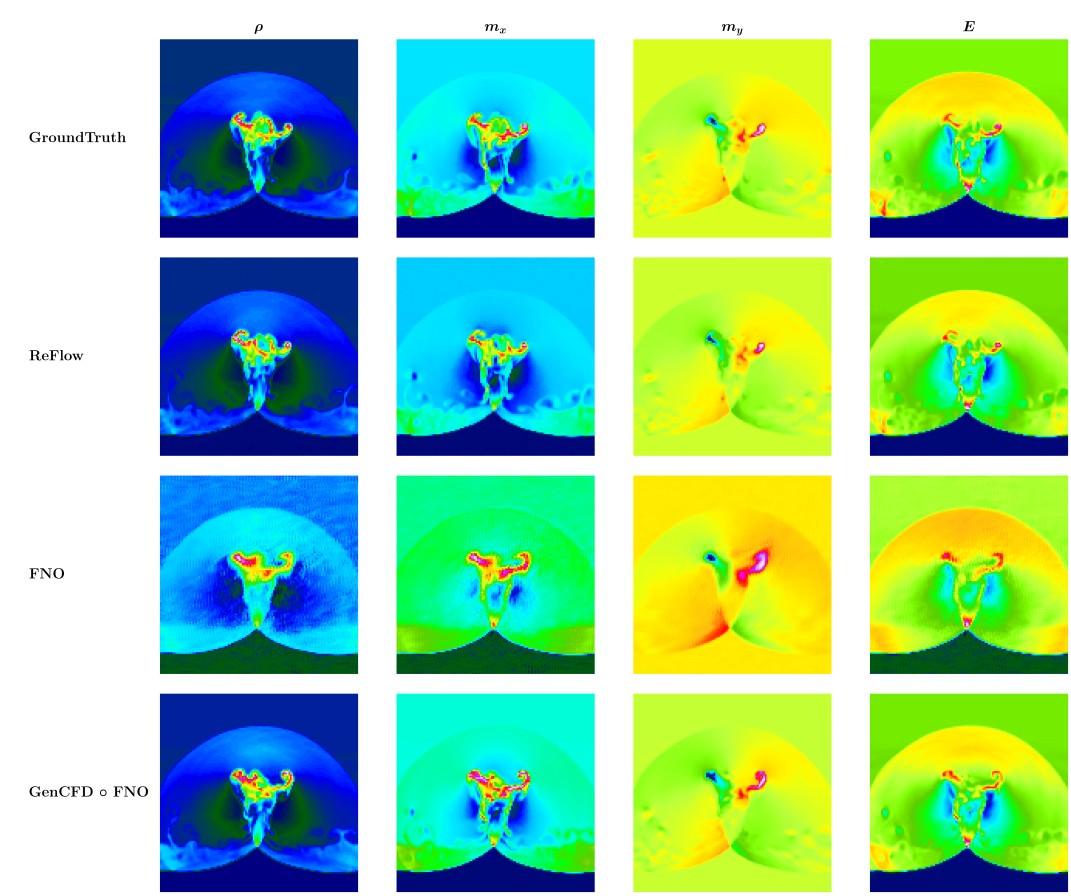

Figure 28: **Cloud–Shock Individual Sample Comparison**: A representative random test sample from the Cloud–Shock dataset, displayed across all physical channels. While all models produce plausible results, ReFlow and the hybrid GenCFD ∘ FNO most closely align with the ground truth, particularly in preserving fine-scale detail. The spectral reconstruction effect of the hybrid model is especially evident, recovering high-frequency structures from the coarse FNO prediction and enhancing the overall visual fidelity.

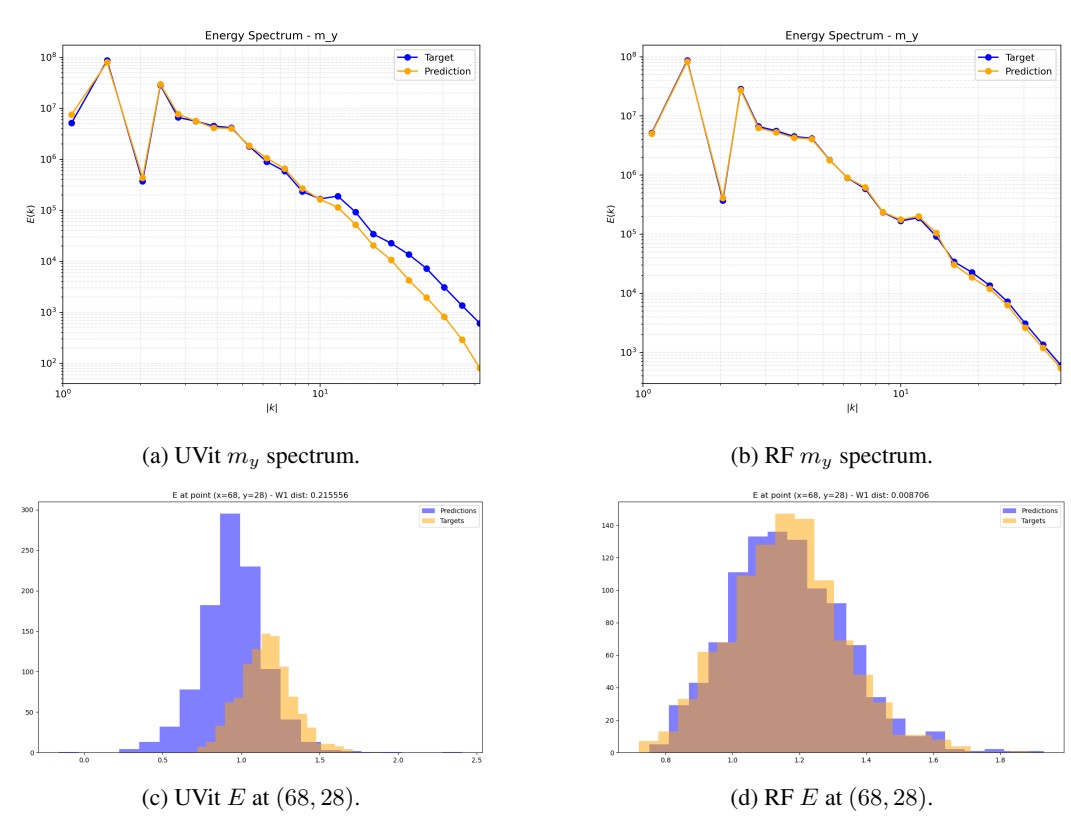

(a) UVit $m_y$ spectrum.

(b) RF $m_y$ spectrum.

(c) UVit $E$ at $(68, 28)$.

(d) RF $E$ at $(68, 28)$.

Figure 29: Cloud-shock: spectral and pointwise energy comparisons.

