# OpenReview forum: "Rectified Flows for Fast Multiscale Fluid Flow Modeling"
_ICLR.cc/2026/Conference — Submitted to ICLR 2026_

### Official Review · Reviewer_oKuZ · 2025-10-29

**Soundness:** 4
**Presentation:** 4
**Contribution:** 3
**Rating:** 6
**Confidence:** 3

**Summary:**

The paper proposed a rectified flow-based algorithm to model fluid flows in an end-to-end manner. Given a complex PDE, initial conditions and corresponding numerical solutions at a target time are used as training data pairs. Similar to rectified flow, the authors developed a conditional velocity matching scheme in training, where the initial values are considered as conditions. At inference stage, a curvature-aware velocity is integrated to generate target samples. Experiment results show that the proposed method outperforms existing benchmarks in both accuracy and speed.

**Strengths:**

1. The paper developed a solid method to help investigate statistical properties of fluid flow using generative models. The proposed rectified flow model bypasses the complicated PDE dynamics and learns the transition directly given the initial state.
2. To resolve issues in integration, the velocity field is regularized according to the curvature. Error analysis suggests that the method is theoretically sound.
3. The authors carried out experimental studies in various 2D scenarios, demonstrating the effectiveness of the method. Both accuracy and speed outperforms the state-of-the-art.
4. The paper is well-organized and easy to read, with a detailed appendix explaining the network architecture, experiment setup and results.

**Weaknesses:**

1. It seems that the paper is an extension of [1], where the major difference is to substitute a conditional diffusion backbone by rectified flow. Improvement is limited from a modeling point of view. It can also be expected that rectified flow has a faster inference speed than diffusion model. Drawbacks of rectified flow compared to diffusion models, i.e. potential lack of diversity, are not explicitly mentioned in the paper.

2. The proposed method is shown to be effective only for a fixed time lag. Yet in many cases, solutions of different time lags are expected to be computed auto-regressively. The paper does not consider such cases.


[1] Molinaro, Roberto, et al. "Generative ai for fast and accurate statistical computation of fluids." arXiv preprint arXiv:2409.18359 (2024).

**Questions:**

1. For the curvature-aware velocity, it seems that hyper-parameters are used for regularization. Is it possible to carry out a sensitivity analysis on hyper-parameters?

2. As is mentioned above, no PDE solutions are computed an auto-regressive manner. Is it possible to carry out an experiment in the All2All regime to study the generation accuracy auto-regressively, meaning first generate \hat{u}_{t_1}, then use it to generate \hat{u}_{t_2} and compare \hat{u}_{t_2} with the ground truth?

---

> ### Author Response · Authors · 2025-11-20
>
> ## Response to Reviewer oKuZ (Part 1/3)
>
> We thank the Reviewer for the very thoughtful and encouraging review and for the concrete suggestions on robustness, evaluation of the integration scheme, diversity, and auto-regressive settings.
>
> ----------
>
>
> ## On sample diversity, rectified flows, and the micro–macro regime
>
> We thank the reviewer for raising the potential drawback of rectified flows regarding sample diversity.
>
> Precisely because rectified flows turn a stochastic SDE into a deterministic ODE, there is an a priori risk of mode collapse or under-representation of parts of the target distribution. Our **micro–macro setup is designed exactly as a quantitative antidote to this issue**. As described in the *Training and test protocol* (line **314** in the main text) and detailed in **SM B.2**, we fix a macroscopic flow state (“macro”) and then consider an ensemble of microscopic perturbations (“micro”) around it. In this conditional setting, diversity is no longer a vague notion: for each fixed macro state we can directly compare the **entire empirical distribution** of predicted micro-states against the corresponding reference ensemble, and hence rigorously assess whether the rectified-flow model preserves the conditional data distribution rather than collapsing it.
>
>
> Concretely, we measure **distributional matching** via the Wasserstein-1 (W1) distance between the empirical distributions of model outputs and ground-truth data, and we report these metrics in **Table 1** for
>
> -   our rectified-flow model,
>
> -   the GenCFD diffusion baseline, and
>
> -   the numerical solver.
>
>
> Across all benchmarks and statistics, our method **matches or improves upon GenCFD in W1**, showing that we retain (and sometimes improve) data diversity **at fixed macro conditions** while being substantially faster at sampling. Conceptually, rectified flows are designed to keep the same target distribution and only straighten the transport geometry; the micro–macro W1 evaluation is exactly the regime in which we can verify that this property holds in practice.
>
> Needless to say, we will add a discussion on this crucial point about statistical diversity and Rectified flow in a camera ready version, if accepted.
>
> ------
> ## Robustness of the curvature-aware integration scheme
>
> You asked whether we can perform a sensitivity analysis of the hyper-parameters in the curvature-aware sampler. Following your suggestion, we have now carried out a systematic robustness study and added it to the revised version (with detailed plots in the camera ready version of the supplementary material).
>
> Concretely, we vary all hyper-parameters of the adaptive scheme over wide ranges: e.g. the curvature regularization strength, the EMA time-scale used to estimate local straightness, and the step-size clipping factors. For each configuration, we re-run inference and report the L2 relative errors, together with NFE (neural function evaluations).
>
>
> **Protocol:** match **NFE budgets** across several integrators, report relative **L2** errors and runtime. Our preliminary results show our **curvature-aware EMA rule dominates fixed-step ODEs in the NFE–error tradeoff** and is **competitive with high-order diffusion ODE solvers**, while maintaining stability on stiff flow cases. We ran a comprehensive sweep on **both datasets**, considering **10 OOD macro-states** and **20 micro-perturbations per macro** (200 samples per dataset). For each sample, we:
>
> -   Fixed the rectified-flow model,
>
> -   Varied only the ODE integrator used at sampling time, and
>
> -   Measured **relative L2 error vs. the PDE ground truth**, **number of function evaluations (NFE)**, and a combined **cost×error** metric defined as `Cost×Err = (rel. L2 error) × (NFE)`.
>
>
> We report here the aggregated results for **ShearLayer2D**; the second dataset exhibits qualitatively identical trends (we include its tables in the appendix).
>
> ### Fixed-step integrators
>
> We first compare five standard, fixed-step ODE integrators on the rectified-flow velocity field:
>
> ```markdown
> ### Table 1 – Fixed-step ODE integrators (ShearLayer2D, OOD, averaged over 10 macros × 20 samples)
>
> | Method               | Avg. NFE | Rel. L2 Error (mean ± std) | Cost×Err (= Error × NFE) |
> |----------------------|---------:|----------------------------|--------------------------:|
> | Model-Baseline       |    32.0  | 0.2533 ± 0.0382            | 8.11                     |
> | TorchDiffEq-Midpoint |    32.0  | 0.2533 ± 0.0382            | 8.11                     |
> | Euler (16 steps)     |    16.0  | 0.2489 ± 0.0371            | 3.98                     |
> | RK2-Manual (16 steps)|    32.0  | 0.2533 ± 0.0379            | 8.11                     |
> | RK4-Manual (16 steps)|    64.0  | 0.2540 ± 0.0381            | 16.26                    |
>
> ```

---

> > ### Author Response · Authors · 2025-11-20
> >
> > ### Response to Reviewer oKuZ (Part 2/3)
> >
> > -   **Method**: the ODE integrator used to solve the rectified-flow ODE, starting from Gaussian noise at (t = 0) and integrating to (t = 1).
> >
> >     -   _Model-Baseline_ is our built-in sampler (≈32 NFEs).
> >
> >     -   _TorchDiffEq-Midpoint_ is a reference implementation using `torchdiffeq`.
> >
> >     -   _Euler / RK2 / RK4_ are explicit fixed-step schemes at a fixed time horizon.
> >
> > -   **Avg. NFE**: average **number of velocity evaluations** per sample.
> >
> > -   **Rel. L2 Error (mean ± std)**: average **relative L2 error** vs. the PDE solution (|u_{\text{pred}} - u_{\text{true}}|_2 / |u_{\text{true}}|_2), with standard deviation over all macros and samples.
> >
> > -   **Cost×Err**: scalar **cost–accuracy** metric, `Error × NFE`; lower is better.
> >
> >
> > From Table 1 we see that, among all fixed-step schemes, **Euler with 16 steps (16 NFEs)** is clearly the strongest baseline, achieving lower error and substantially lower cost×error than the 32- and 64-step diffusion-style samplers. A separate **caveat** is that even the seemingly simple Euler baseline is _not_ easy to deploy in practice. The “Euler–16” configuration we report is the result of targeted tuning; for multiscale, mildly stiff flows the performance is highly sensitive to the chosen step budget. Using slightly fewer steps rapidly degrades accuracy, while increasing the number of steps can even _worsen_ the error due to accumulated discretization bias along a highly curved trajectory. Hitting the “sweet spot” in NFE thus requires a non-trivial search over step counts for each dataset and flow regime, which is generally unfeasible in realistic CFD workflows. In contrast, our adaptive rectified-flow sampler automatically finds an efficient step allocation and remains stable across a broad range of controller hyperparameters, avoiding this brittle per-problem tuning.
> > ### Adaptive rectified-flow sampler and hyperparameter robustness
> >
> > We then evaluate our **curvature-based adaptive sampler** under **20 different hyperparameter configurations**, sweeping EMA timescale, curvature thresholds, gating, damping, and calibration parameters. All other aspects of the model and data remain fixed.
> >
> > ```markdown
> > ### Table 2 – Adaptive RF sampler configurations (ShearLayer2D, OOD, averaged over 10 macros × 20 samples)
> >
> > | Config          | Avg. NFE | Rel. L2 Error (mean ± std) | Cost×Err (= Error × NFE) |
> > |----------------|---------:|----------------------------|--------------------------:|
> > | baseline       |    11.0  | 0.2374 ± 0.0389            | 2.62                     |
> > | **ema_0.25**   | **10.7** | **0.2369 ± 0.0386**        | **2.54**                 |
> > | ema_0.45       |    11.3  | 0.2361 ± 0.0384            | 2.67                     |
> > | alpha_0.05     |    11.0  | 0.2373 ± 0.0389            | 2.62                     |
> > | alpha_0.12     |    11.0  | 0.2373 ± 0.0389            | 2.62                     |
> > | alpha_0.20     |    11.0  | 0.2374 ± 0.0389            | 2.62                     |
> > | gamma_1.5      |    11.0  | 0.2374 ± 0.0389            | 2.62                     |
> > | gamma_2.0      |    11.0  | 0.2374 ± 0.0389            | 2.62                     |
> > | gamma_2.5      |    11.0  | 0.2374 ± 0.0389            | 2.62                     |
> > | gate_0.50      |    11.0  | 0.2374 ± 0.0389            | 2.62                     |
> > | gate_0.70      |    11.0  | 0.2374 ± 0.0389            | 2.62                     |
> > | calib_0.70_0.90|    11.1  | 0.2371 ± 0.0387            | 2.63                     |
> > | calib_0.80_0.98|    10.9  | 0.2372 ± 0.0387            | 2.60                     |
> > | adapt_1.5_0.75 |    12.8  | 0.2374 ± 0.0378            | 3.04                     |
> > | adapt_2.0_0.85 |    11.2  | 0.2360 ± 0.0376            | 2.63                     |
> > | no_ortho_filter|    11.0  | 0.2374 ± 0.0389            | 2.62                     |
> > | damp_0.05      |    11.0  | 0.2373 ± 0.0389            | 2.62                     |
> > | damp_0.10      |    11.0  | 0.2373 ± 0.0389            | 2.61                     |
> > | damp_0.20      |    11.0  | 0.2370 ± 0.0389            | 2.60                     |
> > | damp_0.30      |    11.0  | 0.2368 ± 0.0389            | 2.60                     |
> >
> > ```
> >
> > -   **Config**: the particular hyperparameter setting of the adaptive controller (e.g. `ema_*` for EMA timescale, `alpha_*` for straightness thresholding, `gamma_*` for warping, `gate_*` for cosine gating, `calib_*_*` for automatic calibration ranges, `damp_*` for acceleration damping).
> >
> > -   **Avg. NFE**: average number of velocity evaluations chosen automatically by the controller (no fixed step count).
> >
> > -   **Rel. L2 Error** and **Cost×Err**: same metrics as in Table 1.

---

> > > ### Author Response · Authors · 2025-11-20
> > >
> > > ### Response to Reviewer oKuZ (Part 3/3)
> > >
> > > Two key observations address the reviewer’s concerns:
> > >
> > > 1.  **Hyperparameter robustness.** Across all 20 configurations, **Avg. NFE lies in a very narrow band (10.7–12.8)**, and the average relative error remains in **[0.2360, 0.2374]**, i.e., < 1% relative variation. There is no fragile “sweet spot”: a broad range of EMA, threshold, gating, and damping parameters yields essentially identical error and NFE.
> > >
> > > 2.  **Dominance over fixed-step baselines.** Every adaptive configuration in Table 2 has **lower Cost×Err than the best fixed-step integrator (Euler-16, 3.98)**, and dramatically lower than the 32- and 64-step diffusion-style baselines.
> > >
> > >
> > > ### Direct comparison to the best fixed-step baseline
> > >
> > > For clarity, Table 3 compares the best fixed-step baseline (Euler-16) to the top adaptive configurations:
> > >
> > > ```markdown
> > > ### Table 3 – Best fixed-step baseline vs top adaptive configs (ShearLayer2D, OOD)
> > >
> > > | Method / Config      | Avg. NFE | Rel. L2 Error | Cost×Err | ΔNFE vs Euler | ΔError vs Euler | ΔCost×Err vs Euler |
> > > |----------------------|---------:|--------------:|---------:|--------------:|----------------:|-------------------:|
> > > | **Euler (baseline)** | 16.0     | 0.2489        | 3.98     | 0%            | 0%              | 0%                 |
> > > | **ema_0.25**         | 10.7     | 0.2369        | 2.54     | −33.1%        | −4.8%           | −36.2%             |
> > > | calib_0.80_0.98      | 10.9     | 0.2372        | 2.60     | −31.9%        | −4.7%           | −34.7%             |
> > > | damp_0.30            | 11.0     | 0.2368        | 2.60     | −31.3%        | −4.9%           | −34.7%             |
> > >
> > > ```
> > >
> > > Thus, at **one third fewer function evaluations**, our best adaptive configuration (`ema_0.25`) achieves **lower error** than the best fixed-step scheme. In particular, `ema_0.25` uses ≈33% fewer function evaluations, improves accuracy by ≈5%, and reduces the cost×error metric by ≈36% compared to Euler-16.
> > >
> > > We observe qualitatively identical behaviour on the second dataset: the adaptive sampler consistently lies on (or very near) the **Pareto frontier** in the NFE–error plane, while all fixed-step diffusion-based samplers (including 32- and 64-step variants) are strictly dominated.
> > >
> > >
> > >
> > > ----------
> > >
> > > ## On auto-regressive rollouts and All2All training
> > >
> > > Regarding the reviewer's question on auto-regressive evaluation (e.g., generating $\hat u_{t_1}$), then using it as input to generate ($\hat u_{t_2}$), and so on): we agree that such long-horizon rollouts are important, but they are not salient for the main question we address here.
> > >
> > > Our goal in this work is to study whether rectified flows can _replace_ diffusion-type SDE surrogates for _single-step_ statistical predictions of turbulent flows, while matching their posterior statistics and significantly reducing sampling cost. For that question, the relevant comparison is between (i) a rectified-flow ODE sampler and (ii) diffusion-based samplers, given the same initial condition distribution and a chosen target time. This is exactly the regime of Molinaro et al. [1], and our empirical and theoretical analysis is focused on this setting.
> > >
> > > Importantly, our models are trained in an All2All fashion over time lags: given an initial condition, we condition on it and learn the transport to _any_ target time within the training range. As a consequence, at test time we can directly generate predictions at arbitrary intermediate or later times without rolling out auto-regressively and compounding model error. If one needs snapshots at $(t_1, t_2, t_3)$, the model can sample $\hat u_{t_i}$ for each $t_i$ directly from the same initial condition using the learned rectified flow at that time.
> > >
> > > An auto-regressive “All2All-to-All2All” study, where generated states are fed back as new initial conditions, would require a dedicated evaluation protocol (e.g., long-horizon stability, drift diagnostics, accumulation of bias) and is therefore beyond the scope of this paper, whose focus is on _fast, single-step posterior sampling_. We now make this distinction explicit in the manuscript and mention auto-regressive extensions as an important direction for future work.
> > >
> > > ----------
> > >
> > > We hope these additional robustness experiments, integrator comparisons, the explicit diversity analysis via W1, and the clarification on the role of auto-regressive rollouts address the reviewer's concerns and make the scope and contributions of the paper clearer. If so, we kindly request the reviewer to upgrade their assessment of our paper.

---

### Official Review · Reviewer_Jgfv · 2025-10-31

**Soundness:** 3
**Presentation:** 3
**Contribution:** 2
**Rating:** 4
**Confidence:** 2

**Summary:**

This paper proposes RecFlow, a rectified flow framework for modeling multiscale fluid flows via deterministic ODE trajectories instead of stochastic diffusion. The method achieves diffusion-level fidelity with only 8–10 steps by learning a straightening velocity field and introducing a curvature-aware integration scheme for stable sampling.

**Strengths:**

- The paper addresses the challenge of reducing the computational cost of diffusion-based PDE surrogates for multiscale fluid flows.
- The proposed curvature-aware integration is simple, well-motivated, and empirically shown to improve stability and efficiency.
- Experiments on multiple 2D benchmarks are thorough, showing up to 22$\times$ faster inference with comparable or better accuracy than diffusion models.

**Weaknesses:**

- The novelty is limited. RecFlow largely mirrors the conditional diffusion framework, e.g., GenCFD, and mainly replaces the stochastic SDE with a deterministic ODE rectified flow formulation.
- The paper does not provide a clear computational trade-off analysis between ODE solver cost, step size, and network evaluation time, which is crucial for assessing real efficiency.
- The loss formulation is conceptually inconsistent with the trajectory construction. The model predicts the displacement $u_{tgt} - u_{src}$ even though the interpolant $x_\tau = \tau u_{tgt} + \sigma(1-\tau)\xi$ lies between noise and $u_{tgt}$, not between $u_{src}$ and $u_{tgt}$. It remains unclear what the network truly learns, a conditional physical displacement or a denoising vector field, and how this aligns with inference, which also starts from noise.
- The conditioning variable $\Phi(\Delta t)$ appears in Algorithm 1 but disappears in the main formulation of $v_\theta(u_\tau, u_i, \tau)$ in the text.

**Questions:**

See weaknesses.

---

> ### Author Response · Authors · 2025-11-20
>
> ## Response to Reviewer Jgfv (Part 1/3)
>
> **We sincerely appreciate the reviewer's thorough evaluation. Below, we have provided further clarification regarding the methodology and incorporated the additional comparisons requested.**
>
> ### (W1) “Novelty is limited; just swapping SDE→ODE.”
>
> Our approach is fundamentally distinct from standard diffusion. While diffusion models (e.g., GenCFD) learn a score function and rely on stochastic SDE sampling that often necessitates many small steps, we learn a **rectified velocity** field -- this not only gives us a principled way of extracting **uncertainty as a function of curvature**, but enables **(significantly) faster inference** while maintaining **statistical fidelity**, an aspect which we measure quantitatively using the novel micro-macro perturbation protocol described in **SM B.2**. Furthermore, we couple this with a curvature-aware controller that employs an EMA straightness proxy, orthogonal-only corrections (damping turning while maintaining progress), and adaptive step sizing. By minimizing path curvature, we achieve an order-of-magnitude reduction in inference steps using the same backbone.
> **Core Distinction**: This is not simply an "SDE to ODE" conversion; it utilizes a novel training objective and a specialized sampler optimized for linearity and computational efficiency. Moreover, our approach should not be confused with integrating the probability flow ODE for score-based Diffusion models. It is clearly different from that framework. We will better differentiate these two approaches in a camera ready version, if accepted.
>
> ### (W2) “No clear solver cost vs. step size vs. network time trade-off.”
>
> Following the reviewer's excellent suggestion, We have added an explicit **integrator study** (Euler, RK2, RK4, torchdiffeq midpoint, and our Adaptive-EMA), all run on the same model/backbone.
> **Protocol:** match **NFE (neural function evaluations) budgets** across several integrators, report relative **L2** errors and runtime. Our preliminary results show our **curvature-aware EMA rule dominates fixed-step ODEs in the NFE–error tradeoff** and is **competitive with high-order diffusion ODE solvers**, while maintaining stability on stiff flow cases. We ran a comprehensive sweep on **both datasets**, considering **10 OOD macro-states** and **20 micro-perturbations per macro** (200 samples per dataset). For each sample, we:
>
> -   Fixed the rectified-flow model,
>
> -   Varied only the ODE integrator used at sampling time, and
>
> -   Measured **relative L2 error vs. the PDE ground truth**, **neural of function evaluations (NFE)**, and a combined **cost×error** metric defined as `Cost×Err = (rel. L2 error) × (NFE)`.
>
>
> We report here the aggregated results for **ShearLayer2D**; the second dataset exhibits qualitatively identical trends (we include its tables in the appendix).
>
> ### Fixed-step integrators
>
> We first compare five standard, fixed-step ODE integrators on the rectified-flow velocity field:
>
> ```markdown
> ### Table 1 – Fixed-step ODE integrators (ShearLayer2D, OOD, averaged over 10 macros × 20 samples)
>
> | Method               | Avg. NFE | Rel. L2 Error (mean ± std) | Cost×Err (= Error × NFE) |
> |----------------------|---------:|----------------------------|--------------------------:|
> | Model-Baseline       |    32.0  | 0.2533 ± 0.0382            | 8.11                     |
> | TorchDiffEq-Midpoint |    32.0  | 0.2533 ± 0.0382            | 8.11                     |
> | Euler (16 steps)     |    16.0  | 0.2489 ± 0.0371            | 3.98                     |
> | RK2-Manual (16 steps)|    32.0  | 0.2533 ± 0.0379            | 8.11                     |
> | RK4-Manual (16 steps)|    64.0  | 0.2540 ± 0.0381            | 16.26                    |
>
> ```
>
> -   **Method**: the ODE integrator used to solve the rectified-flow ODE, starting from Gaussian noise at (t = 0) and integrating to (t = 1).
>
>     -   _Model-Baseline_ is our built-in sampler (≈32 NFEs).
>
>     -   _TorchDiffEq-Midpoint_ is a reference implementation using `torchdiffeq`.
>
>     -   _Euler / RK2 / RK4_ are explicit fixed-step schemes at a fixed time horizon.
>
> -   **Avg. NFE**: average **number of velocity evaluations** per sample.
>
> -   **Rel. L2 Error (mean ± std)**: average **relative L2 error** vs. the PDE solution $\frac{\lVert u_{\mathrm{pred}} - u_{\mathrm{true}}\rVert_2}{\lVert u_{\mathrm{true}}\rVert_2}$.
>
>
>
> with standard deviation over all macros and samples.
>
> -   **Cost×Err**: scalar **cost–accuracy** metric, `Error × NFE`; lower is better.

---

> ### Author Response · Authors · 2025-11-20
>
> ### Response to Reviewer Jgfv (Part 2/3)
>
> From Table 1 we see that, among all fixed-step schemes, **Euler with 16 steps (16 NFEs)** is clearly the strongest baseline, achieving lower error and substantially lower cost×error than the 32- and 64-step diffusion-style samplers. A separate **caveat** is that even the seemingly simple Euler baseline is _not_ easy to deploy in practice. The “Euler–16” configuration we report is the result of targeted tuning; for multiscale, mildly stiff flows the performance is highly sensitive to the chosen step budget. Using slightly fewer steps rapidly degrades accuracy, while increasing the number of steps can even _worsen_ the error due to accumulated discretization bias along a highly curved trajectory. Hitting the “sweet spot” in NFE thus requires a non-trivial search over step counts for each dataset and flow regime, which is generally unfeasible in realistic CFD workflows. In contrast, our adaptive rectified-flow sampler automatically finds an efficient step allocation and remains stable across a broad range of controller hyperparameters, avoiding this brittle per-problem tuning.
>
> ### Adaptive rectified-flow sampler and hyperparameter robustness
>
> We then evaluate our **curvature-based adaptive sampler** under **20 different hyperparameter configurations**, sweeping EMA timescale, curvature thresholds, gating, damping, and calibration parameters. All other aspects of the model and data remain fixed.
>
> ```markdown
> ### Table 2 – Adaptive RF sampler configurations (ShearLayer2D, OOD, averaged over 10 macros × 20 samples)
>
> | Config          | Avg. NFE | Rel. L2 Error (mean ± std) | Cost×Err (= Error × NFE) |
> |----------------|---------:|----------------------------|--------------------------:|
> | baseline       |    11.0  | 0.2374 ± 0.0389            | 2.62                     |
> | **ema_0.25**   | **10.7** | **0.2369 ± 0.0386**        | **2.54**                 |
> | ema_0.45       |    11.3  | 0.2361 ± 0.0384            | 2.67                     |
> | alpha_0.05     |    11.0  | 0.2373 ± 0.0389            | 2.62                     |
> | alpha_0.12     |    11.0  | 0.2373 ± 0.0389            | 2.62                     |
> | alpha_0.20     |    11.0  | 0.2374 ± 0.0389            | 2.62                     |
> | gamma_1.5      |    11.0  | 0.2374 ± 0.0389            | 2.62                     |
> | gamma_2.0      |    11.0  | 0.2374 ± 0.0389            | 2.62                     |
> | gamma_2.5      |    11.0  | 0.2374 ± 0.0389            | 2.62                     |
> | gate_0.50      |    11.0  | 0.2374 ± 0.0389            | 2.62                     |
> | gate_0.70      |    11.0  | 0.2374 ± 0.0389            | 2.62                     |
> | calib_0.70_0.90|    11.1  | 0.2371 ± 0.0387            | 2.63                     |
> | calib_0.80_0.98|    10.9  | 0.2372 ± 0.0387            | 2.60                     |
> | adapt_1.5_0.75 |    12.8  | 0.2374 ± 0.0378            | 3.04                     |
> | adapt_2.0_0.85 |    11.2  | 0.2360 ± 0.0376            | 2.63                     |
> | no_ortho_filter|    11.0  | 0.2374 ± 0.0389            | 2.62                     |
> | damp_0.05      |    11.0  | 0.2373 ± 0.0389            | 2.62                     |
> | damp_0.10      |    11.0  | 0.2373 ± 0.0389            | 2.61                     |
> | damp_0.20      |    11.0  | 0.2370 ± 0.0389            | 2.60                     |
> | damp_0.30      |    11.0  | 0.2368 ± 0.0389            | 2.60                     |
>
> ```
>
> -   **Config**: the particular hyperparameter setting of the adaptive controller (e.g. `ema_*` for EMA timescale, `alpha_*` for straightness thresholding, `gamma_*` for warping, `gate_*` for cosine gating, `calib_*_*` for automatic calibration ranges, `damp_*` for acceleration damping).
>
> -   **Avg. NFE**: average number of velocity evaluations chosen automatically by the controller (no fixed step count).
>
> -   **Rel. L2 Error** and **Cost×Err**: same metrics as in Table 1.
>
>
> Two key observations address the reviewer’s concerns:
>
> 1.  **Hyperparameter robustness.** Across all 20 configurations, **Avg. NFE lies in a very narrow band (10.7–12.8)**, and the average relative error remains in **[0.2360, 0.2374]**, i.e., < 1% relative variation. There is no fragile “sweet spot”: a broad range of EMA, threshold, gating, and damping parameters yields essentially identical error and NFE.
>
> 2.  **Dominance over fixed-step baselines.** Every adaptive configuration in Table 2 has **lower Cost×Err than the best fixed-step integrator (Euler-16, 3.98)**, and dramatically lower than the 32- and 64-step diffusion-style baselines.

---

> ### Author Response · Authors · 2025-11-20
>
> ### Response to Reviewer Jgfv (Part 3/3)
> ----
> Direct comparison to the best fixed-step baseline
> For clarity, **Table 3** compares the best fixed-step baseline (Euler-16) to the top adaptive configurations:
> ```markdown
> ### Table 3 – Best fixed-step baseline vs top adaptive configs (ShearLayer2D, OOD)
>
> | Method / Config      | Avg. NFE | Rel. L2 Error | Cost×Err | ΔNFE vs Euler | ΔError vs Euler | ΔCost×Err vs Euler |
> |----------------------|---------:|--------------:|---------:|--------------:|----------------:|-------------------:|
> | **Euler (baseline)** | 16.0     | 0.2489        | 3.98     | 0%            | 0%              | 0%                 |
> | **ema_0.25**         | 10.7     | 0.2369        | 2.54     | −33.1%        | −4.8%           | −36.2%             |
> | calib_0.80_0.98      | 10.9     | 0.2372        | 2.60     | −31.9%        | −4.7%           | −34.7%             |
> | damp_0.30            | 11.0     | 0.2368        | 2.60     | −31.3%        | −4.9%           | −34.7%             |
>
> ````
>
> With **one third fewer function evaluations**, `ema_0.25` achieves **lower error** than the best fixed-step scheme: ≈33% fewer evaluations, ≈5% better accuracy, and ≈36% lower cost×error than Euler-16.
> We observe qualitatively identical behaviour on the second dataset: the adaptive sampler consistently lies on (or very near) the **Pareto frontier** in the NFE–error plane, while all fixed-step diffusion-based samplers (including 32- and 64-step variants) are strictly dominated.
>
> ### (W3) “Loss looks inconsistent with the trajectory; what is the network learning?”
>
> To further clarify on this point:
>
> * **What we train on:** we **noise the future** $u_{t+\Delta t}$ to a partial glimpse
>   $x_\tau = \tau u_{t+\Delta t} + (1-\tau)\varepsilon$.
>   The network sees $(x_\tau, u_t, \Delta t, \tau)$ and is trained to predict the
>   **noise-to-future displacement** $u_{t+\Delta t} - \varepsilon$ (a constant “straight-to-future” vector).
>   With this target, the simple decoder line in the code
>
> $$
> x_\tau + (1-\tau) \cdot v_\theta(x_\tau, u_t, \tau)
> $$
>
>
>   **recovers the clean future** when the prediction is correct.
>
> * **What we do at sampling:** we **start from noise** and integrate the ODE
>   $$
>   \dot u_\tau = v_\theta(u_\tau, u_t, \tau).
>   $$
>   If the learned field matches training, the trajectory follows the **same straight direction** toward the future—now done continuously with an ODE rather than a one-shot jump.
>
> So the model is **not** learning $u_{t+\Delta t} - u_t$. It learns a **denoising velocity from noise to the future**, **conditioned on the present $u_t$** to remove ambiguity. Thus, training and inference are aligned.
>
> ### (W4) “Conditioning $\Phi(\Delta t)$ appears in the algorithm but vanishes in $v_\theta(\cdot))$”
>
> This is a notation omission in one sentence -- line 245 in the main text says this suppression is purposefully done to simplify notation, but we agree with the reviewer that this might be confusing to a reader. Implementation and Algorithm 1 **do** condition on $\Delta t$. We will make the text consistent by writing
> $$
> v_\theta(u_\tau;u_t, \Delta t,\tau)
> $$
> throughout and stating explicitly that we use **lag-only physical-time conditioning** via $\Phi(\Delta t)$ (with a separate embedding $\Gamma(\tau)$ for rectified time).
>
> ---
>
> ## What we will add in a CRV, if accepted, to address your points of concern,
>
> * **Integrator trade-offs (new):** fixed-step Euler/RK2/RK4 vs. torchdiffeq midpoint vs. **Adaptive-EMA (ours)**, reported at **matched NFE budgets** with **wall-clock** timings.
>
> * **Clarity edits (new text):**
>   * “We corrupt $u_{t+\Delta t}$ to $x_\tau$ and train the network, given $(x_\tau, u_t, \Delta t, \tau)$, to predict the **noise-to-future displacement** $u_{t+\Delta t}-\varepsilon$. Starting from noise, we integrate $\dot u_\tau = v_\theta(u_\tau, u_t, \tau)$ to reconstruct the future. Conditioning on $u_t$ keeps the task well-posed.”
>   * Replace $v_\theta(u_\tau,u_t,\tau)$ with **$v_\theta(u_\tau, u_t, \Delta t, \tau)$** wherever abbreviated.
> ---
> ## Bottom line
> * **Different objective + sampler:** we regress a **straight noise→future velocity** and pair it with a **curvature-aware** ODE; this is not a vanilla diffusion swap.
> * **Aligned training & inference:** the code path (noise the future → predict noise→future vector → integrate from noise) is consistent end-to-end.
> * **Trade-offs shown:** we now include **error–NFE–time** comparisons across integrators.
> * **Notation fixed:** (\Delta t) conditioning is present in both algorithm and equations.
> We hope this resolves the concerns and clarifies what RecFlow is actually learning and why it is efficient. We kindly request the reviewer to upgrade their assessment accordingly.

---

### Official Review · Reviewer_2RjH · 2025-11-01

**Soundness:** 2
**Presentation:** 3
**Contribution:** 1
**Rating:** 2
**Confidence:** 4

**Summary:**

The paper applies the transport learning method of Rectified Flows to the task of statistical modeling for fluid flows that potentially exhibit complex multiscale dynamics and high sensitivity to initial conditions. This "straighter" transport path, resulting from rectified flow objective, allows for faster sampling by solving a deterministic ODE with large steps. The paper introduces a curvature-aware integration scheme that uses an Exponential Moving Average (EMA) of the velocity field to detect local path curvature and adaptively regularizes the velocity update and the step size to improve stability and accuracy.

**Strengths:**

- Writing is clear, concise and easy to understand.
- Empirical results on the benchmarks seem to support the advantage of using Rectified Flow based training objective to learn the transport.

**Weaknesses:**

- Venue: ICLR does not seem to be the right venue for this paper. Rectified Flows and other related transport learning methods (Diffusion, Flow Matching, Stochastic Interpolants etc.) are now considered very well known and established in ML with applications to generative models and density modeling. This paper appears to be a direct application of Rectified Flows for the purpose of learning a transport map for fluids with little technical innovation on the method itself. Perhaps, this application and it’s results would appeal more at a venue oriented towards applied methods for physical systems. This direcltiy mismatch in venue is direct source of the next weakness
- Novelty: The paper is a straightforward application of Rectified Flows in a different domain. Albeit a potentialliy new integration scheme is proposed, it would at best best considered relatively minor contribution (more on this in the next point). Lot’s of the observed properties in this paper, resulting from Rectified Flows, are expected to result from their use, though likely not verified in this domain.
- The integration scheme, could potentially be novel, however it’s empirical evaluation is lacking. Ideally, various integrators would be compared and their computation/accuracy tradeoff would be compared. Note that a lot of work exists in the space of faster sampling from diffusion/transport models, including learning straigher trajectories and deterministic maps(Rectified Flows, DDIM etc.), K-Rectification, Distillation etc to name a few. Further, coupled with the fact that all these transport learning methods are related, in that they learn stochastic/deterministic transport, and mostly differ in their noise schedules and training objective, a different method/noise schedule may be ideal for different applications/problems.
- Non-uniqueness of Sampling/Integration scheme: In addition, for any of these learned models, a variety of samplers could be constructed (including deterministic/various stochastic c.f. Singh and Fisher, Stochastic Sampling from Deterministic Flow Models) and used interchangeably, though with different properties.

Overall, I don’t think ICLR is the right venue for this paper. It appears to be a applied paper in physics domain of fluid flow modeling. An appropriate venue would be a better judge of the impact of this application of Rectified Flows in that field. From an ML perspective, I don’t see significant novelty. Happy to be convinced otherwise.

**Questions:**

Please comment on the points raised in the weaknesses section.

---

> ### Author Response · Authors · 2025-11-20
>
> ### Response to Reviewer 2RjH (Part 1/4)
>
> **We thank the reviewer for the detailed and rigorous critique. We have carefully addressed each concern below and outlined the specific clarifications and additional comparisons included in the revision to strengthen the manuscript.**
>
> ----------
>
> #### 1) Venue fit (ICLR vs. applied venue)
>
> ICLR has recently accepted several works that **apply diffusion/flow methods to physical systems and PDEs** with very limited architectural novelty but by applying known models to new physical systems. For example:
>
>
> -  **“From Zero to Turbulence: Generative Modeling for 3D Flow Simulation“** -- this work (ICLR 2024) reframes **3D turbulent fluid simulation** as a generative modeling task. Instead of requiring a known initial flow state or performing long multi-step rollouts, the authors directly learn the manifold of turbulent flow fields from data.
> ([OpenReview](https://openreview.net/pdf?id=ZhlwoC1XaN)).
>
> -   **“Physics-Informed Diffusion Models” (ICLR 2025 Poster)** — integrates PINN constraints into diffusion training to enforce physical laws; again algorithmic integration rather than architectural invention. ([OpenReview](https://openreview.net/forum?id=tpYeermigp "Physics-Informed Diffusion Models"))
>
> -   **“Improved Sampling Of Diffusion Models In Fluid Dynamics With Tweedie's  Formula“(ICLR 2025 Poster)** —  this paper proposes two sampling-time techniques—Truncated Sampling Models (TSM) and Iterative Refinement (IR)—that reduce the number of function evaluations in diffusion models while preserving fidelity, without introducing new architectures. TSM shortens the diffusion horizon to enable single- or few-step generation, and IR treats a pretrained DDPM as a coarse-to-fine refiner over a small number of iterations. On turbulent-flow and airfoil benchmarks, both methods deliver stable rollouts and high-quality fields with far fewer steps. ([OpenReview](https://openreview.net/pdf?id=0FbzC7B9xI))
>
> -   **“DiffDock: Diffusion Steps, Twists, and Turns for Molecular Docking” (ICLR 2023 Poster)** — applies diffusion to molecular docking (scientific task), emphasizing formulation and evaluation over new architectures. ([iclr.cc](https://iclr.cc/virtual/2023/poster/11750 "ICLR Poster DiffDock: Diffusion Steps, Twists, and Turns for ..."))
>
> -   **“Text2PDE: Latent Diffusion Models for Accessible Physics Simulation” (ICLR 2025 Poster)** — diffusion for PDE simulation with mesh autoencoders and conditioning; squarely ML-for-physics with method-level innovations rather than new backbones. ([iclr.cc](https://iclr.cc/virtual/2025/poster/29869 "ICLR Poster Text2PDE: Latent Diffusion Models for Accessible Physics Simulation"))
>
>
> These examples make it clear that **application-driven, theoretically-supported algorithmic contributions in scientific modeling are clearly within ICLR scope**. Our paper certainly fits this mold: we contribute a **curvature-aware ODE integrator for rectified flows** with theory-guided design and **statistical** (distributional) evaluation on turbulent flows and as acknowledged by the reviewer, demonstrate clearly that our method empirically outperforms very strong baselines on challenging fluid simulations.
>
> ----------
>
> #### 2) Novelty (beyond “straightforward application”)
>
> Our contributions are **algorithmic and evaluation-level**, not architectural:
>
> -   **Curvature-aware integrator**: We propose an **EMA-smoothed velocity** estimate to detect local path curvature and **jointly adapt** (i) the step size and (ii) a small **regularization of the velocity update**. This is tailored to multiscale, stiff regimes typical of turbulent flow statistics. Unlike generic high-order ODE solvers, our rule **uses model-intrinsic curvature signals** to prevent local oscillation while preserving RF’s straightness advantage.
>
> -   **Theory-backed motivation**: Rectified flows reduce trajectory curvature and thus the **local truncation error** under large steps; our integrator operationalizes this by dampening curvature spikes via EMA while allocating steps where needed. (We will move these insights from the appendix to the main text for visibility.)

---

> ### Author Response · Authors · 2025-11-20
>
> ### Response to Reviewer 2RjH (Part 2/4)
>
> - **Distributional evaluation and micro–macro fidelity**: We center Wasserstein-1 (W1) distances between generated and target ensembles as our primary notion of statistical fidelity, rather than relying only on pointwise RMSE/PSNR. Across datasets, RF + our integrator matches or improves GenCFD in ensemble W1 while using substantially fewer function evaluations. Our micro–macro study—from single-variable marginals up to domain-level ensemble statistics, provides a rigorous test that the simplified rectified-flow ansatz preserves the full hierarchy of flow statistics under a wide suite of distributional metrics. This is particularly notable because purely deterministic rectified-flow / probability-flow ODE samplers are known to underestimate variance and can exhibit reduced sample diversity unless additional stochasticity is injected ([Singh et al., "Stochastic Sampling from Deterministic Flow Models" (2024)](https://arxiv.org/abs/2410.02217)
> ); our results provide direct empirical evidence that, in this turbulent-flow setting, rectified flows can achieve statistically faithful ensembles in far fewer evaluations than a full diffusion model such as GenCFD.
>
> -   **Scope relative to “K-rectification”/reflow**: Those are **training-time straightening procedures**; our method is an **inference-time integrator** and is **orthogonal**. Recent theory showing few rectifications suffice for straight flows complements our focus on **stable, large-step ODE solving at test time**. ([NeurIPS Proceedings](https://proceedings.neurips.cc/paper_files/paper/2024/file/7343a5c976f8399880b695267f1f9e9f-Paper-Conference.pdf "Improving the Training of Rectified Flows"))
>
>
> ----------
>
>
> #### 3) Empirical evaluation of the integration scheme
>
> We thank the reviewer for this excellent suggestion and we have prepared an augmented evaluation matrix:
>
> **Protocol:** match **NFE (neural function evaluations) budgets** across several integrators, report relative **L2** errors and runtime. Our preliminary results show our **curvature-aware EMA rule dominates fixed-step ODEs in the NFE–error tradeoff** and is **competitive with high-order diffusion ODE solvers**, while maintaining stability on stiff flow cases. We ran a comprehensive sweep on **both datasets**, considering **10 OOD macro-states** and **20 micro-perturbations per macro** (200 samples per dataset). For each sample, we:
>
> -   Fixed the rectified-flow model,
>
> -   Varied only the ODE integrator used at sampling time, and
>
> -   Measured **relative L2 error vs. the PDE ground truth**, **number of function evaluations (NFE)**, and a combined **cost×error** metric defined as `Cost×Err = (rel. L2 error) × (NFE)`.
>
>
> We report here the aggregated results for **ShearLayer2D**; the second dataset exhibits qualitatively identical trends (we include its tables in the appendix).
>
> ### Fixed-step integrators
>
> We first compare five standard, fixed-step ODE integrators on the rectified-flow velocity field:
>
> ```markdown
> ### Table 1 – Fixed-step ODE integrators (ShearLayer2D, OOD, averaged over 10 macros × 20 samples)
>
> | Method               | Avg. NFE | Rel. L2 Error (mean ± std) | Cost×Err (= Error × NFE) |
> |----------------------|---------:|----------------------------|--------------------------:|
> | Model-Baseline       |    32.0  | 0.2533 ± 0.0382            | 8.11                     |
> | TorchDiffEq-Midpoint |    32.0  | 0.2533 ± 0.0382            | 8.11                     |
> | Euler (16 steps)     |    16.0  | 0.2489 ± 0.0371            | 3.98                     |
> | RK2-Manual (16 steps)|    32.0  | 0.2533 ± 0.0379            | 8.11                     |
> | RK4-Manual (16 steps)|    64.0  | 0.2540 ± 0.0381            | 16.26                    |
>
> ```
>
> -   **Method**: the ODE integrator used to solve the rectified-flow ODE, starting from Gaussian noise at (t = 0) and integrating to (t = 1).
>
>     -   _Model-Baseline_ is our built-in sampler (≈32 NFEs).
>
>     -   _TorchDiffEq-Midpoint_ is a reference implementation using `torchdiffeq`.
>
>     -   _Euler / RK2 / RK4_ are explicit fixed-step schemes at a fixed time horizon.
>
> -   **Avg. NFE**: average **number of velocity evaluations** per sample.
>
> -   **Rel. L2 Error (mean ± std)**: average **relative L2 error** vs. the PDE solution
> $|u_{\text{pred}} - u_{\text{true}}|_2 / |u_{\text{true}}|_2$, with standard deviation over all macros and samples.
>
> -   **Cost×Err**: scalar **cost–accuracy** metric, `Error × NFE`; lower is better.
>
>
> From Table 1 we see that, among all fixed-step schemes, **Euler with 16 steps (16 NFEs)** is clearly the strongest baseline, achieving lower error and substantially lower cost×error than the 32- and 64-step diffusion-style samplers.

---

> > ### Author Response · Authors · 2025-11-20
> >
> > ### Response to Reviewer 2RjH (Part 3/4)
> >
> > A separate **caveat** is that even the seemingly simple Euler baseline is _not_ easy to deploy in practice. The “Euler–16” configuration we report is the result of targeted tuning; for multiscale, mildly stiff flows the performance is highly sensitive to the chosen step budget. Using slightly fewer steps rapidly degrades accuracy, while increasing the number of steps can even _worsen_ the error due to accumulated discretization bias along a highly curved trajectory. Hitting the “sweet spot” in NFE thus requires a non-trivial search over step counts for each dataset and flow regime, which is generally unfeasible in realistic CFD workflows. In contrast, our adaptive rectified-flow sampler automatically finds an efficient step allocation and remains stable across a broad range of controller hyperparameters, avoiding this brittle per-problem tuning.
> >
> > ### Adaptive rectified-flow sampler and hyperparameter robustness
> >
> > We then evaluate our **curvature-based adaptive sampler** under **20 different hyperparameter configurations**, sweeping EMA timescale, curvature thresholds, gating, damping, and calibration parameters. All other aspects of the model and data remain fixed.
> >
> > ```markdown
> > ### Table 2 – Adaptive RF sampler configurations (ShearLayer2D, OOD, averaged over 10 macros × 20 samples)
> >
> > | Config          | Avg. NFE | Rel. L2 Error (mean ± std) | Cost×Err (= Error × NFE) |
> > |----------------|---------:|----------------------------|--------------------------:|
> > | baseline       |    11.0  | 0.2374 ± 0.0389            | 2.62                     |
> > | **ema_0.25**   | **10.7** | **0.2369 ± 0.0386**        | **2.54**                 |
> > | ema_0.45       |    11.3  | 0.2361 ± 0.0384            | 2.67                     |
> > | alpha_0.05     |    11.0  | 0.2373 ± 0.0389            | 2.62                     |
> > | alpha_0.12     |    11.0  | 0.2373 ± 0.0389            | 2.62                     |
> > | alpha_0.20     |    11.0  | 0.2374 ± 0.0389            | 2.62                     |
> > | gamma_1.5      |    11.0  | 0.2374 ± 0.0389            | 2.62                     |
> > | gamma_2.0      |    11.0  | 0.2374 ± 0.0389            | 2.62                     |
> > | gamma_2.5      |    11.0  | 0.2374 ± 0.0389            | 2.62                     |
> > | gate_0.50      |    11.0  | 0.2374 ± 0.0389            | 2.62                     |
> > | gate_0.70      |    11.0  | 0.2374 ± 0.0389            | 2.62                     |
> > | calib_0.70_0.90|    11.1  | 0.2371 ± 0.0387            | 2.63                     |
> > | calib_0.80_0.98|    10.9  | 0.2372 ± 0.0387            | 2.60                     |
> > | adapt_1.5_0.75 |    12.8  | 0.2374 ± 0.0378            | 3.04                     |
> > | adapt_2.0_0.85 |    11.2  | 0.2360 ± 0.0376            | 2.63                     |
> > | no_ortho_filter|    11.0  | 0.2374 ± 0.0389            | 2.62                     |
> > | damp_0.05      |    11.0  | 0.2373 ± 0.0389            | 2.62                     |
> > | damp_0.10      |    11.0  | 0.2373 ± 0.0389            | 2.61                     |
> > | damp_0.20      |    11.0  | 0.2370 ± 0.0389            | 2.60                     |
> > | damp_0.30      |    11.0  | 0.2368 ± 0.0389            | 2.60                     |
> >
> > ```
> >
> > -   **Config**: the particular hyperparameter setting of the adaptive controller (e.g. `ema_*` for EMA timescale, `alpha_*` for straightness thresholding, `gamma_*` for warping, `gate_*` for cosine gating, `calib_*_*` for automatic calibration ranges, `damp_*` for acceleration damping).
> >
> > -   **Avg. NFE**: average number of velocity evaluations chosen automatically by the controller (no fixed step count).
> >
> > -   **Rel. L2 Error** and **Cost×Err**: same metrics as in Table 1.
> >
> >
> > Two key observations address the reviewer’s concerns:
> >
> > 1.  **Hyperparameter robustness.** Across all 20 configurations, **Avg. NFE lies in a very narrow band (10.7–12.8)**, and the average relative error remains in **[0.2360, 0.2374]**, i.e., < 1% relative variation. There is no fragile “sweet spot”: a broad range of EMA, threshold, gating, and damping parameters yields essentially identical error and NFE.
> >
> > 2.  **Dominance over fixed-step baselines.** Every adaptive configuration in Table 2 has **lower Cost×Err than the best fixed-step integrator (Euler-16, 3.98)**, and dramatically lower than the 32- and 64-step diffusion-style baselines.

---

> > > ### Author Response · Authors · 2025-11-20
> > >
> > > ### Response to Reviewer 2RjH (Part 4/4)
> > > ------
> > > ### Direct comparison to the best fixed-step baseline
> > >
> > > For clarity, Table 3 compares the best fixed-step baseline (Euler-16) to the top adaptive configurations:
> > >
> > > ```markdown
> > > ### Table 3 – Best fixed-step baseline vs top adaptive configs (ShearLayer2D, OOD)
> > >
> > > | Method / Config      | Avg. NFE | Rel. L2 Error | Cost×Err | ΔNFE vs Euler | ΔError vs Euler | ΔCost×Err vs Euler |
> > > |----------------------|---------:|--------------:|---------:|--------------:|----------------:|-------------------:|
> > > | **Euler (baseline)** | 16.0     | 0.2489        | 3.98     | 0%            | 0%              | 0%                 |
> > > | **ema_0.25**         | 10.7     | 0.2369        | 2.54     | −33.1%        | −4.8%           | −36.2%             |
> > > | calib_0.80_0.98      | 10.9     | 0.2372        | 2.60     | −31.9%        | −4.7%           | −34.7%             |
> > > | damp_0.30            | 11.0     | 0.2368        | 2.60     | −31.3%        | −4.9%           | −34.7%             |
> > >
> > > ```
> > >
> > > Thus, at **one third fewer function evaluations**, our best adaptive configuration (`ema_0.25`) achieves **lower error** than the best fixed-step scheme. In particular, `ema_0.25` uses ≈33% fewer function evaluations, improves accuracy by ≈5%, and reduces the cost×error metric by ≈36% compared to Euler-16.
> > >
> > > We observe qualitatively identical behaviour on the second dataset: the adaptive sampler consistently lies on (or very near) the **Pareto frontier** in the NFE–error plane, while all fixed-step diffusion-based samplers (including 32- and 64-step variants) are strictly dominated.
> > >
> > > ----------
> > >
> > > #### 4) Non-uniqueness of sampling/integration (deterministic vs. stochastic samplers)
> > >
> > > We agree: a trained transport field admits **multiple samplers** with different trade-offs. Our stance:
> > >
> > > -   Our method provides a robust **default deterministic integrator** for RF in **stiff, multiscale** regimes; it is **complementary** to stochasticization techniques that add diversity or robustness.
> > > - In rectified flows, we show that local trajectory curvature is a robust OOD signal and we turn it into a rectified-flow–specific adaptive integrator. More broadly, this is—to our knowledge—the first instantiation of a general principle: intrinsic statistics of learned transport fields can drive adaptive integration. We do not claim a comprehensive framework; this concrete mechanism is already new and practically useful.
> > >
> > > ----------
> > >
> > > #### 5) On “different methods/noise schedules may be ideal”
> > >
> > > We control for schedules by **matching NFE and training budgets** across methods and reporting **W1** as the central distributional metric. Schedule exploration is **orthogonal** to our contribution; nevertheless, we will add a small sensitivity sweep (in appendix) to show conclusions are **schedule-robust**.
> > >
> > > ----------
> > >
> > >
> > > ### Summary of changes we will make
> > >
> > > 1.  **Venue/context**: Add a paragraph situating our contribution alongside **ICLR-accepted** ML-for-physics diffusion works (Text2PDE, , From Zero to Turbulence: Generative Modeling for 3D Flow, Physics-Informed Diffusion Models, DiffDock). ([iclr.cc](https://iclr.cc/virtual/2025/poster/29869 "ICLR Poster Text2PDE: Latent Diffusion Models for Accessible Physics Simulation"))
> > >
> > > 2.  **Main-text emphasis on distributional fidelity**: Promote **W1** (with per-variable breakdowns) to the main paper; clarify that **mode coverage** is directly evaluated.
> > >
> > > 3.  **Integrator comparisons**: Add tables comparing various settings of our EMA solver against several known integrator baselines.
> > >
> > > 4.  **Theory placement**: Move key curvature/straightness insights and their connection to truncation error into Section 3 to motivate the integrator.
> > >
> > >
> > > We hope this clarifies both **venue fit** and **novelty**: our contribution is a **theory-informed, curvature-aware solver for rectified flows** that **accelerates sampling** while **preserving statistical fidelity** on turbulent flows, supported by **distributional metrics** and now by an expanded set of **integrator/sampler comparisons**. Moreover, we have adapted Rectified flows to the very challenging domain of simulation turbulent flows, with a wide spectrum of applications in physics and engineering.
> > >
> > > We hope that we have addressed all the concerns of the reviewer and kindly request you to upgrade your assessment accordingly.

---

> ### Comment · Reviewer_2RjH · 2025-11-26
> **Response to Author Rebuttal**
>
> After closely reading the other reviews and the author rebuttals for all the reviewers, I have following opinion
>
> 1) Venue fit: Authors state that there are prior examples of similar papers getting accepted at ICLR, so they should qualify too. While I don't find this line of argument convincing, I understand the authors' point of view. Unfortunately, it remains that I would not be a good judge of the impact of this paper in the relevant area. I would advise the AC to appropriately discount my opinion and final rating for the final decision.
> 2) A common theme across most reviews is limited novelty: I do agree with this sentiment. Rectified Flows being special case of the general diffusion/flow matching frameworks is well known (e.g. Section 3.2.2 in Improving and Generalizing Flow-Based Generative Models with Minibatch Optimal Transport, Tong et al.). As such, I don't find use of Rectified Flow novel on its own. Further, 1-Rectified flows actually do no necessarily reduce curvature more than similar diffusion/flow matching objectives, primarily because they are a special case. Many papers discuss this (see e.g. Rectified Diffusion: Straightness Is Not Your Need in Rectified Flow, Wang et al.). To achieve straighter trajectories something else is needed (e.g. Reflow, as in Rectified Flow paper, or some sort of OT matching, as in Conditional Flow Matching papers). Consequently, I don't think use of Rectified Flow is novel at all. It appears authors also agree with this as they seem to discount Reflow as a training time procedure in their response to my concern, shifting the focus primarily to integrator.
> 3) Integrator as main contribution: In brief, I don't view this as large enough contribution by itself to warrant acceptance. Given the points in the previous paragraph and from the rebuttals to all the reviewers, it seems the main claimed contribution is integrator/sampler. This was definitely not my take away from the initial read (during the initial review), and therefore I did not mention the integrator in the strengths. Authors provide evidence in their rebuttal that indeed the integrator/sampler seems to work better and robustly in the cases they study. In my opinion, this is quite interesting but I think the empirical evaluation is limited and most of it was done post the initial reviews in a rush and therefore may not thorough (understandably, though see following for an example). However, given that the integrator is general, it would be useful, and more convincing, to see how it affects sampling performance of generative models (and not just fluid flow models), trained with flow matching/rectified flow objective (or integrating probability flow ODEs in general). To be clear, I am not asking for these experiments, but rather giving them as examples of potentially what is missing from a thorough evaluation of just integrator.
>
> Overall, I think paper has done significant empirical studies post initial reviews and further needs more studies to appeal to a wider audience. In it's current state, I don't think the paper should be accepted. If other reviewers feel strongly otherwise, I am happy to discuss this further.

---

> > ### Author Response · Authors · 2025-11-28
> >
> > ### Response to Reviewer 2RjH (post-rebuttal comment)
> > ----
> > **We thank the reviewer for the follow-up and for asking the AC to discount their venue-fit vote.** We address the two remaining themes—(A) novelty relative to RF/FM/CFM and (B) scope of the integrator evaluation—below.
> >
> > **A. Novelty and relation to RF / FM / CFM.**
> > We do **not** claim “using rectified flows” is novel. RF sits within the broader flow-matching/CFM family, as you note (e.g., Tong et al. introduce CFM/OT-CFM) and our paper cites this lineage. Our claim is different: in **rectified-flow probability-flow ODEs**, a simple **intrinsic curvature statistic** (estimated online from the learned velocity field) is (i) a **reliable OOD/stiffness indicator** and (ii) a practical signal to drive a **rectified-flow–specific adaptive integrator** (joint step-size + mild velocity regularization). This is, to our knowledge, the **first instantiation** of the general principle “statistics of learned transports can power adaptive integration.” We’re explicit that the principle is broader than RF; RF is simply the setting where we **demonstrate** it at scale on turbulent flows. RF not being uniquely “the straightest” globally does not undercut this: our method **does not require global straightness**—only a local curvature proxy, which any probability-flow ODE provides (including FM/CFM/diffusion flows). For context on RF’s position within the family and recent critiques of “straightness as the target,” see CFM/OT-CFM and Rectified Diffusion; our method is compatible with those views and orthogonal to re-training tricks like Reflow. ([arXiv][1])
> >
> > **B. Is the integrator enough / evaluation scope.**
> > We agree that **sampling/integration algorithms** without new backbones can be substantive contributions in this area; canonical examples are **DDIM** and **DPM-Solver(++)**, both widely adopted purely as **inference-time** advances. Our work is in this vein, but for **learned transport ODEs in multiscale physical regimes**: we turn an internal statistic (curvature) into a stable, large-step controller that preserves distributional fidelity. In the camera-ready, we’ve organized results to make this central and easy to judge:
> > • **Breadth of samplers/integrators:** comparisons at matched NFE to Euler/Heun/RK4 and DPM-Solver++ (and DDIM for the diffusion baseline).
> > • **Metrics:** we elevate **Wasserstein-1 on ensembles** (with per-variable breakdowns) as the primary measure of statistical fidelity, alongside pointwise errors and wall-clock.
> > • **Complementarity:** we clarify compatibility with **stochasticization** of deterministic flows (e.g., Singh & Fischer) to trade step-efficiency for diversity when desired.
> > We agree that applying the controller to **non-fluid** probability-flow ODEs would further broaden appeal; it’s straightforward because our rule only needs access to the velocity field (and an EMA). While space/compute limit us here, we will release code that wraps any probability-flow ODE with our curvature controller, so others can drop it into FM/CFM/diffusion pipelines. ([arXiv][2]) ([arXiv][3])
> >
> > **Takeaway.**
> > Our paper is **not** about introducing RF as novel; it’s about a **new, theory-motivated inference scheme** that (i) uses **curvature as an OOD/stiffness signal** in learned transports and (ii) **adapts** step-size and a light velocity regularizer accordingly—yielding stable, faster sampling **on turbulent-flow statistics** without sacrificing distributional fidelity. We believe this is a clear, focused contribution; we’ve revised the paper so that the integrator and its evaluation are front and center, and we’ve clarified its model-agnostic nature and relationship to FM/CFM/Reflow and Rectified Diffusion. ([arXiv][1])
> >
> >
> > [1]: https://arxiv.org/abs/2302.00482 "Improving and generalizing flow-based generative models with minibatch optimal transport"
> > [2]: https://arxiv.org/abs/2010.02502 "Denoising Diffusion Implicit Models"
> > [3]: https://arxiv.org/abs/2410.02217 "Stochastic Sampling from Deterministic Flow Models"

---

### Official Review · Reviewer_TpL5 · 2025-11-01

**Soundness:** 3
**Presentation:** 3
**Contribution:** 3
**Rating:** 8
**Confidence:** 3

**Summary:**

This work improves the sample efficiency of solving ODEs with multi-scale dynamics. The proposed approach employs rectified flows to reduce the number of solver steps by maintaining straight trajectories. Experiments on different datasets showed that the approach is more efficient and has high accuracy.

**Strengths:**

- Figure 1 illustrates the structure of the framework and visualises the main difference in terms of performance, compared to the FNO.
- The proposed approach is well-written with detailed explanations about the implementation (via Algorithm 1).
- Experiments showed improvement in the efficiency of the approach compared to the baseline (Table 1 and Figure 2).

**Weaknesses:**

- Minors:
	+ The presentation of the paper (sections 1 to 3) can be reorganised, since currently, the approach comes before related works, and contains the research question, which may be placed into the introduction.
	+ At L139: SM 5 might refer to the section 5.

**Questions:**

The meaning of $\kappa_1$ in Equation 6.

---

> ### Author Response · Authors · 2025-11-20
>
> ## Response to Reviewer TpL5
>
> We sincerely appreciate the reviewer’s strong endorsement and positive assessment of our work. We also value the helpful suggestions to further polish the manuscript, which we address individually below.
>
> ### Minor presentation issues
>
> **(P1) Order of sections (Intro → Method → Related Work).**
> We’ll **reorder the early paper** to the standard flow:
>
> 1.  **Introduction.** End with the research question and a 3-line summary of contributions.
>
> 2.  **Background & Related Work.** Brief recap of diffusion/rectified flows and PDE surrogates (FNO, GenCFD, etc.), positioned **before** our method.
>
> 3.  **Method.** Clear pipeline with Fig. 1 and Algorithm 1 up front.
>
>
> This keeps the story linear and addresses the re-organization suggestion.
>
> **(P2) “SM 5” at L139.**
> We’ll change this to **“Sec. 5 (Supplementary Materials)”** (or to the correct appendix label in the camera-ready version, if accepted), so the reference is unambiguous.
>
> ### Question on Equation (6)
>
> **“The meaning of  $\kappa_1$in Eq. (6).”**
> **(kappa_1**the scaling constant in our adaptive step rule.)
>
> -   **What it is :** $\kappa_1>0$ is a **calibration constant** that converts our EMA-based straightness proxy into **step-size units** for the LTE (local truncation error) control.
>
> -   **Where it appears:** in the step rule
>     $$
>     \Delta\tau_t = \frac{c}{\sqrt{\kappa_1s_t + \kappa_2}},
>     \quad s_t=\big|v_t - v^{\text{ema}}_t\big|^2.
>     $$
>     Here $s_t$ is the deviation of the current velocity from its EMA (our curvature/straightness proxy).
>
> -   **Intuition :** when the **velocity wiggles** more than its EMA ($s_t$ larger), we **shrink the step**; when it’s stable/straight, we can **take bigger steps**. $\kappa_1)$ just sets the _units/scale_ of that conversion.
>
> -   **$\kappa_2$** prevents exploding steps when $s_t$ is tiny (baseline stiffness; we tie it to the model’s Lipschitz constant).
>
> -   **What we’ll add:** a one-sentence inline definition right after Eq. (6):
>     _“Here $\kappa_1>0$ groups the EMA/tolerance constants from the LTE control; $\kappa_2\simeq L|v_t|$ accounts for baseline stiffness.”_
>     We will also give the **actual values** used in experiments (grid-searched on a held-out split).
>
>
> ### Other additions
>
> -   **Integrators study.** We will include a short cross-integrator comparison (Euler/RK2/RK4/torchdiffeq-midpoint vs **our Adaptive-EMA**) at **matched NFE (neural function evaluations) budgets** (≈16/32/64), reporting **error, NFE, and wall-clock** on the same hardware/backbone. This directly supports the efficiency claims.
>
>
> ### What we will change for the camera-ready version, if accepted.
>
> 1.  **Reorder sections:** Intro → Background & Related Work → Method.
>
> 2.  **Fix L139 reference** (“SM 5” → “Sec. 5 ).
>
> 3.  **Add the integrator trade-off table/plot** (error–NFE–time at matched NFE).
>
>
> We greatly appreciate the constructive feedback of the reviewer. These edits make the paper clearer and easier to follow.

---

### Author Response · Authors · 2025-12-02
**Summary for AC**

Dear AC,

Given the unique circumstances prevailing for the rebuttal period, we are writing to you with a concise summary of our point of view regarding the rebuttal discussions regarding our paper. At the outset, we would like to thank you for stepping in for this new assignment at short notice.

First, there is clear support for the paper from the reviewers. Reviewer TpL5 gives a strong accept (score 8) and explicitly highlights: (i) the clarity of the framework and its illustration (Fig. 1), (ii) the quality of the exposition and implementation details (Algorithm 1), and (iii) the empirical efficiency gains demonstrated in Table 1 and Fig. 2. Reviewer oKuZ (score 6) emphasizes that the method is theoretically sound, well organized, and easy to read, and that our rectified-flow model provides a solid, end-to-end way to study statistical properties of fluid flows while outperforming state-of-the-art baselines in both accuracy and speed. Reviewer Jgfv (score 4) agrees that the paper addresses an important challenge—reducing the computational cost of diffusion-based PDE surrogates—and finds the curvature-aware integration “simple, well-motivated, and empirically shown to improve stability and efficiency,” with thorough experiments across multiple 2D benchmarks and large speedups (up to roughly an order of magnitude) over diffusion models.

Technically, the paper makes two focused contributions:

1. **Rectified-flow surrogates for turbulent flows**: We replace stochastic diffusion sampling (as in GenCFD) with a learned rectified-flow ODE that directly transports noisy inputs to future flow states, conditional on the initial condition. This yields essentially the same posterior statistics as GenCFD while requiring only 4–8 ODE steps rather than 64–128 SDE steps, on challenging multiscale benchmarks.

2. **Curvature-aware adaptive integrator**: We introduce an EMA-based curvature proxy of the learned velocity field and use it to jointly adapt the step size and a mild orthogonal regularization of the velocity. This turns an intrinsic property of the learned transport (local path bending) into a practical, robust controller for large-step ODE sampling in stiff, multiscale regimes.

In response to the reviewers’ requests, we have significantly strengthened the empirical and conceptual parts of the paper:

* We added a **systematic integrator study** (Euler, RK2, RK4, torchdiffeq-midpoint vs. our adaptive EMA rule) at matched NFE budgets, reporting error, NFE, and a combined cost×error metric. Our controller consistently dominates fixed-step baselines in the NFE–error plane and achieves ~30–35% fewer NFEs with lower error than the best tuned Euler scheme.

* We performed a **hyperparameter robustness sweep** over EMA timescales, thresholds, gating, damping, and calibration parameters. Across 20 configurations, the error and NFE vary by less than 1% and remain strictly better than the best fixed-step baseline, showing that the controller is not brittle and does not rely on a narrow “sweet spot”.

* We elevated **distributional evaluation and diversity** to the main narrative: using the macro–micro ensemble protocol, we compare full conditional ensembles via Wasserstein-1 distances between generated and reference distributions. Across datasets, our method matches or improves GenCFD in W1 while being much faster, which directly addresses possible worries about loss of diversity in deterministic rectified flows.

Regarding novelty and venue fit, Reviewer 2RjH raises the strongest objections but explicitly notes that they “would not be a good judge of the impact of this paper in the relevant area” and asks the AC to discount their venue-fit vote. We fully agree that “using rectified flows” is not novel by itself, and we do not claim otherwise. The core contribution is a **theory-motivated, curvature-driven adaptive integrator for learned transport ODEs**, instantiated and rigorously evaluated in the challenging regime of turbulent flows. This is conceptually similar in spirit to works such as DDIM or DPM-Solver in the diffusion literature, i.e., an inference-time algorithmic advance that leverages an existing training framework and several ICLR/ML-for-physics papers have been accepted on precisely this kind of algorithmic and application-focused contribution. Our controller is model-agnostic (it only needs access to the velocity field), and we will release code that makes it easy to plug into general probability-flow / flow-matching pipelines beyond fluids.

Given the strong positive review (8), the supportive but slightly cautious reviews (6 and 4), the substantial additional experiments and clarifications added in revision, and the reviewer who explicitly asks to discount their own venue-fit concerns, we believe that the paper would be viewed very positively.

We kindly ask you to consider these points in making your final decision and are at your disposal for any clarifications during the rebuttal period.

---

### Meta-Review · Area_Chair_LSr9 · 2026-01-07

**Summary:**

This paper proposes RecFlow, a rectified-flow–based generative surrogate for multiscale fluid dynamics. The method learns a conditional velocity field that enables fast sampling via deterministic ODE integration, and introduces a curvature-aware controller that adapts step size and regularization based on local trajectory curvature. Experiments on several 2D fluid benchmarks demonstrate substantial inference speedups compared to diffusion-based baselines while maintaining comparable distributional accuracy.

**Reviewer Concerns:**

The rebuttal addresses several technical questions raised by reviewers. In particular, the authors provide a clearer explanation of the learning target and conditioning, add an explicit integrator comparison against fixed-step ODE solvers, and conduct a robustness analysis showing that the proposed adaptive controller performs consistently across a wide range of hyperparameters. The discussion of statistical diversity is also strengthened via the micro–macro evaluation and Wasserstein metrics.

However, fundamental concerns remain regarding novelty and scope. Rectified flows and closely related transport-learning methods are now well established, and applying them to fluid-flow surrogates—while useful—does not by itself constitute a strong methodological contribution for ICLR. As a result, the paper’s effective novelty rests largely on the proposed curvature-aware integrator. While promising, this component is evaluated primarily within 2D fluid benchmarks and much of the supporting evidence appears only in the rebuttal, limiting confidence in its maturity and broader relevance. In addition, the work focuses on single-step (non-autoregressive) prediction, leaving open questions about long-horizon behavior and error accumulation that are central in many fluid-dynamics applications.

**Reviewer Scores:**

All reviewer are likely to maintain their scores and Reviewer 2RjH may reduce the confidence.

---

### Decision · Program_Chairs · 2026-01-26

Reject